# Cyclone generation Algorithm including a THERmodynamic module for Integrated National damage Assessment (CATHERINA 1.0) compatible with Coupled Model Intercomparison Project (CMIP) climate data

**Théo Le Guenedal**[1], **Philippe Drobinski**[2], **and Peter Tankov**[1]

[1]CREST, ENSAE, Institut Polytechnique de Paris, Palaiseau, France
[2]LMD-IPSL, Ecole Polytechnique, Institut Polytechnique de Paris, ENS, PSL Research University, Sorbonne Université, CNRS, Palaiseau, France

**Correspondence:** Théo Le Guenedal (theo.le-guenedal@ensae.fr)

**Abstract.** Tropical cyclones are responsible for a large share of global damage resulting from natural disasters, and estimating cyclone-related damage at a national level is a challenge attracting growing interest in the context of climate change. The global climate models, whose outputs are available from the Coupled Model Intercomparison Project (CMIP), do not resolve tropical cyclones. The Cyclone generation Algorithm including a THERmodynamic module for Integrated National damage Assessment (CATHERINA), presented in this paper, couples statistical and thermodynamic relationships to generate synthetic tracks sensitive to local climate conditions and estimates the damage induced by tropical cyclones at a national level. The framework is designed to be compatible with the data from CMIP models offering a reliable solution to resolve tropical cyclones in climate projections. We illustrate this by producing damage projections in representative concentration pathways (RCPs) at the global level and for individual countries. The algorithm contains a module to correct biases in climate models based on the distributions of the climate variables in the reanalyses. This model was primary developed to provide the economic and financial community with reliable signals allowing for a better quantification of physical risks in the long term, to estimate, for example, the impact on sovereign debt.

## 1 Introduction

Climate-related physical risks pose a growing threat to humanity, and the design and implementation of adequate adaptation and mitigation measures require assessment of future physical risks at a national and global scale. The projections of global climate models are an important source of information about the future climate. However, the spatial resolution of these models is unfortunately still not sufficient to fully resolve extreme events, particularly tropical cyclones. On the other side of the spectrum, integrated assessment models (IAMs) directly assess the impact of the future climate on economic activity. Although these models are used to calibrate optimal mitigation or adaptation pathways, most of them only embed a very simple physical damage module usually limited to a generic temperature damage function. The objective of this paper is to fill the gap between climate models and integrated assessments: we build synthetic cyclones based on the climate data produced by the global climate models and evaluate the economic damage of these synthetic cyclones under various assumptions regarding the socioeconomic scenarios. Tools to assess the impact of future cyclones in shared socioeconomic pathways are starting to appear in the literature. For example, Geiger et al. (2021) evaluate the population exposure. Our study instead focuses on the damage costs of tropical cyclones with the aim of including these advanced signals in integrated economic modeling.

The physics of tropical cyclones has been thoroughly studied in the literature. The thermodynamic cyclone theory builds upon the seminal contributions by Emanuel (1988) and followed by Holland (1997) and Emanuel (1999). Concerning the impact of climate change, it is well known that the presence of greenhouse gases in the atmosphere increases the radiative forcing, which leads to a progressive warming of the atmosphere (Butchart et al., 2000) and the rise in sea surface temperatures (Solomon et al., 2007; Pachauri et al., 2014). This phenomenon increases the amount of energy available for cyclones to grow in intensity (Emanuel, 1991), and this growth is already starting to be measurable (Emanuel, 2005). A 2010 review concluded that it is unclear whether past changes in tropical cyclone (TC) activity have exceeded the variability expected from natural causes (Knutson et al., 2010), while a more recent review (Walsh et al., 2016) confirms that studies are beginning to suggest climate trends of various kinds in TC data over the past few decades. This review states that no trend in average cyclone frequency or intensity has been detected globally, but a substantial increase in the proportion of intense cyclones has been demonstrated, both globally and in individual basins, with the exception of the North Pacific.

Models relying on statistical relationships (James and Mason, 2005; DeMaria and Kaplan, 1994; Kaplan and DeMaria, 1995) available in the literature produce synthetic cyclones with properties closely resembling those of observed cyclones. Recently, Bloemendaal et al. (2020) developed a modeling framework to simulate realistic synthetic tropical cyclone tracks: the Synthetic Tropical cyclOnes geneRation Model (STORM). This model computes the maximum pressure intensity (MPI) associated with the sea surface temperature (SST) and uses this potential as a predictor in the central pressure dynamics (James and Mason, 2005). In line with Merrill (1987), we find that, although the SST plays a major role, this variable alone is not a reliable predictor of whether a given storm will intensify. Thus, we prefer to rely on the formulation of Holland (1997) and to model the effect of climate change on the maximum potentials in the different scenarios through a better description of the underlying thermodynamic phenomenon, well described by Emanuel (1988), Holland (1997) or Emanuel (1999). We therefore develop an alternative to STORM by adding a thermodynamic module from the perspective of producing cyclones in different climate scenarios. In particular, we retrieve two additional variables (relative humidity and tropopause temperature) from climate models to bridge the gap between data from global circulation models (GCMs) and the theory of the intensification of tropical depressions.

Risk assessments have been developed based on hurricane potential intensity maps to assess the damage in the US and around the world (Emanuel et al., 2008; Emanuel, 2011; Mendelsohn et al., 2012). On the damage modeling side, Bresch (2017), Lüthi (2019), and Aznar Siguan and Bresch (2019) set up a platform for physical risk estimation

(CLIMADA), coupled with a database of estimated values of local assets (Eberenz et al., 2020a, b). The asset resolution (30 arcsec) and geospatial description of extreme events are particularly advanced. We propose a simplification of this work that applies in the context of national level assessment. The CLIMADA framework focuses on damage modeling based on global aggregated temperature projections and does not make use of the climate data produced by atmosphere–ocean general circulation models (AOGCMs). Coupling CLIMADA and STORM methodology with an extended thermodynamic module fitted on four climate variables, our approach provides a novel long-term tail risk assessment at a national level, offering an adaptive framework to estimate investments required to mitigate this risk.

This paper makes three contributions. First, we provide an algorithm to generate synthetic cyclones from climate data inspired by Bloemendaal et al. (2020) fitted on four physical variables extracted from ERA5 reanalysis and including a thermodynamic module to better describe cyclone physics. Second, we build an algorithm generating synthetic tracks directly from CMIP models, expose the biases in CMIP5 datasets and propose a correction module based on Vrac et al. (2012). Third, we bridge the gap between climate data and damage modeling by using the physical asset values from Eberenz et al. (2020b) and computing the damage along cyclone tracks using the region-specific damage functions designed in the CLIMADA project. Combining open data sources and methodologies allows us to propose a complete bottom-up integrated physical risk assessment model for tropical cyclones presented in Fig. 1: the Cyclone generation Algorithm including a THERmodynamic module for Integrated National damage Assessment (CATHERINA).

The process is the following. The cyclones are initiated with spatial and seasonal distribution estimated on the International Best Track Archive for Climate Stewardship (IBTrACS) database, similarly to Bloemendaal et al. (2020). Their movement is described with a simple autoregressive stochastic process (James and Mason, 2005; Bloemendaal et al., 2020). Along the simulated cyclone tracks, we retrieve climate variables from climate models and compute locally the maximum potential intensity based on the simplified expression in Holland (1997). Some controls such as the maximum pressure drop observed for the corresponding temperature and the decay relationship for cyclones evolving over land are also fitted for each basin and applied in the synthetic track-generation algorithm. Extracting the climate variables from different models allows us to correct the biases and evaluate model uncertainty. Next, we use the physical asset values (Eberenz et al., 2020b) and regional damage functions from CLIMADA to evaluate the cyclone-related damage at a national level. This step requires us to extract the local physical asset values and aggregate them on tiles of length defined proportionally to the average radius to the maximum wind (approximately 50 km) along the cyclone path. Summing the losses along tracks for each year and for each country allows

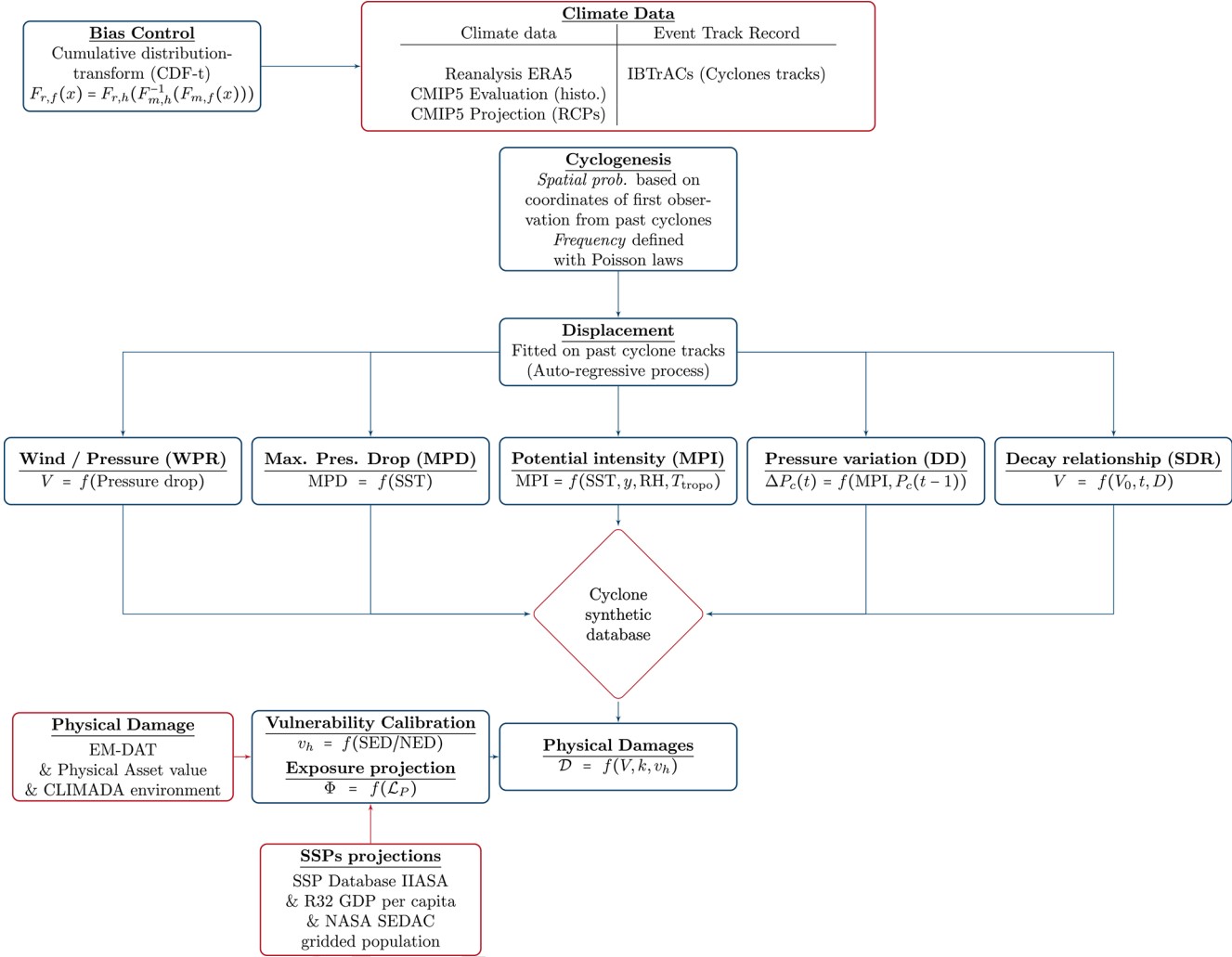

**Figure 1.** Cyclone generation Algorithm including a THERmodynamic module for Integrated National damage Assessment (CATHERINA) framework.

us to establish a national assessment of the damage generated by tropical cyclones under various assumptions concerning future adaptation to cyclone risk.

CATHERINA aims to provide country level estimates for future damage from tropical cyclones, consistent with climate model projections, with mainly economic and financial applications in mind. Examples of applications include estimating the impact of cyclones on the creditworthiness of government debt, providing a physical risk module for integrated assessment models, creating physical risk scenarios for stress testing the resiliency of the financial system at a country level and at a global level, etc. Given this aggregate country level analysis objective, our model is certainly a simplification compared to state-of-the-art cyclone dynamics models, does not aim for precise prediction of individual cyclone tracks and does not integrate a bottom-up description of damage to individual assets. At the same time, our paper improves earlier studies of cyclone damage on the aggregate

level. For example, compared to Mendelsohn et al. (2012), our model uses several climate scenarios with state-of-the-art bias correction as well as shared socioeconomic pathways (SSPs) to project population and regional domestic products and is based on precise physical asset value distribution.

The paper is organized as follows. Section 2 first describes the datasets used to fit the model on ERA5 and to generate synthetic cyclones based on both the ERA5 and CMIP5 model datasets (Sect. 2.1). Next, we present the statistical calibration process and the details of the thermodynamic instrumental variables (Sect. 2.2 to 2.4). Section 3 presents the dataset used to represent physical asset exposure and the SSP framework used to project their future value. Section 4 recalls the calibration methods implemented in the CLIMADA environment to fit the regional damage functions, defines the parameters of these functions using Eberenz et al. (2021) and applies them along the synthetic tracks to study the distribution of national annual damages. Section 5 explores the

properties of the synthetic tracks produced with the ERA5 and seven CMIP models, assesses climate uncertainty and introduces the bias-correction module in the context of changing climate conditions. To close this section, we present the global and regional projections of cyclone damage between 2070 and 2100 obtained with CATHERINA and illustrate briefly how these damage estimates may be used to quantify the future impact of tropical cyclones on sovereign default risk.

## 2 Modeling tropical cyclones

Our model structure follows Bloemendaal et al. (2020) with three main modeling steps: genesis, displacement of the eye and calibration of the cyclone properties. The STORM model relies on statistical relationships (James and Mason, 2005; DeMaria and Kaplan, 1994; Kaplan and DeMaria, 1995). This simulation method differs from the purely thermodynamic approach developed by Kerry Emmanuel (Emanuel, 1999; Emanuel et al., 2008).

The major change in our specification compared to Bloemendaal et al. (2020) is that we use the local definition of the available thermodynamic potential based on climate data. In particular, we use relative humidity (RH) and tropopause temperature (at 50 hPa, $T_{tropo}$) for better theoretical representation of the physics underlying the intensification process. In this section, we present the process of generating synthetic tracks, characterized by the maximum wind ($V_t$) and central pressure ($P_t^c$) at each time step, given the climate conditions extracted from climate models.

### 2.1 Input data

In the CATHERINA framework, we fit the properties of historical cyclones (IBTrACS database) on past climate reanalyses (ERA5) from the perspective of describing future cyclones based on global climate model outputs (CMIP) with a lower spatial and temporal resolution. This perspective constrains us to use monthly data.

#### 2.1.1 Climate model data (CMIP)

CATHERINA aims at generating cyclone tracks with properties drawn from climate models to enable national damage assessments, bridging the gap between AOGCM outputs and damage assessments. To reduce the bias in the variables produced by climate models and to evaluate the performance of CATHERINA on past data by comparing the simulated cyclone damages to the realized ones, we use historical simulations (as opposed to future climate projections) from the Coupled Model Inter-comparison Project (Phase 5) models (Taylor et al., 2012). We use the historical climate simulations at the monthly frequency for the RH at 2 m, SST, mean sea level pressure (MSLP) and tropopause temperature ($T_{tropo}$) (at a pressure level of 50 hPa) from the Coper-

**Table 1.** Climate data resolution.

|  | Resolution |
| --- | --- |
| ERA5 (reanalysis) | $0.25 \times 0.25$ |
| ACCESS1-0 (BoM-CSIRO, Australia) | $1.88 \times 0.93$ |
| CanESM2 (CCCMA, Canada) | $1.41 \times 0.94$ |
| GISS-E2-H (NASA, USA) | $2.5 \times 2$ |
| NorESM1-ME (NCC, Norway) | $1 \times 1$ |
| bcc-csm1-1-m (BCC, China) | $1 \times 0.74$ |
| IPSL-CM5A-MR (IPSL, France) | $1 \times 1$ |
| INMCM4 (INM, Russia) | $2 \times 1.5$ |

nicus climate data store.[1] CMIP5 data are used in the fifth assessment report of the Intergovernmental Panel on Climate Change (IPCC). The latest synthesis report in 2022 (IPCC AR6) uses CMIP6 datasets, but in the present paper, we use CMIP5 data because of the broader availability of climate variables.

We use models from the following climate centers: NASA and the Goddard Institute for Space Studies (GISS-E2-H, USA), Institut Pierre Simon Laplace (IPSL-CM5A-NR, France), Bureau of Meteorology – Commonwealth Scientific and Industrial Research Organisation (ACCESS1-0, BoM-CSIRO, Australia), Beijing Climate Center (bcc-csm1-1-m, China), Institute of Numerical Mathematics (INMCM4, Russia), Norwegian Climate Centre (NorESM1-ME, Norway), and Canadian Centre for Climate Modelling and Analysis (CanESM2, Canada). The spatial resolution goes from 0.75 to 2.5° depending on the model (see Table 1). Each climate model produces a potentially biased estimate of multiple climate variables at the spatial resolution given in Table 1 and on a monthly basis. The choice of the GCMs was driven by the availability of the variables of interest in the CDS in the representative concentration pathways used in the exercise (RCPs 2.6, 4.5 and 8.5 W m$^{-2}$) in both single level and multiple pressure level monthly data in the same ensemble (r1i1p1). We also aimed at having multiple regions represented.

To reduce the influence of model bias, we use a large number of models and consider the distribution of results provided by all the models. Then we correct, variable by variable and for each basin and each model, the biases with respect to the reanalysis along the same tracks.

#### 2.1.2 ERA5 reanalysis

Climate reanalyses describe the historical climate conditions, obtained by assimilating all available observations into the models. They provide numerical estimates of atmospheric parameters (e.g., air temperature, pressure and wind) at different altitudes/pressure levels and surface parameters (such as rainfall, soil moisture content, ocean-wave height and sea

---

[1]Climate data are available at the Copernicus Climate data store: https://cds.climate.copernicus.eu/ (last access: 14 October 2022).

surface temperature) on a single level. We use reanalyses to calibrate the cyclone generation algorithm based on the most realistic available estimates of climate variables.

We use the ERA5 reanalysis from the European Centre for Medium-Range Weather Forecasts (ECMWF) to fit the CATHERINA model[2] (Hersbach et al., 2020). This dataset covers the Earth on a 30 km grid ($\sim 0.25°$) and resolves the atmosphere using 137 levels from the surface up to a height of 80 km. In this paper, to ensure compatibility with CMIP5 models, we extract MSLP, SST, sea level RH and tropopause temperature ($T_{\mathrm{tropo}}$) at the monthly frequency. Because ERA5 better resolves past tropical cyclones than climate models, the historical mean sea level pressure values in ERA5 are influenced by their presence. Consequently, we retrieve the mean sea level pressure and environmental relative humidity 500 km ($\sim 5°$ longitude) away from the storm center to extract a value for $P_{\mathrm{env}}$, which is meant to represent the pressure – at a given latitude and season – in normal environmental conditions.[3]

### 2.1.3 Historical cyclone tracks (IBTrACS)

We use the IBTrACS database (Knapp et al., 2010).[4] This database provides information on past cyclone tracks at 3 h frequency. We remove the events classified as disturbance or extratropical and do not consider the South Atlantic basin (see Fig. A1 for more information). Climate reanalysis availability requires us to focus on post-1980 cyclones, which reduces the database to 4574 cyclones. In the context of an integrated damage assessment, to focus on the events that have a potentially substantial impact on assets, we select only tropical cyclones with maximum wind speed exceeding 35 m s$^{-1}$, obtaining 2966 on the full database and 1451 focusing on tropical cyclones between 1980 and April 2020. In Fig. 2, we plot the central pressure along each cyclone life. This graph suggests that the cyclone phases are fully represented in the

---

[2]Climate data are available at the Copernicus Climate data store: https://cds.climate.copernicus.eu/ (last access: 14 October 2022).

[3]We retrieve both MSLP and RH to define $P^{\mathrm{env}}$ and $RH^{\mathrm{env}}$ (Holland, 1997) away from the center in the reanalysis because tropical cyclone thermodynamic potential intensity – through thermodynamic efficiency and moist entropy (Eqs. 7 and 6) – arises from the deviations from the normal conditions. Monthly averaging may smooth values so that the data extracted along historical tracks may not represent the conditions at the time of cyclone passage. Therefore, using monthly means, this translation is mainly done for reasons of theoretical coherence. In future studies, this model will be applied with higher temporal resolution, and performing this translation would be more important. In the present version of our paper, because the CMIP5 projections of the sea level temperature were only available at monthly frequency in the CDS, we chose to perform the exercise using monthly data to illustrate our approach.

[4]See http://ibtracs.unca.edu/ (last access: 14 October 2022) for a browser of the data and https://www.ncdc.noaa.gov/ibtracs/index.php?name=ib-v4-access (last access: 14 October 2022) for the full dataset.

database, i.e., from the genesis to dissipation. The northern Indian basin has the lowest number of reported events with wind speed above 35 m s$^{-1}$ (50 compared to 291 for the eastern Pacific, 185 for the North Atlantic, 305 for southern India, 158 for the South Pacific and 515 for the western Pacific) with variable reporting quality, which explains the more erratic shapes of the central pressure. For example, the return to normal of some events does not seem to be completely reported for this basin, as indicated in Fig. 2. We extract the maximum wind speed, cyclone eye pressure and coordinate variations of the eye from this database.

### 2.2 Cyclone genesis

The scientific consensus is that climate change will induce a reduction in tropical cyclone frequency: "Existing modeling studies also consistently project decreases in the globally averaged frequency of tropical cyclones, by 6 to 34 %. Balanced against this, higher resolution modeling studies typically project substantial increases in the frequency of the most intense cyclones" (Knutson et al., 2020, p. 1). Although thermodynamic descriptions of cyclone genesis exist in the literature (Gray, 1975; DeMaria et al., 2001), we choose to rely on a simple statistical model based on past frequencies for the genesis.

The number of synthetic cyclones each year is determined by the Poisson distribution in each basin, with the intensity parameter defined as the average number of cyclones per year in the historical data. We use the parameters given in Bloemendaal et al. (2020), i.e., $\hat{\lambda}_{\mathrm{EP}} = 14.5$ for the eastern Pacific, $\hat{\lambda}_{\mathrm{NA}} = 10.8$ for the North Atlantic, $\hat{\lambda}_{\mathrm{NI}} = 2.0$ for northern India, $\hat{\lambda}_{\mathrm{SI}} = 12.3$ for southern India, $\hat{\lambda}_{\mathrm{SP}} = 9.3$ for the South Pacific and $\hat{\lambda}_{\mathrm{WP}} = 22.5$ for the western Pacific. The parameters would have been smaller if estimated using our filtered database of tropical cyclones with wind speeds above 35 m s$^{-1}$: $\hat{\lambda}_{\mathrm{EP}}^{35} = 7.31$, $\hat{\lambda}_{\mathrm{NA}}^{35} = 4.43$, $\hat{\lambda}_{\mathrm{NI}}^{35} = 1.6$, $\hat{\lambda}_{\mathrm{SI}}^{35} = 6.81$, $\hat{\lambda}_{\mathrm{SP}}^{35} = 4.00$ and $\hat{\lambda}_{\mathrm{WP}}^{35} = 11.86$. However, we maintain the original parameters used in STORM to take into consideration the fact that some events will be generated in conditions not favorable for the development of cyclones and cleared out of the database. The number of cyclones making landfall being critical for damage estimation (Hall and Jewson, 2007; Lee et al., 2018; Arthur, 2021), we ensured that the simulated landfall rates are in line with the observations over the historical period: Fig. 3 shows that the parameters of Bloemendaal et al. (2020) lead to approximately the same number of intense cyclones making landfall per year in each basin in the historical data, in our simulations based on reanalysis data and in simulations based on GCMs. We note however that the framework can be further improved by choosing the intensity parameter to match exactly the average historical number of cyclones making landfall.

Similarly, the temporal and spatial initial positions of synthetic future cyclones (longitude $x$, latitude $y$ and starting month $m$) are generated by independent sampling from the

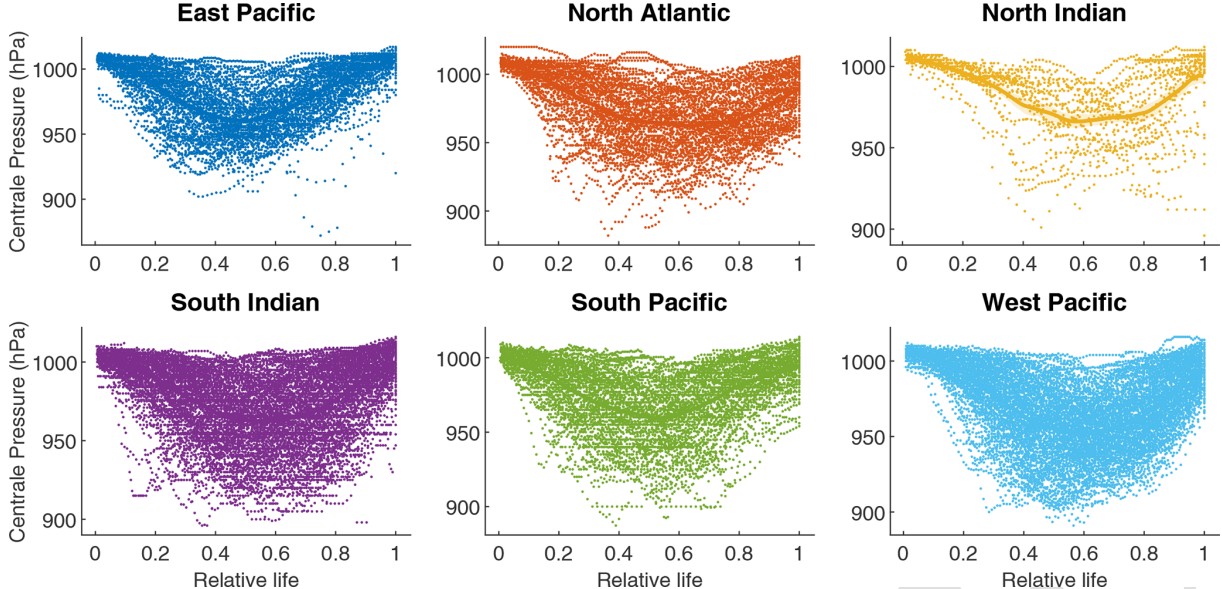

**Figure 2.** Normalized evolution of the central pressure (hPa) during a cyclone's life. The central pressure values are retrieved from IBTrACS. For the western Pacific we reduced the sample to events between 1995 and 2020 for visualization purposes.

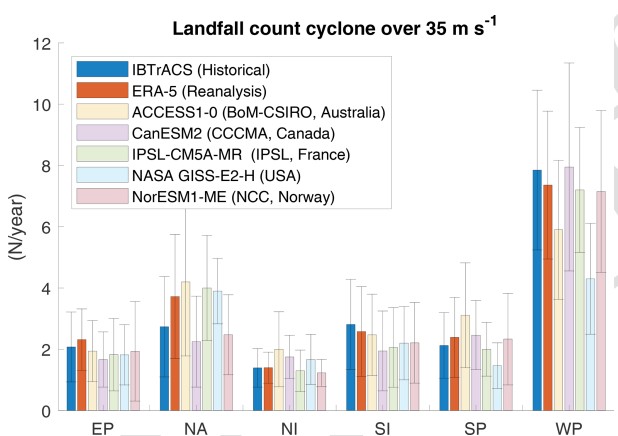

**Figure 3.** Number of tropical cyclones over $35\,\mathrm{m\,s^{-1}}$ making landfall simulated using Bloemendaal et al. (2020) parameters over the historical period (1980–2010).

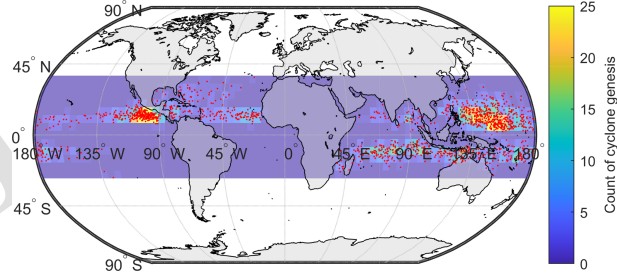

**Figure 4.** Spatial distribution of genesis points (tropical cyclones over $35\,\mathrm{m\,s^{-1}}$ in IBTrACS). The color scale corresponds to the count of cyclones per $5 \times 5°$ box (truncated to 25 for scaling reasons). The genesis location corresponds to the first reported point of each track.

historical distribution of these variables. Figure 4 shows the geographical distribution of cyclones retrieved from IB-TrACS (i.e., $x_{\mathrm{obs}}$, $y_{\mathrm{obs}}$), and the histograms in Fig. 5 show the monthly distribution ($m_{\mathrm{obs}}$) of cyclones in each basin.

## 2.3 Cyclone trajectories

A rich literature focuses on cyclone tracking algorithms; see, e.g., Neu et al. (2013) for a comprehensive review. Although more advanced definitions have been proposed (Hall and Jewson, 2007; Fabregat et al., 2016), we choose, in line with Bloemendaal et al. (2020), to implement a simple autoregressive model for cyclone coordinates. Following James

and Mason (2005), the time evolution of the latitude and longitude of the cyclone center is described by the following stochastic dynamics:

$$\Delta x_t = a_0 + a_1 \Delta x_{t-1} + \varepsilon_t^x, \qquad \varepsilon_t^x \sim \mathcal{N}(0, \sigma_x), \quad (1)$$

$$\Delta y_t = b_0 + b_1 \Delta y_{t-1} + \frac{b_2}{y_t} + \varepsilon_t^y, \qquad \varepsilon_t^y \sim \mathcal{N}(0, \sigma_y). \quad (2)$$

Here $x_t$ and $y_t$ are the latitude and longitude of the cyclone center sampled with a 3 h time step. $\Delta x_t = x_t - x_{t-1}$, $\Delta y_t = y_t - y_{t-1}$, $\varepsilon_t^x$ and $\varepsilon_t^y$ are i.i.d. (independent and identically distributed) noises independent of one another, and the constants $a_0$, $a_1$, $b_0$, $b_1$, $b_2$, $\sigma_x$ and $\sigma_y$ are fitted on IBTrACS data independently for each basin by least squares regression. The nonlinear term in the incremental variation of the latitude is justified by the tendency for cyclones to move away

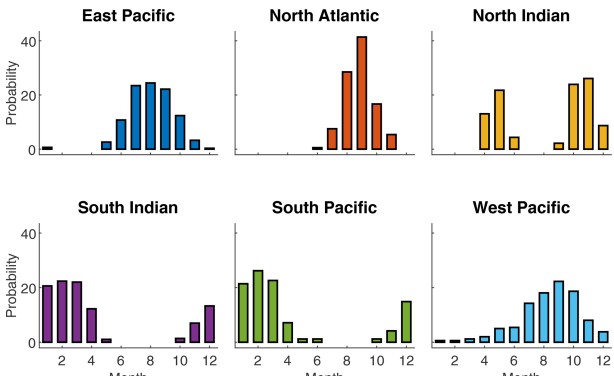

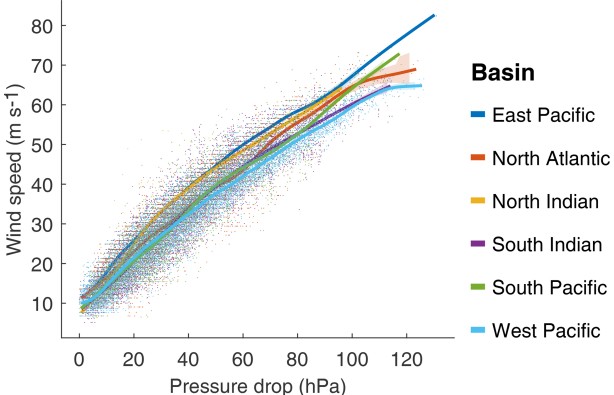

**Figure 5.** Relative monthly distribution (%) of cyclones in each basin defined by potentially damaging cyclones (over $35\,\mathrm{m\,s^{-1}}$) in IBTrACS. Each bar gives the probability of each cyclone being allocated to a given month.

**Figure 6.** Maximum wind ($\mathrm{m\,s^{-1}}$) and pressure drop (hPa) values from IBTrACS together with the fit of Eq. (3). The coefficients $a$ and $b$ fitted per basin are provided in Table A2.

from the Equator, especially at very low latitudes (James and Mason, 2005, p. 183).

To take into account the dependency of the cyclone displacement on the location of the eye, and following Bloemendaal et al. (2020), we fitted these relationships locally using an additional grouping by 5° longitude and latitude sections and months. Figure A2 illustrates the parameters $a_0$ and $a_1$, the adjusted $R^2$ and the number of observations averaged over months used to fit Eq. (1). We note that the trajectories in the northern Indian basin are less well captured in some areas, particularly longitudinal movements near the coast from Yemen, Oman, and the United Arab Emirates (see the maps of adjusted $R^2$ in Fig. A2).

This statistical definition of cyclone trajectories does not consider changes in track behavior. For example, observed trends in tropical cyclone translation speed (Kossin, 2018) and poleward migration of maximum intensity (Kossin et al., 2014) could be considered to improve the projections of tropical activity. This indeed has potential implications for tropical-cyclone-related risk in some areas where vulnerabilities are high and the present-day frequency of tropical cyclones is low (Bruyère et al., 2019).

## 2.4 Thermodynamic description of cyclone intensity

The intensity of cyclones in our model is defined through the following five steps described in detail in subsequent paragraphs. The wind–pressure relationship (WPR) links the central pressure to the maximum 10 min sustained wind speed. The local thermodynamic MPI is determined from local meteorological variables. The maximum pressure drop (MPD) is determined from historically observed pressures. The depression dynamics (DD) along tracks is defined using an autoregressive stochastic equation. When the cyclone arrives on land, a statistical decay relationship (SDR) dictates the evolution.

### 2.4.1 WPRs

We describe the cyclone intensity through its central pressure $P_t^c$, which is linked to the maximum wind through an empirical relationship. Let $V_t$ be the maximum 10 min sustained wind speed $(\mathrm{m\,s^{-1}})$[5] of the cyclone at time $t$. This maximum wind is observed around 50 km away from the storm center on average[6] and reported in the IBTrACS dataset for historically observed cyclones.

The WPR is calibrated separately for each basin and takes the following form:

$$V_t = a \left( P_{\mathrm{env}}(x_t, y_t, t) - P_t^c \right)^b, \tag{3}$$

where $P_{\mathrm{env}}(x, y, t)$ is the MSLP extracted 500 km away from the eye location at this time in ERA5 and $P_t^c$ is the central pressure extracted from IBTrACS. This relationship is illustrated in Fig. 6, and the parameters $a$ and $b$ are fitted on ERA5 and IBTrACS data using nonlinear least squares.

We acknowledge that most cyclone track data use WPRs to determine $P_c$. The conversion back to wind speed from the reported $P_c$ using a basin-specific WPR still introduces errors, as different WPRs are used to operationally estimate $P_c$ within each basin (Harper, 2002; Courtney, 2009; Courtney and Burton, 2018; Courtney et al., 2021). However, given the similarity of the relationships, we find that basin level estimations are a sufficient proxy in the context of this illustration of the CATHERINA framework.

---

[5]For the data from the World Meteorological Organisation (WMO) and the agencies reporting 1 or 3 min sustained wind speed, we performed the conversion to 10 min sustained wind speed using the coefficients suggested by Knapp et al. (2010). See Fig. A1 in the Appendix for more details about the agencies and reporting bias.

[6]Radii of maximum wind are also reported in IBTrACS, but this information is not central to national level assessment.

### 2.4.2 Local thermodynamic MPI

We compute the local thermodynamic MPI following the thermodynamic relationships defined in Holland (1997). This is particularly relevant in the context of a national damage assessment under a changing climate. Greenhouse gas emissions not only warm up the oceans, but also cool down the lower stratosphere (Butchart et al., 2000; Forster et al., 2007; Ramaswamy et al., 2006). Thus, the tropopause temperature – that is, the temperature corresponding to a pressure of 50 hPa or to an altitude of approximately 20 km available in a multi-level CMIP dataset ($T_{\text{tropo}}$) – must be included in the modeling of the intensification process. Indeed, the thermodynamic efficiency factor $\mathcal{E}_t$ proportional to the difference between the tropopause and sea surface temperatures plays an essential role in the determination of the central pressure of tropical cyclones. The relative humidity (RH) (which changes with climate change; see Sherwood et al., 2010) is also an influential parameter allowing for a better description of thermodynamic potential enabling cyclone intensification. Adding these two climate variables enables the CATHERINA model to better take into account the additional energy potential due to the widening of temperature differences between the sea surface and upper troposphere and variation in moist entropy.

Following the seminal formulation in Emanuel (1988) and integrating additional simplifications proposed in the subsequent paper (Emanuel, 1991) leads to the following framework summarized in Holland (1997).

$$\text{MPI}_t = \text{MSLP}(x_t, y_t, t) \cdot \exp^{-X_t}, \tag{4}$$

$$X_t = \frac{\mathcal{E}_t \cdot \text{SST}(x_t, y_t, t) \cdot \Delta S_t^{\text{m}} - \frac{f(y_t)^2 r_{\text{env}}^2}{4}}{R_{\text{d}} \cdot \text{SST}(x_t, y_t, t)}, \tag{5}$$

$$\mathcal{E}_t = \frac{\text{SST}(x_t, y_t, t) - T_{\text{tropo}}(x_t, y_t, t)}{\text{SST}(x_t, y_t, t)}, \tag{6}$$

$$\Delta S_t^{\text{m}} = R_{\text{d}} \ln\left(\frac{\text{MSLP}(x_t, y_t, t)}{P_{t-1}^{\text{c}}}\right) + \frac{L_\upsilon(q_{\text{c}\,t}^\star - q_t^{\text{env}})}{\text{SST}(x_t, y_t, t)}, \tag{7}$$

with moist entropy potential defined along track using specific humidity in the eye vs. at environmental conditions:

$$q_{\text{c}\,t}^\star = \frac{3.08}{P_{t-1}^{\text{c}}} \exp\left(\frac{(\text{SST}(x_t, y_t, t) - 273.15)}{\text{SST}(x_t, y_t, t) - 29.65}\right), \tag{8}$$

$$q_t^{\text{env}} = \frac{3.08 \cdot \text{RH}(x_t, y_t, t)}{\text{MSLP}(x_t, y_t, t)}$$
$$\times \exp\left(\frac{17.67\,(\text{SST}(x_t, y_t, t) - 273.15)}{\text{SST}(x_t, y_t, t) - 29.65}\right). \tag{9}$$

where $(x_t, y_t, t)$ are the coordinates of the eye defined in Eqs. (1) and (2), $\text{SST}(x_t, y_t, t)$ and $T_{\text{tropo}}(x_t, y_t, t)$ are, respectively, the sea surface and tropopause temperatures, $R_{\text{d}} = 287.058\,\text{J kg}^{-1}\,\text{K}^{-1}$ is the specific gas constant for dry air, $\text{MSLP}(x_t, y_t, t)$ is the mean local sea level pressure,

$\text{RH}(x_t, y_t, t)$ is the near-surface relative humidity at 2 m extracted from the monthly dataset of ERA5 climate reanalysis or CMIP climate models. $f(y_t) = 2\omega \sin(y_t)$ is a Coriolis parameter depending on the latitude, $r_{\text{env}}$ is the distance between the eye and the area under regular conditions (fixed at 500 km), $q_{\text{env}}$ and $q_{\text{c}}^*$, respectively, are the specific humidity at environmental conditions and at saturation, i.e., for RH = 100 %, in the eye. $\Delta S^{\text{m}}$ is the difference of moist entropy between the environment and the storm center and $L_\upsilon$ is the latent heat of vaporization. The distributions of the climate variables and the instrumental indicators computed from ERA5 climate variables along IBTrACS involved in this step are shown in Fig. 7.

### 2.4.3 MPD

Several papers, including Bloemendaal et al. (2020), link the sea surface temperature directly to the pressure drop, or equivalently the wind speed, via a statistical relationship (DeMaria and Kaplan, 1994). Merrill (1987) suggests that this predictor alone does not provide a good indication of whether a given storm will intensify. However, in line with Emanuel (1988) and DeMaria and Kaplan (1994), the sea surface temperature can be used to fix a limit for the pressure drop. Thus, in CATHERINA, to prevent the pressure drop from diverging in the projection, we cap it by the maximum observed pressure drop for the corresponding sea surface temperature:

$$P_t^{\text{c}} := \max(P_t^{\text{c}}, \text{MSLP}(x_t, y_t, t) - \text{MPD}(\text{SST}(x_t, y_t, t))),$$

where the maximum pressure drop function is given by the following equation:

$$\text{MPD}(\text{SST}) = A + B \cdot e^{C(\text{SST}(x_t, y_t, t) - T_0)}, \tag{10}$$

with $T_0 = 30\,°\text{C}$. To fit this functional relationship, we first retrieve, for each basin, and for each value of the sea surface temperature SST, rounded to 0.1, the maximum observed value of the pressure drop in the basin. These values are shown as crosses in Fig. 8. The coefficients $A$, $B$ and $C$ from the relationship (10) are then fitted to these values by nonlinear least squares. The resulting MPD functions for each basin are shown in Fig. 8 as solid line.

The definition of the MPD is identical in STORM and CATHERINA, but the role of this quantity differs in the two models. Figure 8b gives a misleading idea about the strength of correlation between the sea surface temperature and the central pressure: indeed, fitting the distribution on the full sample (instead of just the maximum pressure drop for each temperature value) shows a much weaker influence of sea surface temperature alone (see Fig. 8a), even on a weekly basis (see Jien et al., 2017). However, this instrumental variable is essential for preventing CATHERINA from producing unrealistically low central pressure. On the other hand, limiting the maximum pressure drop in simulations to the parametric function given by Eq. (10) fitted by nonlinear least squares to

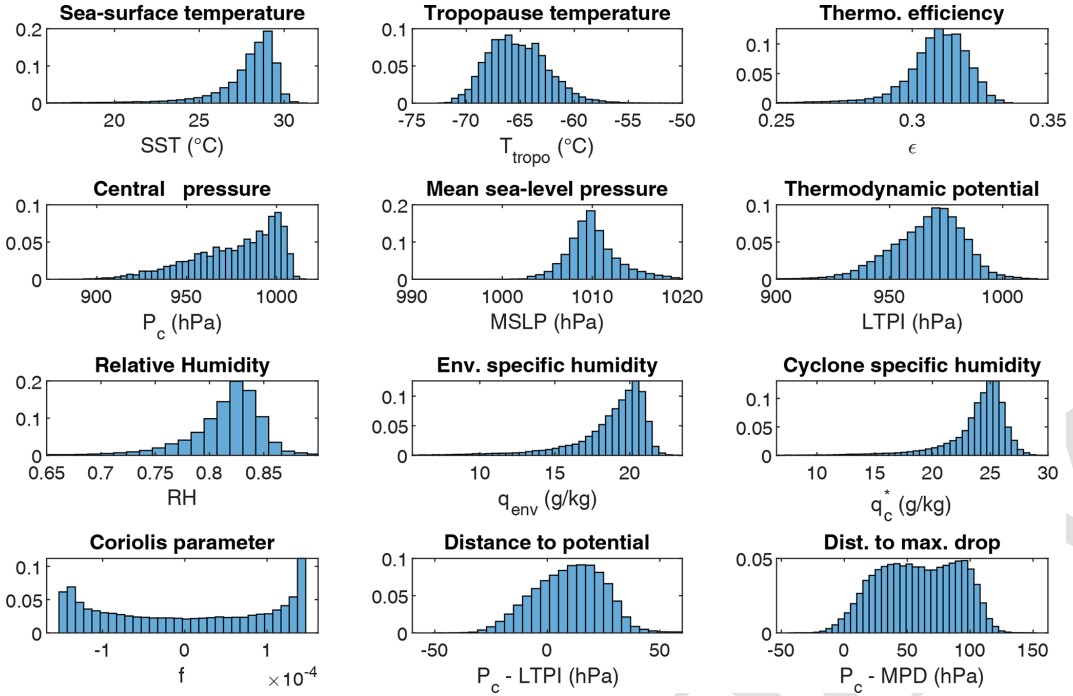

**Figure 7.** Empirical distribution of physical variables and CATHERINA instrumental variables computed from ERA5 along tracks in IB-TrACS.

the observed MPD leads to an excessively strong limitation on the intensity of the simulated cyclones, since many points in Fig. 8 (lower graph) are above the red curve. Therefore, we shift this parametric function upward to the highest observed point to relax this limitation.

### 2.4.4 Depression dynamics

The evolution of the central pressure depending on the intensification factor $Y_t$ (which is defined differently in our approach and in STORM, as we explain below) is described by the following autoregressive stochastic equation (James and Mason, 2005):

$$\Delta P_t^c = c_0 + c_1 \Delta P_{t-1}^c + c_2 e^{-c_3[P_t^c - Y_t]} + \varepsilon_t^P, \tag{11}$$

$$\varepsilon_t^P \sim \mathcal{N}(0, \sigma_{P^c}). \tag{12}$$

This relationship channels the effect of global warming, affecting the thermodynamic potential constructed with climate projections, on the cyclone intensity. Thus, the incremental variation of the central depression of the cyclone is linked to the difference between the central pressure at time $t$ and the potential available in the environment.

The intensification module of CATHERINA is inspired by STORM (Bloemendaal et al., 2020). The main differences are the definition of the thermodynamic MPI used in Eq. (11) and the role played by the MPD. In Bloemendaal et al. (2020), the MPI is defined by subtracting the MPD from the normal environmental pressure (MSLP), where the

MPD is defined as a function of the SST and Bister and Emanuel (2002) values are used to bound their values. On the other hand, we define the thermodynamic intensification factor following Holland (1997) with variables extracted along the synthetic tracks and use the SST–MPD relationship as a capping function (see Fig. 8). Table A3 summarizes the main differences of the two approaches.

As illustrated in Fig. 9, both methods produce a similar dependence of the intensification function (11) on the distance to the potential and maximum pressure drop. When $P_c$ approaches the local maximum potential intensity (or maximum pressure drop), $\Delta P^c$ is more likely to be positive and to decrease the storm intensity. In other words, we can distinguish two phases (see Fig. 9, left graph): the intensification phase (in blue), when the central pressure is above the local MPI threshold, and a decay phase (in red), when the central pressure is below this local MPI.

In contrast to the MPD, the MPI does not represent the maximum achievable pressure and can be exceeded when accounting for additional external factors not reflected in climate data. For example, Chen et al. (2021) suggest that rapid intensification also depends on dynamical factors (e.g., upper divergence and wind shear). While the James and Mason (2005) formulation implicitly assumes that these factors are accounted for in the residual term of Eq. (12), it does not consider that the distance to the maximum potential thus defined can take negative values. Indeed, this specification originally suggests using the maximum achievable central pressure, i.e.,

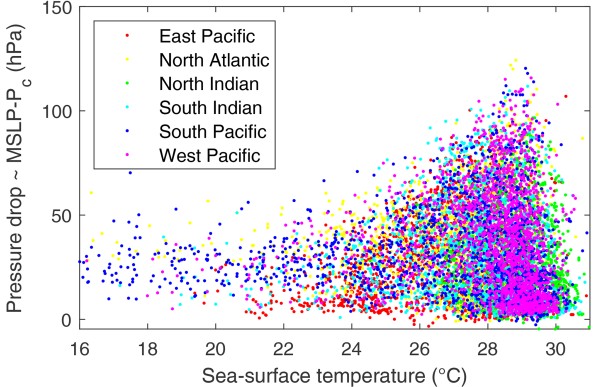

**(a) Sea-surface temperature and corresponding pressure drop**

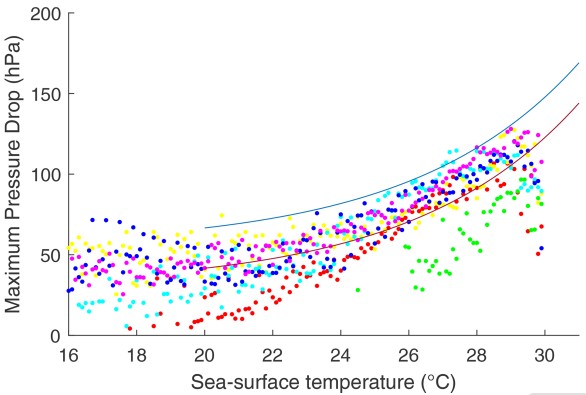

**(b) Sea-surface temperature and maximum pressure drop**

**Figure 8. (a)** Full distribution of observed pressure drop (hPa) values for a given SST (°C). **(b)** Maximum observed pressure drop values for a given SST. The red line shows the least squares fit of Eq. (10), which corresponds to parameter values $A = 30.6$, $B = 86.3$ and $C = 0.19$. The blue curve is the capping function used to prevent unrealistic pressure drops, obtained by shifting the red curve upwards.

that $P_c - \text{MPI} > 0$. As a result, the two intensification factors are not defined in the same domain (see Fig. 7). Using the local thermodynamic MPI, negative values, corresponding to a situation where the central pressure is below the MPI ($P_c < \text{MPI}$), are associated with a positive response of the pressure dynamic module and a decrease in storm intensity. Using the MPD, the response is likely to become positive when the distance to the MPD is below 40 to 60 hPa, depending on whether the function has been applied to the local sea surface temperature or the maximum per grid box ($5° \times 5° \times$ month). Given the similar response provided, using the local MPI instead of MPD as the intensification factor offers a better theoretical representation of the conditions affecting cyclone intensification in the cyclone dynamic module. The central pressure dynamics used for the fitting and dynamics of the synthetic tracks produced in the North Atlantic basin are illustrated in Fig. 10. The intensification of synthetic cyclones is in line with historical observations.

### 2.4.5 SDR over land

We model the evolution of the cyclone after landfall using an exponential decay function considering that tropical cyclone intensity decreases as a function of the time and distance the tropical cyclone has covered whilst being over land (Kaplan and DeMaria, 1995). Similarly to Bloemendaal et al. (2020), after three steps on land we suppose that the wind at time $t_L$ follows:

$$V_{t_L} = V_b + (R \cdot V_0 - V_b)e^{-\alpha t_L} - f_1(t_L)\left(\ln\frac{D^l}{D_0}\right) + f_2(t_L) \quad (13)$$
$$= V(t_L, D^l, V_0),$$

where $D^l$ is the distance to coast computed using natural Earth coastlines (available at https://www.naturalearthdata.com/downloads/10m-physical-vectors/, last access: 14 October 2022), $V_0$ is the wind at landfall and $t_L$ the time spent on land by the eye. This function was fitted on IB-TrACS using nonlinear least squares. In our procedure, we use the global parameters: $R = 0.79$, $V_b = 15\,\text{m s}^{-1}$, $\alpha = 0.044\,\text{h}^{-1}$, and $f_1(t_L) = \tilde{c}_1 t_L(t_{0,L} - t_L)$, $\tilde{c}_1 = 3.35 \times 10^{-4}\,\text{m s}^{-1}\,\text{h}^{-2}$, $t_{0,L} = 172\,\text{h}$, $f_2(t_L) = d_1 t_L(t_{0,L} - t_L)$, $d_1 = -0.00186\,\text{m s}^{-1}\,\text{h}^{-2}$ and $D_0 = 1\,\text{km}$. Kaplan and DeMaria (1995) introduced this function to model the decay of tropical cyclones over land in a simple way and showed that it provides an acceptable approximation for $t_L \geqslant 12\,\text{h}$. As each time step is 3 h, we let the TC intensity be driven by Eq. (11) and the first three steps and apply the decay function after three steps, that is, for $t_L \geqslant 12\,\text{h}$. A more sophisticated description could integrate for instance, cyclone physics, kinetic energy, and non-meteorological parameters such as ground topology. The SDR puts a strong constraint on the cyclone evolution after three steps. However, in the context of national damage assessment, we reiterate that reported damage costs are the combination of a series of various impacts including storm surge and not only extreme wind and that the most exposed area is at landfall. We consider therefore that the hypothesis of a rapid decay is acceptable and in line with observations.

### 2.5 Cyclone generation algorithm

The full cyclone generation procedure is presented in Algorithm 1. The cyclone wind speed is initiated at $20\,\text{m s}^{-1}$, and the initial pressure is determined from the WPR (Eq. 3). While the cyclone is over the sea, the pressure evolution $\Delta P_c$ is entirely determined from Eq. (11) based on the local thermodynamic potential. To prevent the model from producing an unrealistically low central pressure, we cap the MPD using Eq. (10). With this truncation, the lower bound for the pressure is given by the observed low-pressure values in similar sea surface temperature conditions. While the cyclone is over the sea, the wind is defined with the WPR (Eq. 3). When the cyclone arrives on land, the MPI is computed from the last known climate variables for the first three steps and

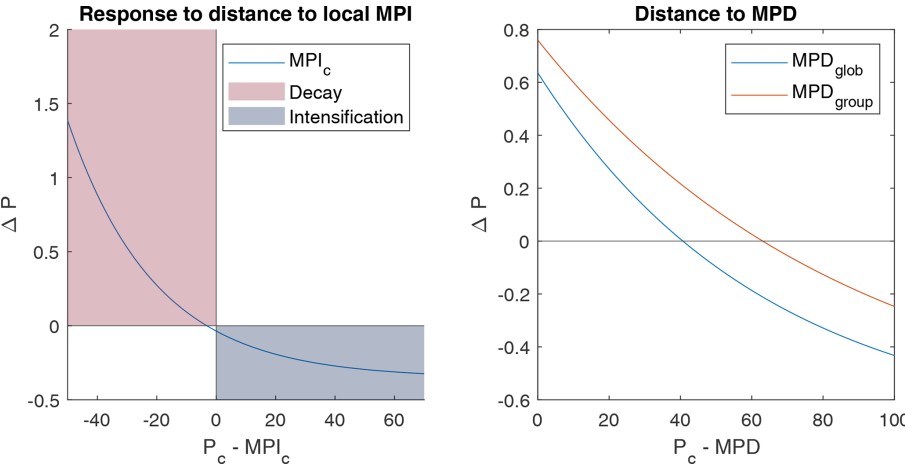

**Figure 9.** Response of the depression dynamics model to distance to thermodynamic potentials over intensification factor domains (hPa).

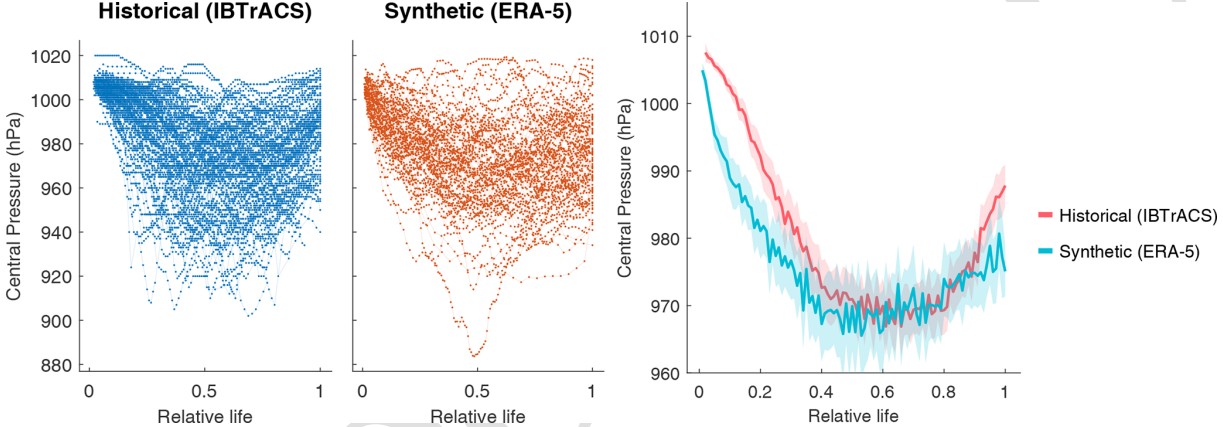

**Figure 10.** Individual historical vs. synthetic depression dynamics in the North Atlantic basin, and the confidence interval.

the pressure still follows the relationship (11). After three steps (9 h) on land, we start applying the decay relationship (Eq. 13) to define the wind. The variations of longitude and latitude are always defined using Eqs. (1) and (2). We force cyclones to remain in their genesis basins in this exercise. For example, running the algorithm on IPSL climate projections between 2075 and 2100 in RCP85 produces the output plotted in Fig. 11.

The cyclone intensification process used is inspired by the STORM model from Bloemendaal et al. (2020), which includes a single climate variable, and extended following Holland (1997) and Emanuel (1988) to encompass two more variables. We found that this extension provides statistically significant instrumental variables in the description of tropical cyclone intensification, which is the aim of the algorithm. Consequently, even if some thermodynamic processes have been simplified in our approach, it is still a step forward with respect to the existing state of the art in the field of integrated assessment modeling for climate impact analysis. Indeed, our approach is easy to implement, more sophisticated in terms

of processes included than most existing IAMs, has low bias due to our state-of-the-art bias-correction module described below, and can integrate any CMIP simulation with a limited set of available variables (only a few vertical levels, some only available at a monthly timescale, with some variables not always available).

## 3 Exposure in the shared socioeconomic pathways

### 3.1 Physical asset exposure

Eberenz et al. (2020a) present a methodology to downscale physical asset values on a high-resolution grid using a combination of nightlight intensity, population data, and global country indicators and make their dataset fully available (Eberenz et al., 2020b). These estimates of physical asset values are based on the light intensity $L_i$ – from nighttime lights of the Black Marble 2016 annual composite of the VIIRS day–night band (DNB) (Román et al., 2018) in 2016 at the 15 arcsec resolution – and the population $P_{pix}$ per pixel –

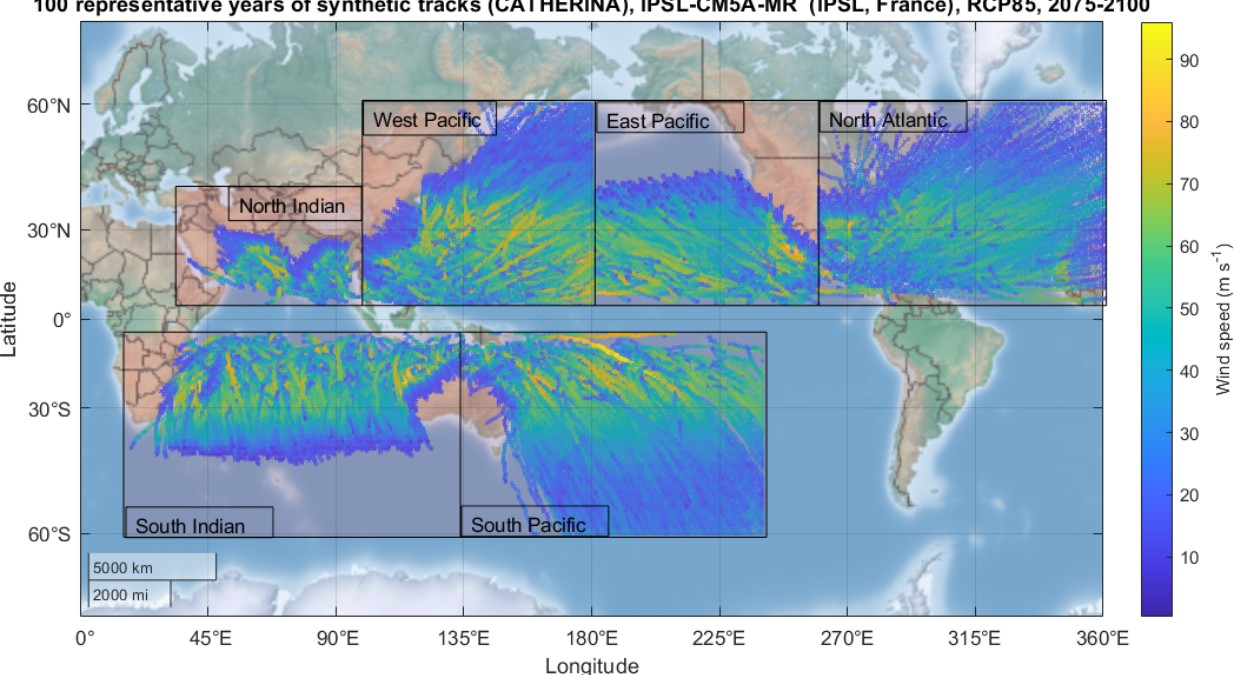

**Figure 11.** Example of 100 representative years of synthetic tracks produced with CATHERINA on IPSL-CM5A-MR raw climate data in RCP85 between 2075 and 2100 (i.e., four runs over the 25-year period).

from the Gridded Population of the World (GPW) database (Center for International Earth Science Information Network (CIESIN), 2017) in 2015 at the 30 arcsec resolution for 224 countries and various additional sources[7] allowing us to define the total asset ($A_{\text{tot}}$) for each country. This value is distributed to each grid cell proportionally to the light intensity $L_i$ times the local population $P_{\text{pix}}$:[8]

$$A_{\text{pix}} = A_{\text{tot}} \frac{L_i \cdot P_{\text{pix}}}{\sum_{\text{pix}_i}^{N} \left( L_i \cdot P_{\text{pix}_i} \right)}. \tag{14}$$

The physical asset value is expressed in USD as of 2014. Using this dataset in the future requires correction (either simulated or reported damages) for inflation using, for instance, the consumer price index (available at https://fred.stlouisfed.org, last access: 14 October 2022).

This method for allocation of national assets has limitations. For example, the distribution of assets near the coast,

industrial production sites or agricultural facilities may not be well represented. However, with this approach, asset values are defined on a uniform grid across countries and can be projected by multiplication by appropriate dynamic factors.

## 3.2   The SSP framework

Future exposure is sensitive to the scenarios of population growth and economic development. To take this into account, we use the framework of the SSPs introduced in O'Neill et al. (2014). These narratives are used in the IPCC development scenarios and provide a reference framework for risk assessment. A growing segment of the literature is dedicated to measuring the feasibility, costs and implications of achieving these scenarios (Riahi et al., 2017, 2021), and multiple IAMs were launched on assumptions based on these narratives (Riahi et al., 2017; Rogelj et al., 2018; Gidden et al., 2019).[9] Figure 12 displays the projections of the main features used by CATHERINA, the global domestic production (GDP) and population in the five SSPs at the global level by the IIASA GDP model.[10] CATHERINA uses these two indicators at the regional level (32 regions are available) to compute future local exposure (see Sect. 3.3).

---

[7]Produced capital, comprehensive global estimate of produced capital stock, i.e., the value of produced or manufactured assets per country (World Bank, 2018) – 2014/140 countries; the gross domestic product (GDP)-to-wealth ratio from the Global Wealth Report (Credit Suisse, 2017) – 2017/84 countries; the GDP per country from the World Bank Open Data portal (World Bank, 2019) – 2014/224 countries; subnational equivalent of GDP (GRP) from varying sources – 2012–2017/504 regions in 14 countries.

[8]The values of $A_{\text{pix}}$ on a 30 arcsec grid are available at https://www.research-collection.ethz.ch/handle/20.500.11850/331316 (last access: 14 October 2022).

[9]For example, for the variable of interest, the outputs of the IIASA GDP, IIASA-WiC POP, NCAR, PIK GDP-32, and OECD Env-Growth models are available.

[10]Variables relative to SSPs are available here: https://tntcat.iiasa.ac.at/SspDb/ (last access: 14 October 2022).

**Algorithm 1** Cyclone generation algorithm.

---

$V \leftarrow 20\,\mathrm{m\,s^{-1}}$

$P_c(s=0) \leftarrow \mathrm{MSLP} - \left(\dfrac{V}{a}\right)^{1/b}$

**if** $\mathrm{MSLP} - P_c > 0$ & $V > v_m$ & $s_1 < 4$ **then**

  while the pressure is below normal, wind is above threshold
  and we are not on land, do

$x(s) \leftarrow x(s-1) + \Delta x(s)$

  where  $\Delta x(s) \propto$ Eq. (1)

$y(s) \leftarrow y(s-1) + \Delta y(s)$

  where  $\Delta y_t \propto$ Eq. (2)

$\mathrm{MPI}(s) \leftarrow f_{\mathrm{MPI}}(y(s), P_c(s-1), \mathrm{SST}(s),$
$\qquad\qquad T_{\mathrm{tropo}}(s), \mathrm{MSLP}(s), \mathrm{RH}(s))$
$\qquad\qquad f_{\mathrm{MPI}} \propto$ Eq. (4)

$P_c(s) \leftarrow \max(P_c(s) + \Delta P_c(s), \mathrm{MSLP}(s) - \mathrm{MPD}(s))$
$\qquad\qquad \mathrm{MPD} \propto$ Eq. (10) &
$\qquad\qquad \Delta P_c(s) \propto$ Eq. (11)

$V(s) \leftarrow a(\mathrm{MSLP} - P_c(s))^b$

  **if** on land = TRUE **then**
    $s_1 \leftarrow s_1 + 1$
  **end if**
**else**
  **if** $s_1 > 4$ **then**
    Same functional for $x$ and $y$ but, compute distance to land
    $D(s)$ from natural earth coastlines and do

$V(s) \leftarrow V_b + (R \cdot V_0 - V_b)e^{-\alpha s_1}$
$\qquad - f_1(t_L)\left(\ln\dfrac{D}{D_0}\right) + f_2(t_L)$
$\qquad\quad V(s) \propto$ Eq. (13)

    $s_1 \leftarrow s_1 + 1$
  **end if**
**end if**

Note: this algorithm assumes step-wise extraction of climate
data in the Monte Carlo process. Another way, closer to the
framework suggested in Bloemendaal et al. (2020), would be
to (i) compute the tracks without properties, (ii) retrieve all cli-
mate variables and (iii) determine the properties using the ex-
tracted climate conditions in the last step.

---

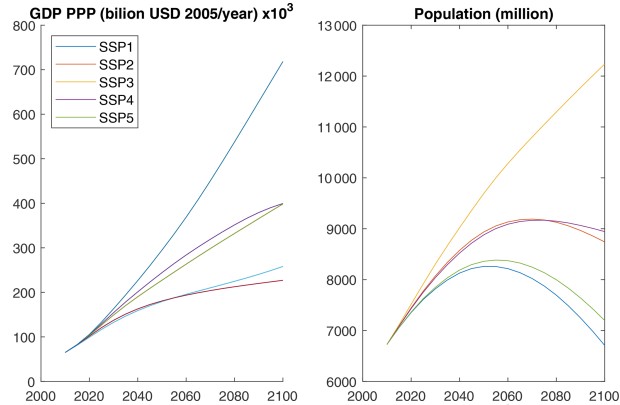

**Figure 12.** SSP GDP and population variation until 2100, at the
global scale, by the IIASA GDP model. TS1

ulation (see Fig. 12). SSP4 corresponds to a scenario with the
highest inequality, and SSP5 is the most likely to lead to the
higher concentration pathways (RCP8.5), with extensive use
of fossil fuel reserves but higher economic development and
global market integration.

Based on these storylines, it is clear that the physical ex-
posure to tropical cyclones will be driven by different fac-
tors in different SSP scenarios. Scenarios with steady growth
of GDP are generally associated with an increase in urban-
ization but a decrease in the global population by 2050. In
general, all narratives, except SSP3, present relatively sim-
ilar population dynamics at the global level. On the other
hand, the scenario with the lowest economic growth (SSP3)
assumes a sustained increase in the global population (cf.
Fig. 12), in particular in rural areas. We can therefore ex-
pect, in the former case, the physical asset value exposure to
be driven by the increase in regional wealth and mainly by
the growth of the exposed population in the latter.

In this paper, we specify economic and climate parame-
ters independently, while the literature generally associates
specific SSPs and RCPs, in particular in the CMIP6 exer-
cise. Indeed, integrated assessment modeling demonstrates
that specific temperature targets can only be reached under
certain socioeconomic conditions. The socioeconomic and
representative concentration pathways are therefore intrinsi-
cally linked at the global scale. For example, CMIP6 refers
to the following scenarios: SSP1–1.9, SSP1–2.6, SSP2–4.5,
SSP3–7.0 and SSP5–8.5. On the other hand, this pairing is
not straightforward at the regional level. In particular, the
SSP2 scenario is associated with regional heterogeneity in
socioeconomic pathways. As a result, although we can ex-
pect a convergence in the long run, we considered it rele-
vant to use economic development scenarios independently
of RCP in this exercise. When looking at the aggregated
level, however, we consider only scenarios that are feasible
(Rogelj et al., 2018). For instance, the RCP85 is achieved
only in the conditions of the SSP5.

Let us recall the main assumptions underlying these nar-
ratives. The "middle road" pathway (SSP2) is used as the
reference in most scenario analyses. It is a plausible base-
line in terms of economic and social resiliency, in which the
urbanization level is relatively high and GDP and popula-
tion are constantly increasing following the observed histor-
ical trend. On the other hand, the "rocky road" or "national-
rivalry" pathway (SSP3) presents totally different properties:
relative stagnation of GDP with a strong increase in the pop-

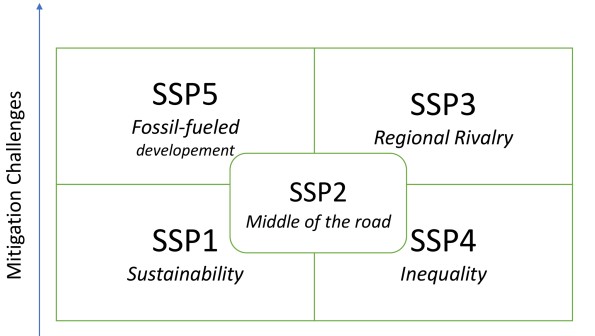

**Figure 13.** SSP matrix O'Neill et al. (2017) (based on O'Neill et al., 2014, Fig. 1, inspired by Kriegler et al., 2012, Fig. 3).

Lastly, we acknowledge another limitation of our approach: the vulnerability parameter (represented by the damage function parameter $v_h$ in our framework) does not depend on the SSP, while we could expect a reduction in the vulnerability parameter in the scenarios where the adaptation challenges are limited, such as SSP5 (cf. Fig. 13). This question is left for further research.

### 3.3 Dynamic projection of local exposure in SSPs

To estimate future exposures along the cyclone tracks in each scenario, we use the downscaled estimation for the exposed wealth (Eberenz et al., 2020b) and the coefficients representing the change between the current state and the future scenario in the framework of the shared socioeconomic pathways (O'Neill et al., 2014, 2017; Jones and O'Neill, 2020). The local physical exposure at the coordinates $(x, y)$ at time $t$ in a region $j$ in scenario $k$ is defined as follows:

$$\Phi(x, y, j, k, t) = \underbrace{(F_{\text{GDP}}^{\text{cap}}(j, k, t))^{\alpha_1}}_{\text{Global macro factor}} \cdot \underbrace{(F_{\text{pop}}(x, y, k, t))^{\alpha_2} \cdot \mathcal{L}_P(x, y)}_{\text{Local factor}}, \quad (15)$$

where $\mathcal{L}_P(x, y)$ is the local population density from Eberenz et al. (2020b), and the factor $F_{\text{GDP}}^{\text{cap}}$ is the projected GDP per capita growth for each region:

$$F_{\text{GDP}}^{\text{cap}}(j, k, t) = \frac{\text{GDP}(j, k, t)/\text{GDP}(j, t = 2020)}{P(j, k, t)/P(j, t = 2020)}, \quad (16)$$

where $P$ is the total population of the region retrieved from the SSP database (Riahi et al., 2017) (https://tntcat.iiasa.ac.at/SspDb/, last access: 14 October 2022) and $F_{\text{pop}}$ is the population exposure growth factor:

$$F_{\text{pop}}(x, y, k, t) = \frac{p(x, y, k, t)}{p(x, y, t = 2020)}, \quad (17)$$

where $p(x, y, k, t)$ represents the local projections of population (Jones and O'Neill, 2020) illustrated in Fig. 15. Figure 14 displays the scenario-based projections of GDP and

population in the five SSPs, at the regional level by the IIASA model.

We introduce the exponents $\alpha_1$ and $\alpha_2$ to disentangle the effects of increased cyclone intensity, GDP growth and population growth on the future damages as well as to account in a simple manner for possible future adaptation to tropical cyclone risk. Indeed, taking $\alpha_1 = \alpha_2 = 1$ amounts to assuming that damages from cyclones of similar intensities grow proportionally to the local GDP, whereas newly accumulated wealth can be more resilient than the existing one and protected by additional adaptation measures. Indeed, Bakkensen and Mendelsohn (2019) test for evidence of adaptation in past cyclone damage data and find that $\alpha_1$ is statistically different from 1 in all countries except the USA. At the global level, they estimate $\alpha_1$ to be equal to 0.364, with a standard error of 0.175, which is significant at the 95 % confidence level.

## 4 Damage assessment at the national level

### 4.1 Damage modeling

The percentage of asset values destroyed by a tropical cyclone depends on multiple parameters. For example, to assess the vulnerability of specific infrastructures to tropical cyclones, precise descriptions of building vulnerability are provided in the Federal Emergency Management Agency (FEMA) reports. Unanwa et al. (2000) propose a series of wind-damage bands depending on building types. They show that the sensitivity is higher for commercial and institutional buildings than for residential and mid-rise buildings and that generalized damages occur above 43–60 m s$^{-1}$, while a sustained wind regime above 73 m s$^{-1}$ could lead to the destruction of the entire superstructure of most buildings (except for mid- and low-rise ones). These bottom-up approaches allow us to set the limits of the potential damage functions, but their use requires a complete inventory of assets and up-to-date values of numerous parameters (age, height, materials, etc.). Therefore, CATHERINA relies on regional damage functions calibrated by Eberenz et al. (2021) on wind speed along the cyclone track (IBTrACS) and reported damages in Guha-Sapir et al. (2018).

Damages provoked by tropical cyclones can be related to several sub-perils. While 40 % of cyclone damages are directly wind-related, another 40 % can generally be attributed to storm surge, and the rest of the damage is generated by heavy precipitation. However, CATHERINA does not distinguish these sub-perils but uses a statistical relationship to estimate the regional damage induced by a cyclone from a proxy variable given by the maximum wind speed. Indeed, the wind speed is the proxy used in the Saffir–Simpson hurricane wind scale to define the intensity of a cyclone. The damage function is fitted on multiple events from the total damage reported in the global disaster database EM-DAT (Guha-

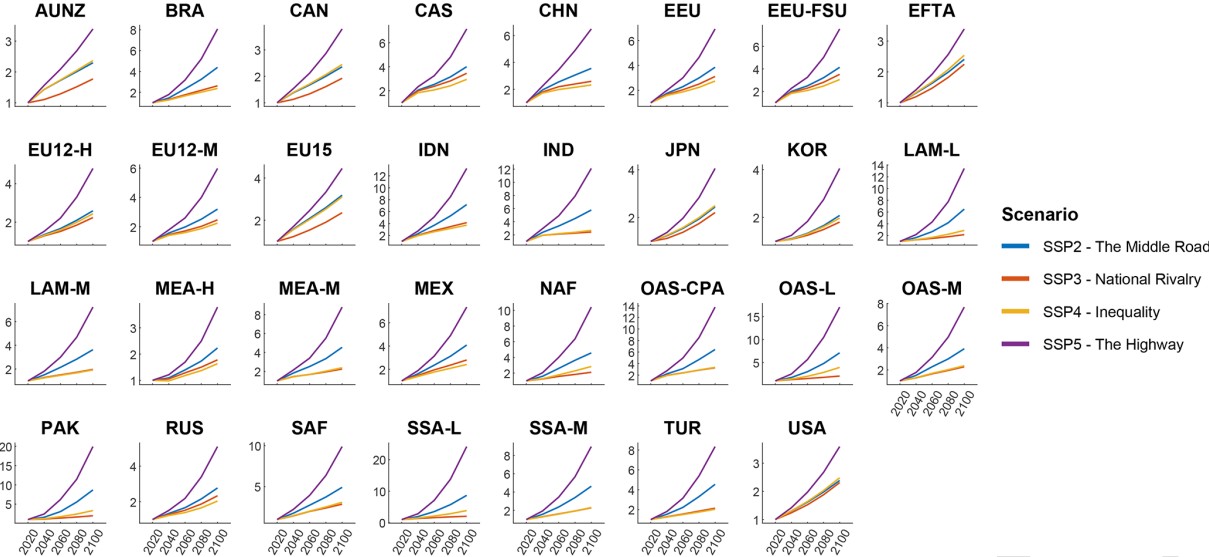

**Figure 14.** Regional $F_{GDP}^{cap}$ factor variation in the SSP IIASA database (R32). Source: https://tntcat.iiasa.ac.at/SspDb/ (last access: 14 October 2022). The country mapping is provided in Table A5.

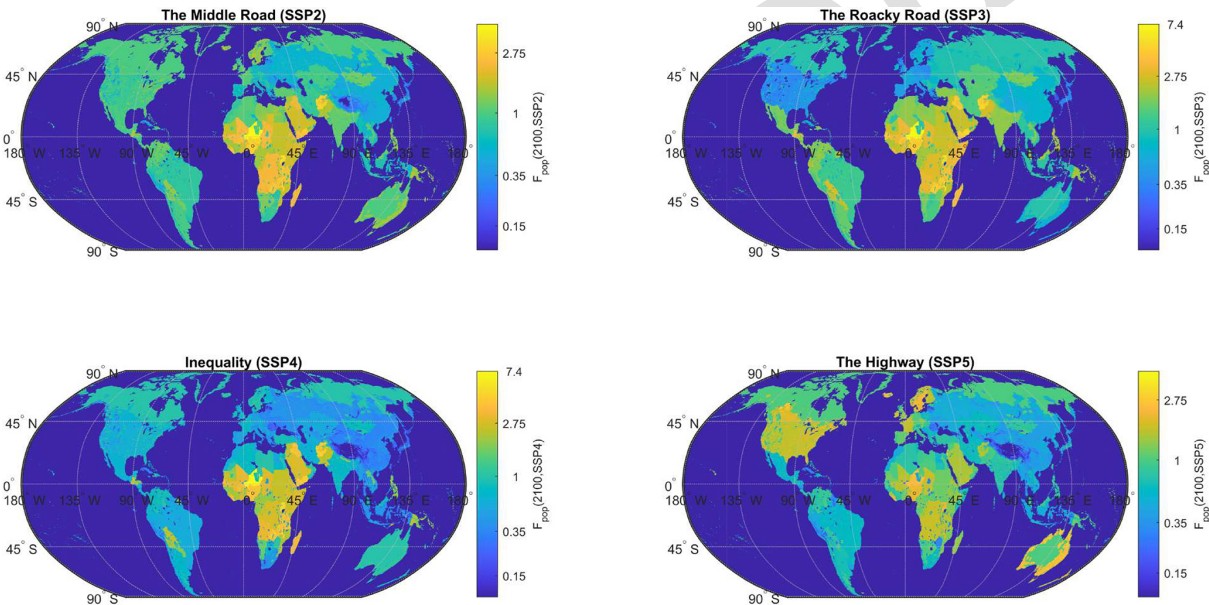

**Figure 15.** Local $F_{pop}$ factor variation in all SSPs in 2100 (source: Jones and O'Neill, 2020, NASA Socio-Economic Data Application Center – SEDAC). The scenario-based population grid generation methodology is detailed by Jones and O'Neill (2020) with a last version downscaled at 1 km following Gao (2020). This population grid is available every 10 years. CATHERINA uses the closest value in the definition of the exposure.

Sapir et al., 2018) (available at http://www.emdat.be (last access: 14 October 2022) This database, used in most studies on the topic, accounts for the total reported damage (sum of all sub-perils) and does not distinguish damages from sub-perils. Filtering the database by subtype "tropical cyclone" allows us to extract 1855 tropical cyclones in the period between 1980 and 2021, among which 1101 events have a reported total damage cost in USD (see Fig. 17). In terms of

damage, tropical cyclones are, using the full set of observations from 1980 to 2021, the most damaging events reported (see Fig. 16). The database includes a start date field (day, month and year) allowing us to map 455 events, with the events reported in IBTrACS using the start year and month

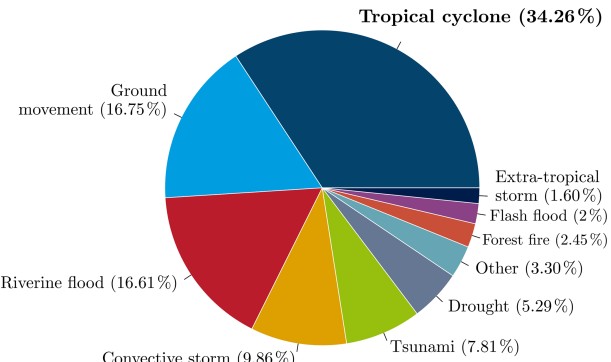

**Figure 16.** Proportion of damage cost (total damage in USD) by disaster subtypes reported in EM-DAT. Using the number of people affected places tropical cyclones after riverine floods and droughts, and the number of deaths places ground movements in the first position (see Table A1 for details).

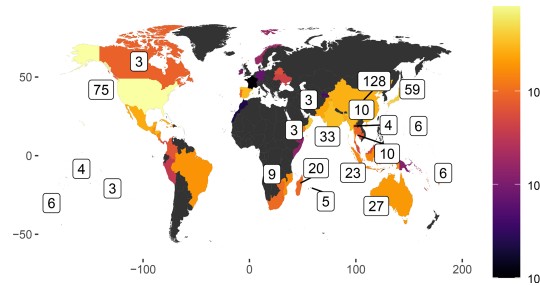

**Figure 17.** Number of tropical cyclones with reported damage cost in the EM-DAT database. The color scale indicates the average reported damage cost in USD in each country for each tropical cyclone.

and country.[11] We use this database to validate our simplified estimation process.

## 4.2 An explicit damage function

Damage functions can take several different forms (Prahl et al., 2015), but the most common choice is a cubic functional of the wind speed. To estimate the fraction of loss from a storm with sustained wind speed $V$, Emanuel (2011) introduced the following formula:

$$f(V, v_h^j) = \frac{(\max(V - v_0, 0))^3}{(v_h^j - v_0)^3 + (\max(V - v_0, 0))^3}, \quad (18)$$

where $f$ is the fraction of the property value lost, $v_0 = 25.7 \, \text{m s}^{-1}$, and $v_h^j$ is a parameter that needs to be calibrated for each region $j$. Figure 18 illustrates the shape of the damage function for different values of this parameter.

To account for adaptation, we could modify this function and introduce an additional threshold value. For instance, to

[11]Eberenz et al. (2021) functions are fitted on a similar sample of 376 tropical cyclones used for calibration.

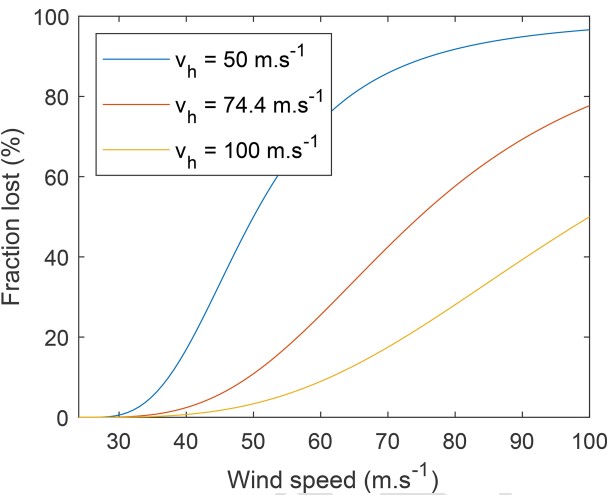

**Figure 18.** Fraction of property value lost as a function of wind speed ($\text{m s}^{-1}$), obtained using Eq. (18) with different values of $v_h$. Source: Emanuel (2011).

account for local adaptation to the wind climate, Leckebusch et al. (2007) suggest scaling the wind value by the 98th percentile of the local wind speed distribution. However, the assumption that adaptation will always keep the damages from the 98th percentile wind at the same level is probably optimistic and would prevent us from using the model to estimate the required investment to balance future damages and economic shocks.

## 4.3 To a regional damage calibration

Using the reported damage estimates from EM-DAT crossed with cyclone tracks (IBTrACS) and geographical and socioeconomic information along these tracks, Lüthi (2019) refined the damage function approach using machine learning techniques introducing region-specific damage functions.

We recall the main steps of the methodology presented in Eberenz et al. (2021) to define the regional damage functions. The authors first defined the event damage ratio (EDR) as a ratio of simulated damage (SED) to normalized reported damage (NRD) for each cyclone:

$$\text{EDR}(i, j) = \frac{\text{SED}(i, v_h(j))}{\text{NRD}(i)}. \quad (19)$$

The total damage ratio (TDR) is then defined in each region by summing over events:

$$\text{TDR}(j) = \frac{\sum_i \text{SED}(i, v_h(j))}{\sum_i \text{NRD}(i)}. \quad (20)$$

For each event, there is a value for $v_h$ allowing us to optimally calibrate the explicit damage function given in Eq. (18). The relatively wide distribution of $v_h$ for the same country shows that there is a large uncertainty in the relationship between the wind speed and the corresponding fraction of losses. Figure A4 shows the uncertainty in regional

**Table 2.** Values of $v_h$ ($\mathrm{m\,s^{-1}}$) obtained using the TDR and RMSF methods for each region from the CLIMADA environment.

| Region | $v_h^\star{}_{\mathrm{TDR}}, \mathrm{m\,s^{-1}}$ | $v_h^\star{}_{\mathrm{RMSF}}, \mathrm{m\,s^{-1}}$ |
|---|---|---|
| Caribbean and Mexico (NA1) | 58.8 | 59.6 |
| Chinese mainland (WP3) | 101.5 | 80.2 |
| USA and Canada (NA2) | 80.5 | 86 |
| Northern Indian (NI) | 63.7 | 58.7 |
| South-East Asia (WP1) | 60.7 | 56.7 |
| Northwestern Pacific (WP4) | 169.6 | 135.6 |
| Philippines (WP2) | 167.5 | 84.7 |
| Oceania (OC) | 56.8 | 49.7 |
| Southern Indian (SI) | 48.5 | 46.8 |
| Global (GLB) | 98.9 | 73.4 |

Coefficient from version 1.5 of the CLIMADA environment. Figure A4 also illustrates the shapes of the functions for the different optimization problems (RMSF vs. TDR) and version (1.0 vs. 1.5).

damage functions depending on the optimization technique used, and Fig. A5 allows us to appreciate, for countries where more than five cyclones were reported, the spread of plausible damage functions.

Eberenz et al. (2021) propose two alternative optimization methodologies to find the value of $v_h^\star$ maximizing the prediction quality of the regional damages: root mean square fraction (RMSF), minimizing the spread of the EDRs,

$$v_h^\star{}_{\mathrm{RMSF}}(j) = \mathrm{argmin}_j \exp\left(\sqrt{\frac{1}{N}\sum(\ln(\mathrm{EDR}(i)))^2}\right), \quad (21)$$

and TDR, finding the value of $v_h^\star$, such that the ratio of total simulated damage – obtained summing over event damages – and total reported damage tends to 1.

$$v_h^\star{}_{\mathrm{TDR}}(j) = \mathrm{argmin}_j |\mathrm{TDR}(j) - 1| \quad (22)$$

The values of $v_h^\star$ obtained by Eberenz et al. (2021) with the two methods are given in Table 2. For most regions, the optimized curves are similar for the two optimization techniques, but the results diverge for the Philippines (WP2) and to a lesser extent for Mainland China (WP3) events. The case of the Philippines, for example, discussed in Eberenz et al. (2021), is explained by the large number of parameters involved in the damage estimation and emphasizes two main limitations of the model. First, this framework lacks an explicit representation of sub-perils that disrupt and damage several sectors and services. Second, differences in exposure and vulnerability between urban and rural areas exposed to tropical cyclones are likely to contribute to the large spread in EDR.

### 4.4 Simplified damage estimation along tracks

In the context of our national level assessment, we propose a simplified damage module. The simulated damage for a given cyclone – in both IBTrACS and our synthetic tracks

– is computed using the following procedure for each individual cyclone. First, a uniform grid of physical asset values with steps given by the average cyclone radius is defined on the map of the affected area. The cyclone track is linearly interpolated, and the tiles affected by the cyclone (containing a part of the interpolated path) are identified (see Fig. 19).

Second, for each tile identified in the previous step, we retrieve the maximum wind speed $V$ and compute the proportion of wealth lost $f(V, v_h^j)$ using the relation (18) with the total damage ratio parameter given in Eberenz et al. (2021). Then, we compute the total simulated damage by aggregating the physical asset exposure multiplied by the proportion of wealth lost on each tile over all tiles affected by the cyclone.

As a result of this procedure, we obtain the total simulated damage $\mathrm{SED}_i(j, t)$ caused by the $i$th cyclone in region $j$, simulated with climate variables for year $t$. Finally, the cyclone damage cost in region $j$ and year $t$ is simulated as follows:

$$\mathcal{D}(j, t) = \sum_i \mathrm{SED}_i(j, t), \quad (23)$$

where the sum is taken over all cyclones occurring in a given year. This procedure can then be repeated many times to obtain the distribution of annual cyclone damages and to compute other statistics such as the mean and quantiles of this distribution.

The damage functions used in the second step are retrieved directly from the CLIMADA environment. These functions were fitted with the same physical asset value. However, in our case, we project these values onto a coarser grid (first step) in such a way that the extraction is simplified for a large number of synthetic tracks in the context of the present global exercise. To ensure that the estimated damages produced with this simplification are consistent with the historical records, we computed simulated damages over the historical tracks and compared the results to EM-DAT. We aggregate asset values on a $0.25 \times 0.25°$ grid. The spread between SED and NRD distribution remains important. To further reduce the errors, we thus divide the simulated damage by the average SED-to-NRD ratios in each region, which are computed using the RMSF damage function on IBTrACS and total reported damage from EM-DAT:

$$\hat{r}_j = \frac{\sum_i \mathrm{SED}(i, v_h^{\mathrm{RMSF}}(j))}{\sum_i \mathrm{NRD}}. \quad (24)$$

Figure 20 presents the estimated vs. actual damages computed using the RMSF damage function and the distribution of the re-scaled estimation by country using the intersection of IBTrACS with lands (762 events), crossed with EMDAT (606). Each dot represents the total damage over a year in a country (211 observations).

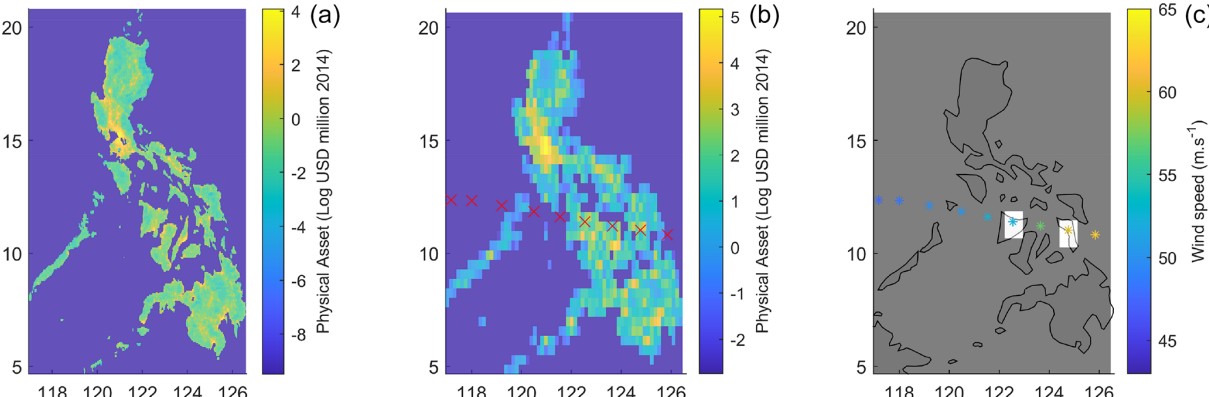

**Figure 19.** Illustration of the high-resolution asset data **(a)** original physical asset value resolution. **(b)** Aggregation of the asset value over $0.25°$ boxes to evaluate asset exposure along the interpolated cyclone track (here corresponding to the 2013 Hayan cyclone). **(c)** Illustration of damage calculation: damage is aggregated over the white boxes which correspond to cyclone locations over land.

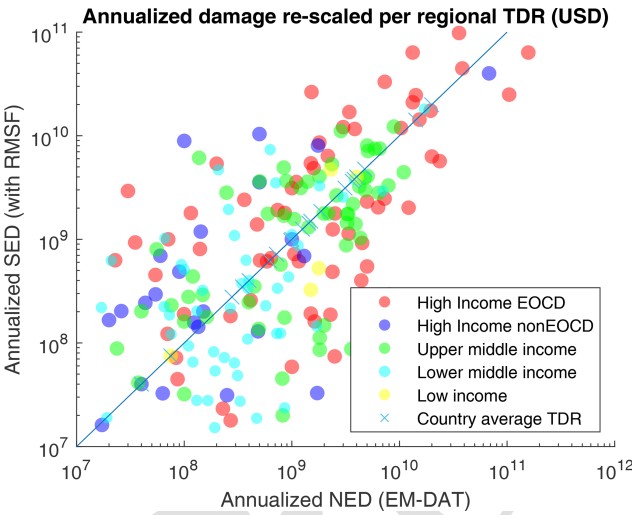

**Figure 20.** Estimated (RMSF) vs. reported damage (USD). The filled dots represent individual year–country pairs with reported damage in EM-DAT. This distribution is obtained using a $0.25 \times 0.25$ resolution projection. Crosses correspond to country averages (over all years).

## 5   Application to RCPs

### 5.1   Climate simulation bias correction for climate change application

The variables from climate model projections used by CATHERINA are subject to multiple biases. To reduce uncertainty caused by these biases, we use the cumulative distribution function transform (CDF-t) method developed in Vrac et al. (2012) and Michelangeli et al. (2009) to correct the distribution of each variable in each basin. Our bias-correction approach is the standard in the climate community (Navarro-Racines et al., 2020) (see http://ccafs-climate.org/bias_correction/, last access: 14 October 2022).

Consider a generic climate variable (denoted by $\chi$) at a fixed location, which is available both from ERA5 reanalysis and from a given CMIP5 model. We are interested in two time periods: the historical period (covered by both the climate model and the reanalysis) and a future time period (covered only by the climate model). Let $F^{\mathrm{h}}_{\mathrm{ERA5}}$ and $F^{\mathrm{h}}_{\mathrm{CMIP}}$ be the distribution functions of $\chi$ under reanalysis and under the climate model for the historical period, and $F^{\mathrm{f}}_{\mathrm{CMIP}}$ is the distribution function of $\chi$ under the climate model for the future period. The distribution function under the climate model is subject to much stronger biases than that under the reanalysis. The CDF-t method constructs the distribution function for $\chi$ with a reduced bias for the future time period, denoted by $\widehat{F}^{\mathrm{f}}_{\mathrm{CMIP}}$ and given by

$$\widehat{F}^{\mathrm{f}}_{\mathrm{CMIP}}(\cdot) = F^{\mathrm{h}}_{\mathrm{ERA5}}(F^{\mathrm{h,-1}}_{\mathrm{CMIP}}(F^{\mathrm{f}}_{\mathrm{CMIP}}(\cdot))), \tag{25}$$

where $F^{\mathrm{h,-1}}_{\mathrm{CMIP}}$ is the inverse function of $F^{\mathrm{h}}_{\mathrm{CMIP}}$. For a given value $\chi^{\mathrm{f}}_{\mathrm{CMIP}}$ of the variable $\chi$ obtained for the future period from the climate model, the corresponding bias-corrected value $\hat{\chi}^{\mathrm{f}}_{\mathrm{CMIP}}$ may then be computed via

$$\begin{aligned}\hat{\chi}^{\mathrm{f}}_{\mathrm{CMIP}} &= \widehat{F}^{\mathrm{f,-1}}_{\mathrm{CMIP}}(F^{\mathrm{f}}_{\mathrm{CMIP}}) \\ &= F^{\mathrm{f,-1}}_{\mathrm{CMIP}}(F^{\mathrm{h}}_{\mathrm{CMIP}}(F^{\mathrm{h,-1}}_{\mathrm{ERA5}}(F^{\mathrm{f}}_{\mathrm{CMIP}}(\chi^{\mathrm{f}}_{\mathrm{CMIP}})))).\end{aligned}$$

When the future period and the historical period coincide, the method reduces to the standard quantile transform:

$$\hat{\chi}^{\mathrm{h}}_{\mathrm{CMIP}} = F^{\mathrm{h,-1}}_{\mathrm{ERA5}}(F^{\mathrm{h}}_{\mathrm{CMIP}}(\chi^{\mathrm{h}}_{\mathrm{CMIP}})). \tag{26}$$

First, we use the method on the historical period to compare the description of the thermodynamic potential and wind speed with and without correction, so Eq. (26) may be used directly. To extract the CDFs of the variables of interest, we generate synthetic track candidates from 1980 (beginning of ERA5) to 2010. We launch the simulation 10 times over these 30 years to obtain 300 representative years. By definition, for the genesis of the cyclones, the time of year and

location are in line with historical cyclone data. However, in this module, the synthetic tracks are generated without climate constraints; i.e., cyclones are allowed to drift relatively far away from their genesis location (in the limits of their initial basin) and therefore can cover conditions which do not lead to the formation of tropical cyclones. At this stage, these tracks are not to be considered "tropical cyclone tracks" but "candidate" tracks. In the following stage, actual cyclone tracks will be generated from candidate tracks by filtering the ones where meteorological conditions for cyclone formation are satisfied. For each point in space and time along these synthetic tracks, we extract the values of the four climate variables from the reanalysis (ERA5) and from the historical simulations of the seven climate models. Then, by comparing the CDF of the climate variables estimated by the models with the reference CDF computed from the reanalysis, we determine the transformation allowing the values estimated by the models to better match those from ERA5.

The sea level pressure distributions are stable over basins and models. The tropospheric temperature and near-surface relative humidity distributions depend largely on the basin and display evidence of non-negligible model uncertainty. The sea surface temperature estimates along the same synthetic tracks in the historical period display much larger uncertainty. The northern Indian basin presents the widest uncertainty for all climate variables, which adds further uncertainty concerning the impact of climate change on tropical cyclones in this area.[12]

Individual variables entering the MPI computation are correlated as shown in Table 3. For example, sea surface temperature and tropopause temperature exhibit a negative correlation of 83 %. Therefore, applying bias correction to individual variables may lead to unrealistic combinations when evaluating the thermodynamic potentials. For example, extremely low tropopause temperatures associated with very high SSTs lead to unrealistically high lapse rates and therefore unrealistically large potentials. To overcome this issue we perform the bias correction on the MSLP, SST, thermodynamic efficiency factor $\mathcal{E}$, and relative humidity – which should not be strongly correlated to other variables according to the level of correlation present in the reanalysis – see Table 3. Figure 21 shows that this correction leads to similar distribution of the thermodynamic potentials in the models and the reanalysis.

We apply the CDF-t correction technique along our historical synthetic tracks and compute the maximum potential intensity following Sect. 2.4.2. The pressure follows the dynamic process introduced in Sect. 2.4.4, and the corresponding wind is derived from the WPR (see Sect. 2.4.1). We define the model error as the relative error ($\frac{\chi_{\text{CMIP}} - \chi_{\text{ERA5}}}{\chi_{\text{ERA5}}}$) be-

tween the value produced by the model and the one produced by the ERA5 reanalysis. Figure 22 displays the average relative errors and shows that a 2 % relative error in the description of the maximum potential intensity can lead to more than 20 % error in the description of the implied wind compared to the result obtained with ERA5. This figure illustrates the efficiency of bias correction in the historical period. Indeed, on average, the CDF-t correction technique clearly reduces the error between the MPI estimated with climate reanalysis and the one computed from modeled climate data as well as (more importantly) the error in the description of the maximum wind speed. However, running CATHERINA on the different GCMs still produces a wide range of results in climate projections (cf. Fig. 23). Although the intensity of the storms increases with each model, underlying climate modeling uncertainty still strongly impacts the synthetic tracks produced.

## 5.2 Results in CMIP5 projections

The international climate modeling community introduced SSPs to translate varying narratives on the development of society in the long term. These projections impact the local physical asset value dynamics (Jones and O'Neill, 2020; Chen et al., 2020) and global macroeconomic variables (O'Neill et al., 2014, 2017). Under the assumptions of constant cyclone genesis frequency and a constant impact ratio (i.e., the damage functions remain the same), CATHERINA allows us to derive damage projections in varying climate and socioeconomic scenarios. Using bias-corrected climate variable projections from the seven climate models over the period 2070–2100, we provide an example of the application of the CATHERINA framework.[13] As expected (cf. Mendelsohn et al., 2012, Fig. 3, for example), socioeconomic change leads to wider differences than climate change.

Figure 24 represents the expected value of future damage, and Table 4 provides the expected values (together with standard errors) along with the 50th, 66th and 95th percentiles of the global annual damage distribution in the simulations under various assumptions of future exposure growth (values of $\alpha_1$ and $\alpha_2$). In the discussion below, CATHERINA simulations of 300 representative years of synthetic tracks using ERA5 data over the 1980–2010 period are used as a baseline.

The choice $\alpha_1 = \alpha_2 = 1$ assumes linear increase in exposure with respect to both GDP per capita and local exposed population. Using $\alpha_1 = 1/3$, which is close to the value estimated in Bakkensen and Mendelsohn (2019), allows us to

---

[12]Figure A7 presents the CDF-t of climate data in the sub-sample. These important biases and uncertainties may be mitigated in the latest launch of the models on the occasion of CMIP6 (Gusain et al., 2020).

[13]Because of the time slicing of the CMIP5 climate data available in the climate data store, we launch the models in 25 consecutive years over this period: 2070–2095 for IPSL, BCC, NCC and CCCMA, 2075–2100 for GISS and 2085–2095 for INM (only a 10-year slice of climate data is available on the CDS for this model). We repeat this process, changing the seed 12 times to obtain 300 representative years in each model (except for INM, for which we only have 120 years).

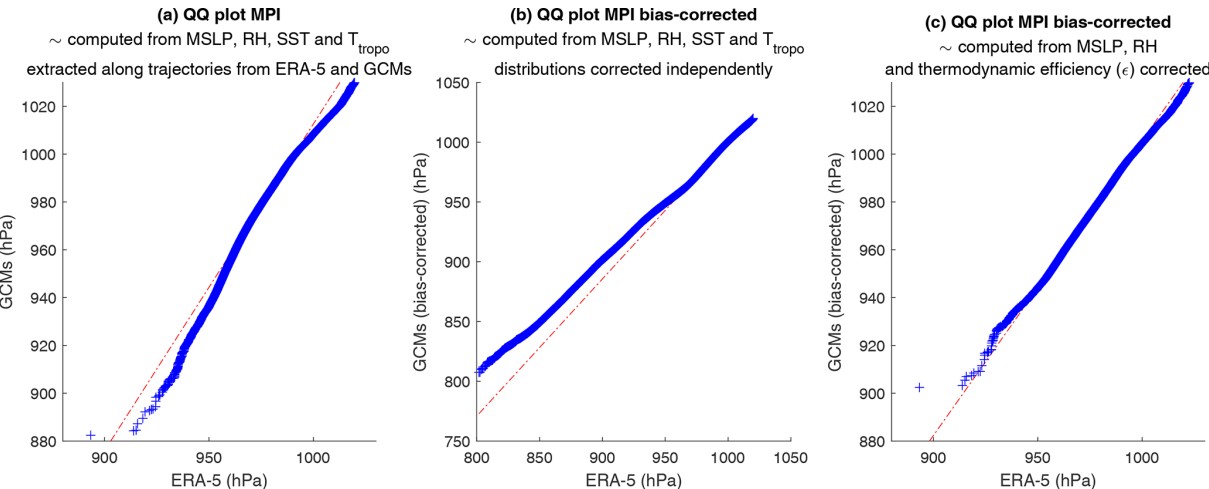

**Figure 21.** Bias-correction module Q–Q plots (GCMs vs. ERA5) of maximum potential intensity (hPa). The Q–Q plots compare two distributions by plotting quantiles of one against the other.

**Table 3.** Correlation levels (global) of modeled variables affecting the MPI with their reference values in ERA5.

|  | $\text{SST}_{\text{ERA5}}$ | $\text{SST}_{\text{GCMs}}$ | $\text{MSLP}_{\text{ERA5}}$ | $\text{MSLP}_{\text{GCMs}}$ | $T_{\text{ERA5}}^{\text{tropo}}$ | $T_{\text{GCMs}}^{\text{tropo}}$ | $\text{RH}_{\text{ERA5}}$ | $\text{RH}_{\text{GCMs}}$ |
|---|---|---|---|---|---|---|---|---|
| $\text{SST}_{\text{ERA5}}$ | 1 | | | | | | | |
| $\text{SST}_{\text{GCMs}}$ | 0.95 | 1 | | | | | | |
| $\text{MSLP}_{\text{ERA5}}$ | 0.46 | 0.39 | 1 | | | | | |
| $\text{MSLP}_{\text{GCMs}}$ | 0.45 | 0.38 | 0.86 | 1 | | | | |
| $T_{\text{ERA5}}^{\text{tropo}}$ | −0.83 | −0.82 | −0.37 | −0.33 | 1 | | | |
| $T_{\text{GCMs}}^{\text{tropo}}$ | −0.77 | −0.79 | −0.31 | −0.27 | 0.85 | 1 | | |
| $\text{RH}_{\text{ERA5}}$ | −0.07 | −0.07 | −0.12 | −0.09 | 0.22 | 0.19 | 1 | |
| $\text{RH}_{\text{GCMs}}$ | −0.27 | −0.30 | −0.18 | −0.14 | 0.33 | 0.37 | 0.47 | 1 |

account for possible future adaptation. The configurations $\alpha_1 = 1, \alpha_2 = 0$ and $\alpha_1 = 0, \alpha_2 = 1$ allow us to decompose the risk contribution between GDP and exposed population. Finally, the choice $\alpha_1 = 0, \alpha_2 = 0$ uses the current exposure value with no socioeconomic growth factor.

Assuming no future adaptation ($\alpha_1 = \alpha_2 = 1$), over the period 2070–2100, the RCP2.6 scenario, which is in line with the Paris Agreement and keeps global warming below 2 °C by 2100, involves a growth of expected global annual financial losses from tropical cyclones by a factor of 4.2 on average. Ignoring socioeconomic and population growth factors ($\alpha_1 = 0$ and $\alpha_2 = 0$), our model suggests that the expected financial loss would grow by a factor of 1.6 due to increasing cyclone intensity. Taking into account adaptation, i.e., limiting the growth of damage with respect to GDP per capita ($\alpha_1 = 1/3$), the expected damage would grow by a factor of 2.6. In the case of SSP2–RCP4.5 (between 1.7 and 3.2 °C warming by 2100) and SSP5–RCP8.5 (between 3.2 and 5.4 °C warming by 2100), the average expected damage will be multiplied by 5.4 and 14.2, respectively, without adaptation. In RCP8.5 the expected damage will still grow by a factor of 2.8, ignoring the change in GDP per capita

and population ($\alpha_1 = 0$ and $\alpha_2 = 0$). Interestingly, in SSP3, accounting for the population factor (that is, moving from $\alpha_2 = 0$ to $\alpha_2 = 1$) induces a decrease in global expected damage. This is due to a significant decrease in population in the USA, Australia, South Korea or Japan in the regions subject to tropical cyclones in this scenario. As these countries represent a large share of tropical-cyclone-related damages, the global expected damage is reduced.

Our expected damage estimates are subject to three types of uncertainty: internal climate variability, climate model uncertainty and socioeconomic uncertainty related to future exposure growth, adaptation measures and concentration pathways. The first two types of uncertainty are quantified by the standard errors in Table 4, while the last one may be evaluated by performing simulations under different assumptions about adaptation, SSP narratives and representative concentration pathways as illustrated in Fig. 24.

Figure 25 displays the average annual damage per country, in the different shared socioeconomic pathways, assuming no future adaptation. We can see that the distributions across countries are slightly different from one SSP to another. Indeed, the distributions in SSP2 and SSP5 are sim-

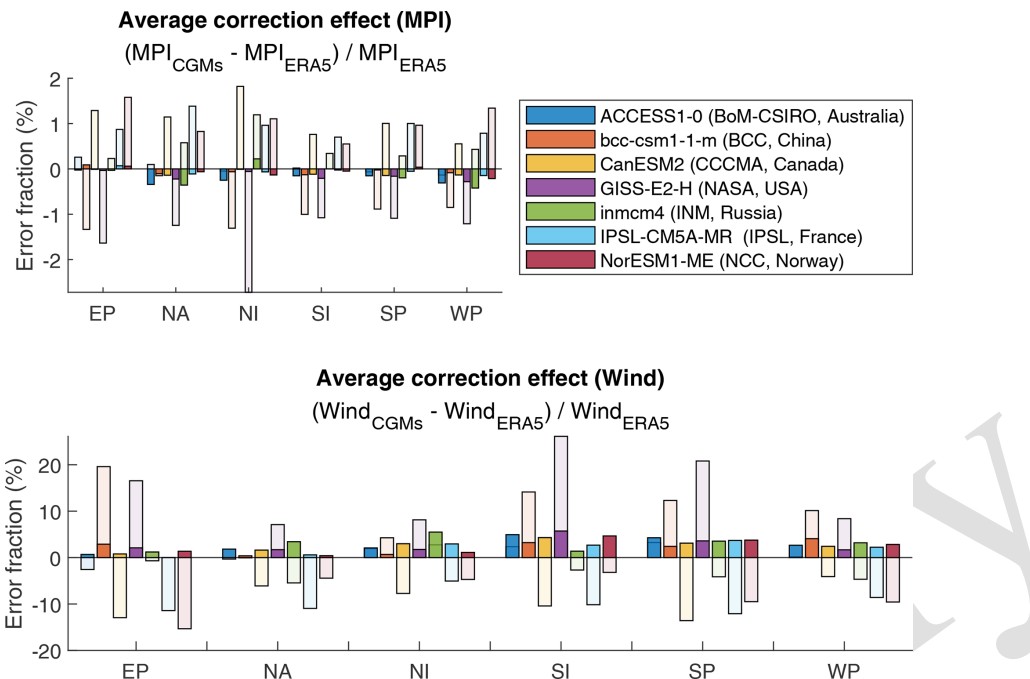

**Figure 22.** Relative error at the level of maximum potential intensity (MPI) computed with ERA5 and climate data produced by the seven climate models for the historical period. The transparent bars represent original errors, and the color parts represent the residual relative error after CDF-t correction averaged over 30 years of cyclones.

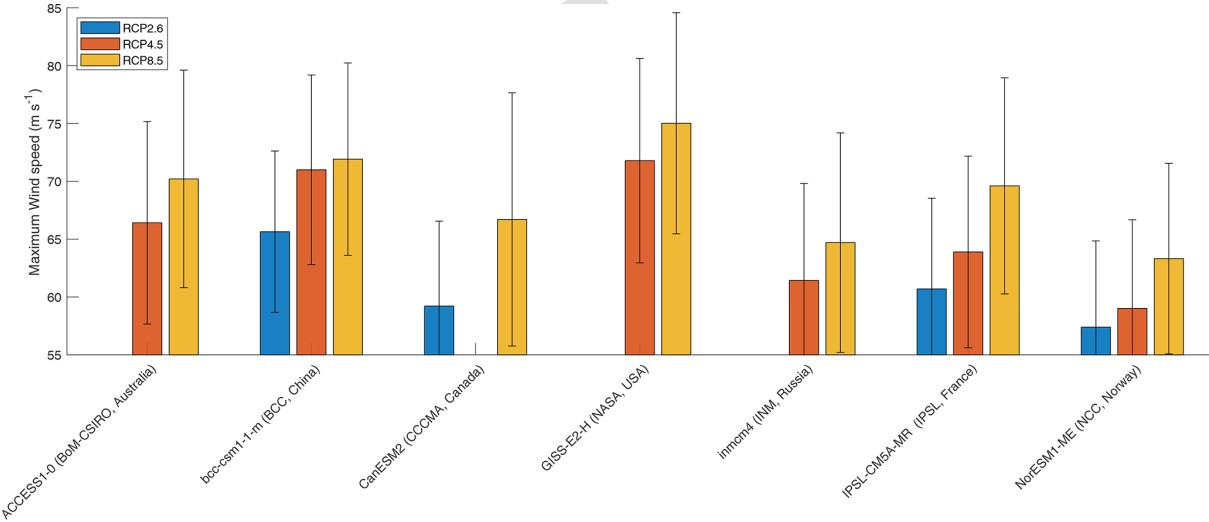

**Figure 23.** Average and standard deviations of the maximum wind speed of tropical cyclones generated with the CMIP5 models after correction. Table A4 provides the average value of maximum wind speed ($\mathrm{m\,s}^{-1}$).

ilar, with higher expected damage in SSP5 because of the growth hypothesis this scenario relies on. However, SSP3 (rocky road) and SSP4 (inequality) are distributed differently. The scenario emphasizing inequalities – and its interpretation by scientists in terms of (i) socioeconomic developments (Riahi et al., 2017) and (ii) population distribution (Jones and O'Neill, 2016) – increases damage concentration in the United States. On the other hand, the rocky-road scenario,

linked to a larger and more rural population, lower GDP and national rivalry, sees the damage more equally distributed among other nations.

Looking at the expected damage values is instructive, but because the aim of the model was also to stress test the resiliency of the financial and economic systems, it is important to examine the risk of extreme events, corresponding to higher quantiles of the loss distribution. Figure 26 and

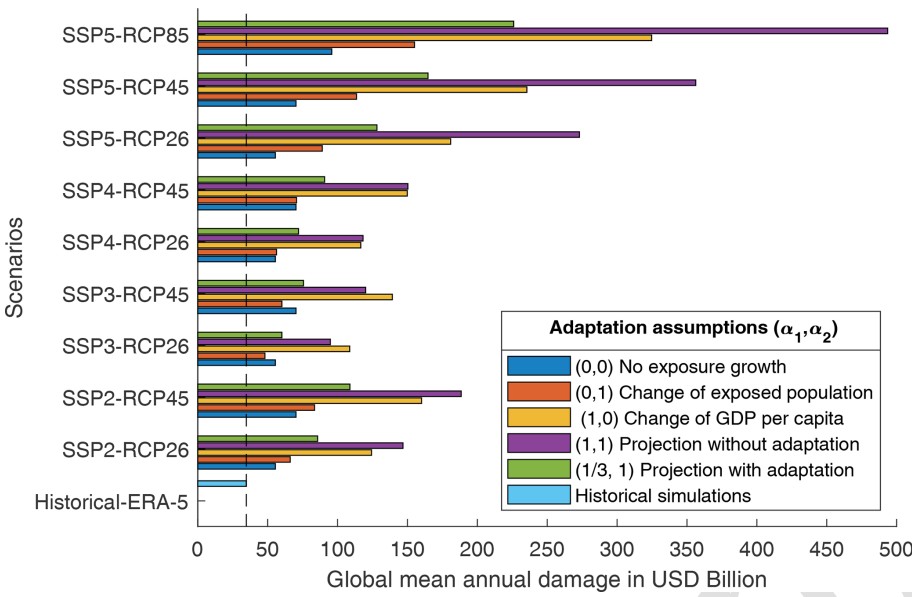

**Figure 24.** Expected value of global annual damage (in USD billion) in different SSP–RCP and exposure projection hypothesis configurations. The vertical dotted line corresponds to historical simulations with ERA5 data.

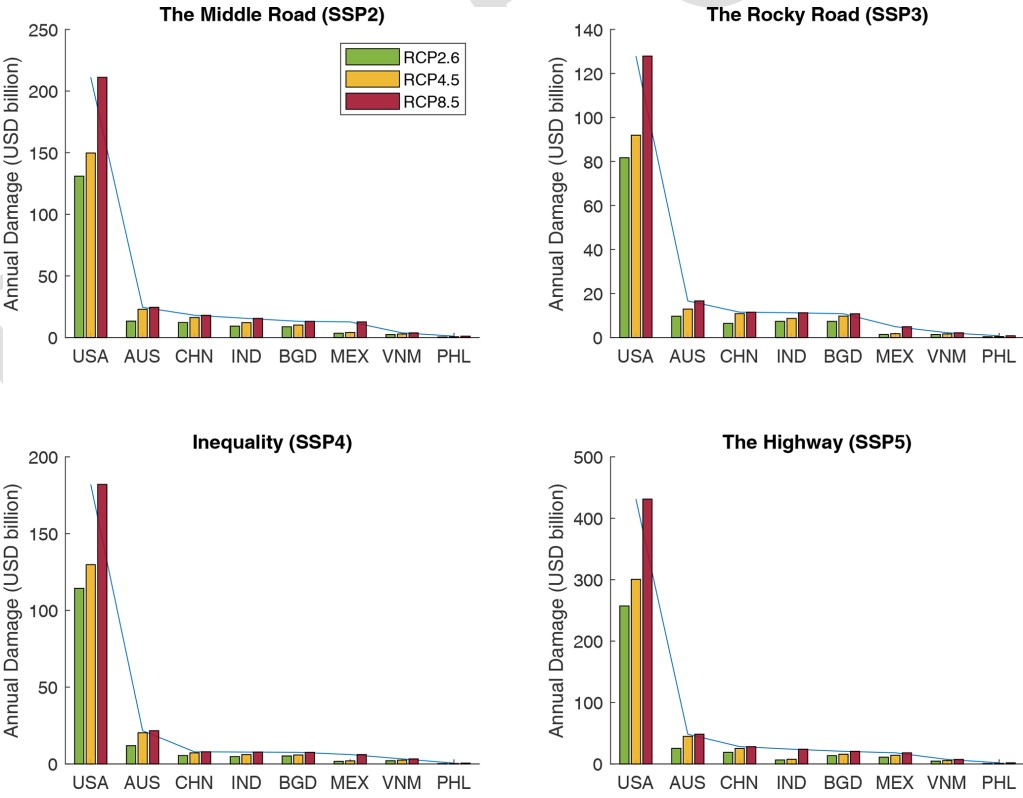

**Figure 25.** Annualized regional expected damage (in USD billion) in SSP and RCP between 2070 and 2100 based on synthetic tracks produced with seven climate models (with bias correction) over 300 representative years launched independently. The scale differs between SSPs.

**Table 4.** Simulation statistics (in USD billion).

| SSP | RCP | $(\alpha_1, \alpha_2)$ | Mean | Standard error | 50th perc. | 66th perc. | 95th perc. |
|---|---|---|---|---|---|---|---|
| | ERA5 | | 34.72 | 5.14 | 6.81 | 15.72 | 155.94 |
| Historical | IBTrACS | | 47.59 | N.R. | 25.72 | 42.94 | 160.88 |
| | Reported EM-DAT | | 21.1 | N.R. | 9.19 | 13.90 | 62.79 |
| | | (1,1) | 146.82 | 11.54 | 40.06 | 83.81 | 583.74 |
| | | (1/3,1) | 85.74 | 7.14 | 21.14 | 45.38 | 362.46 |
| | RCP26 | (0,1) | 66.14 | 5.63 | 15.31 | 32.64 | 286.85 |
| | | (1,0) | 124.31 | 8.82 | 39.61 | 78.72 | 492.96 |
| SSP2 | | (0,0) | 55.51 | 4.29 | 15.47 | 31.35 | 225.87 |
| | | (1,1) | 188.47 | 9.75 | 58.41 | 121.04 | 779.64 |
| | | (1/3,1) | 108.87 | 5.94 | 30.81 | 63.41 | 483.78 |
| | RCP45 | (0,1) | 83.54 | 4.66 | 22.56 | 48.69 | 376.56 |
| | | (1,0) | 160.12 | 7.64 | 57.59 | 112.31 | 637.80 |
| | | (0,0) | 70.28 | 3.64 | 22.43 | 45.00 | 300.45 |
| | | (1,1) | 120.10 | 6.18 | 40.61 | 76.71 | 478.18 |
| | | (1/3,1) | 75.60 | 3.95 | 25.26 | 46.68 | 315.38 |
| SSP3 | RCP45 | (0,1) | 60.18 | 3.18 | 19.61 | 36.83 | 255.95 |
| | | (1,0) | 139.20 | 7.10 | 46.09 | 93.13 | 580.09 |
| | | (0,0) | 70.28 | 3.64 | 22.43 | 45.00 | 300.45 |
| | | (1,1) | 118.24 | 10.09 | 27.27 | 58.87 | 501.24 |
| | | (1/3,1) | 72.07 | 6.18 | 16.29 | 35.57 | 313.51 |
| | RCP26 | (0,1) | 56.35 | 4.85 | 12.74 | 27.92 | 245.90 |
| | | (1,0) | 116.63 | 8.96 | 32.90 | 67.65 | 483.69 |
| SSP4 | | (0,0) | 55.51 | 4.29 | 15.47 | 31.35 | 225.87 |
| | | (1,1) | 150.33 | 8.38 | 40.57 | 85.66 | 664.69 |
| | | (1/3,1) | 90.75 | 5.10 | 24.13 | 52.11 | 413.71 |
| | RCP45 | (0,1) | 70.62 | 3.99 | 18.58 | 40.51 | 325.38 |
| | | (1,0) | 149.90 | 7.68 | 47.48 | 98.81 | 622.79 |
| | | (0,0) | 70.28 | 3.64 | 22.43 | 45.00 | 300.45 |
| | | (1,1) | 356.26 | 18.92 | 104.58 | 219.89 | 1478.45 |
| | | (1/3,1) | 164.69 | 9.22 | 41.80 | 91.28 | 730.27 |
| | RCP45 | (0,1) | 113.56 | 6.48 | 27.15 | 60.71 | 530.05 |
| | | (1,0) | 235.48 | 10.80 | 88.32 | 176.36 | 910.28 |
| SSP5* | | (0,0) | 70.28 | 3.64 | 22.43 | 45.00 | 300.45 |
| | | (1,1) | 493.56 | 27.04 | 140.69 | 281.66 | 2290.21 |
| | | (1/3,1) | 226.00 | 12.67 | 57.58 | 122.55 | 1072.03 |
| | RCP85 | (0,1) | 155.05 | 8.76 | 36.95 | 81.72 | 737.30 |
| | | (1,0) | 324.71 | 15.23 | 127.63 | 226.70 | 1366.12 |
| | | (0,0) | 95.89 | 4.81 | 30.70 | 59.48 | 445.36 |

Note: * the couple SSP5–RCP26 exists in the integrated assessment modeling literature (Rogelj et al., 2018) but is not displayed because the SSP5 is more likely tied to high concentration scenarios.

Table 4 show that the 95 % quantile of the loss distribution, corresponding to losses which arise, on average, once in 20 years, may be as high as 4–5 times the expected loss. This observation is in line with Coronese et al. (2019), who show that the impact of climate change is particularly striking for extreme events (see, for example, Fig. 2a in this reference).

## 5.3 Impact on sovereign bond spreads in RCPs

In this section, we briefly discuss the impact of cyclones on the sovereign bonds of the exposed economies. Particular attention will be given to emerging economies because of their higher vulnerability. We use an econometric model to relate the credit spread of sovereign bonds to the scenario-based distributions of damage developed in previous sections.

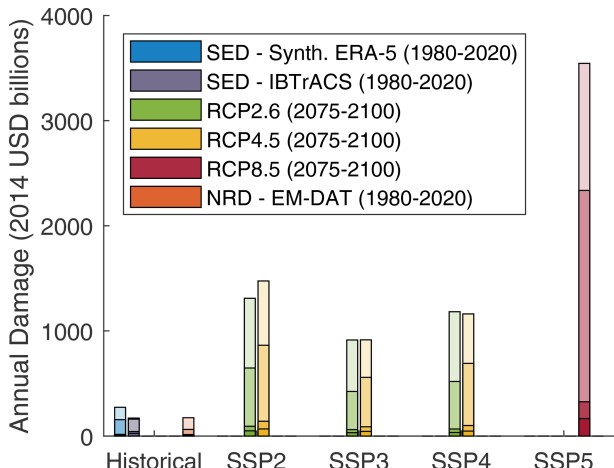

**Figure 26.** Annualized global damage (USD billion) in SSPs and RCPs between 2070 and 2100 based on synthetic tracks produced with seven climate models (with bias correction) over 250 representative years launched independently, assuming no adaptation ($\alpha_1 = \alpha_2 = 1$). Top range: 95 %–98 % (*extremely unlikely* losses), followed by 95 %–66 %, 50 %–66 % and 0 %–50 %. Synthetic historical damages are computed with 300 representative years of synthetic tracks generated with ERA5 between 1980 and 2010. The physical asset value is corrected by inflation between the cyclone year and 2014 (GDP reference year of Litpop). The global SSP–RCP configurations are presented as feasible in Rogelj et al. (2018).

### 5.3.1 Sovereign credit spread model

The sovereign credit spread is a financial measure of the creditworthiness of a country, which is defined as the difference of yield between the country's sovereign bond and a risk-free asset, i.e., an AAA-rated bond. In practice, we use the spread with respect to the US 10-year treasury bond. Following Hilscher and Nosbusch (2010), we calibrate a cross-sectional econometric model for the credit spread based on annual end-of-year data:

$$
\begin{aligned}
\mathrm{CS}_t = {} & \alpha + \beta_1 \Delta \mathcal{C}_t + \beta_2 D_t + \beta_3 \mathrm{VIX}_t + \beta_4 r_t^{10Y} \\
& + \beta_5 \mathrm{TED}_t + \beta_6 \frac{L_t}{\mathrm{GDP}_t} + \beta_7 \frac{\mathrm{reserves}_t}{\mathrm{GDP}_t} \\
& + \beta_8 \mathcal{R}_t + \beta_9 \mathcal{K}_t + \varepsilon_t,
\end{aligned}
\tag{27}
$$

where CS is the end-of-year credit spread from the JP Morgan EMBI position report in BarraOne, $\mathcal{C}$ is the commodity price index from Reuters Refinitiv, $D$ is the average duration of the bonds from the JP Morgan EMBI position report in BarraOne, VIX is the CBOE volatility index from Reuters Refinitiv, $r^{10Y}$ is the 10-year U.S. Treasury bond rate from Reuters Refinitiv, TED is the difference between the 3-month treasury bill and the 3-month LIBOR from Reuters Refinitiv, $L$ refers to the total external debt stocks in USD from the World Bank, GDP refers to the end-of-year GDP from the World Bank, reserves are the total reserves including gold in USD, $\mathcal{R}$ is the credit rating dummy variable, and $\mathcal{K}$ is the

country dummy variable. The model covers 74 countries between 2010 and 2020.

We chose to use all the bonds constituting the index, i.e., including multiple bonds per country per date. Table A7 shows the results of estimation of several submodels of our model. The model (1) assesses the effect of a variation of the debt / GDP ratio only (this is the most important variable that we will use to quantify cyclone damages), the model (2) introduces one rating dummy (it equals zero for all bonds with ratings below B− and one otherwise), the model (3) introduces additional macro variables and the model (4) adds country effect dummies and bond duration.

In line with Edwards (1986), we find that the debt / GDP ratio is significant in all the submodels. More importantly, we note that $\beta_6$ is relatively stable over the modeling frameworks, implying that the sensitivity of the spread to a sudden increase in the debt ratio does not strongly depend on external or idiosyncratic parameters. Including country effects in (4), we obtain a model reaching 71.5 % of the adjusted $R^2$.

### 5.3.2 Cyclone impact on sovereign spreads under representative concentration pathways

We now use the econometric model developed in the previous paragraph to assess the impact of tropical cyclones on emerging country bond spreads under relative concentration pathways. To this end, we make the following simplifying assumptions: (i) the bond spread model parameters remain stable over time, (ii) the cyclone damages are financed directly by the government by issuing new debt, and other variables of the model are not affected by cyclones, and (iii) only direct impact of cyclones is taken into account, and not the total economic costs. For each country $j$, we assess the annual bond spread variation due to cyclone damage in scenario $k$ for the year $t$ using the following formula:

$$
\Delta_{k,t} \mathrm{CS}(j,k,t) = \beta_6 \times \frac{\mathcal{D}(j,k,t)}{F_{\mathrm{GDP}}(j,k,t)\mathrm{GDP}(j,2020)},
\tag{28}
$$

where we recall that $\mathcal{D}$ stands for annualized cyclone damage, and $F_{\mathrm{GDP}}$ is the GDP growth factor for the specified country/scenario. For example, an event simulated in 2077 with GISS-E2-H (NASA, USA) in RCP85 generated USD 158 billion damage in Australia. Normalized by the future projected domestic product in the scenario (e.g., reference value USD 1.3 trillion with growth coefficient 3.1, leading to 4.4 trillion), this damage represents about 7 % of the country's GDP in the year concerned. This would lead to a shock in the Australian sovereign bond yield of 27 basis points (one basis points equals 0.01 %), implying a considerable increase in financing cost.

We compare the spread variation defined from damages of the RCP26 baseline, RCP45 and RCP85 to obtain an annualized financial valuation of the cyclone-related physical climate risk. The impact of average annual cyclone damage on the spread of sovereign bonds of larger emerging coun-

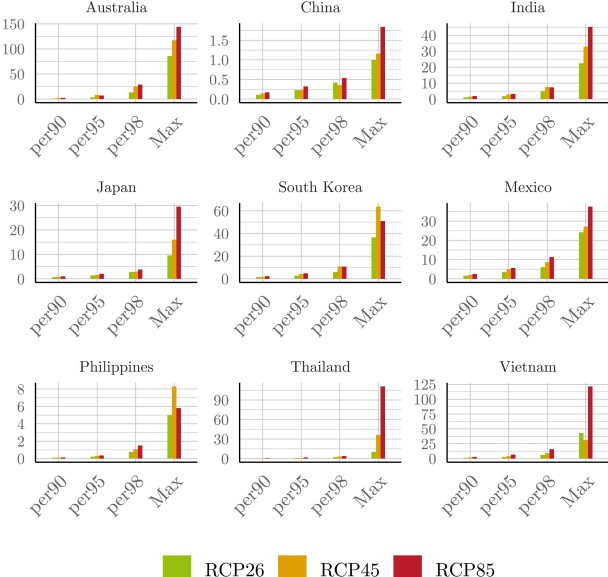

**Figure 27.** Damage cost channeled to excess spread (cost of debt) of different countries, measured in basis points (one basis point = 0.01 %).

tries, channeled by the impact on the debt-to-GDP ratio, is small. This is consistent with the historical data, as countries are not equally affected each year by tropical cyclones, and when they are, they develop a resiliency that is reflected in their damage functions. Therefore, we focus on the tail risk, that is, on the extreme event quantiles. Figure 27 represents the annual excess spread for a sample of countries, when the annual cyclone damages are given by the 90th, 95th and 98th percentiles of the distribution as well as by the maximum cyclone-related shock observed in the simulations. Recall that we have drawn 300 representative years from seven models with independent launches to increase the number of cyclone trajectories. On average, larger countries' excess spreads remain limited, but the maximum observed damages induce a significant excess cost of debt.

## 6 Conclusions

This paper proposes a structural framework to generate synthetic storms based on large-scale climate data and to produce simulations of annual cyclone damages per country. We show that, when used with reanalysis data and CMIP5 models over the historical period, our method produces tracks consistent with historical observations.

The synthetic tracks generated with our model have several applications. The first one is in natural disaster risk management to calibrate adaptation measures. For this purpose, the track-generation algorithm may be enhanced, for instance, by including dependency in the latitudinal and longitudinal incremental displacement, coupling with a meteorological forecasting model, or including ground topography to model the cyclone displacement over land. Another major field of application is to climate financial risk management, where this scenario-based event database can be used to evaluate physical risks and compute portfolio exposures. This would require us to better define asset level vulnerabilities.

The dataset used for the examples of this paper, based on low-resolution data and including a limited number of simulations, may not be accurate enough to properly calibrate adaptation measures. However, we believe that the framework presented here may be used to project a dense set of trajectories, compute expected damage and damage percentiles over the next decades and measure the investment required for adaptation and mitigation measures in the next 50 years. This work also reflects a practical exercise not carried out until now, of cross-referencing the latest datasets developed, putting into perspective both the socioeconomic and climatic development hypotheses and carrying out a bottom-up, rather than top-down, damage calculation.

**Appendix A**

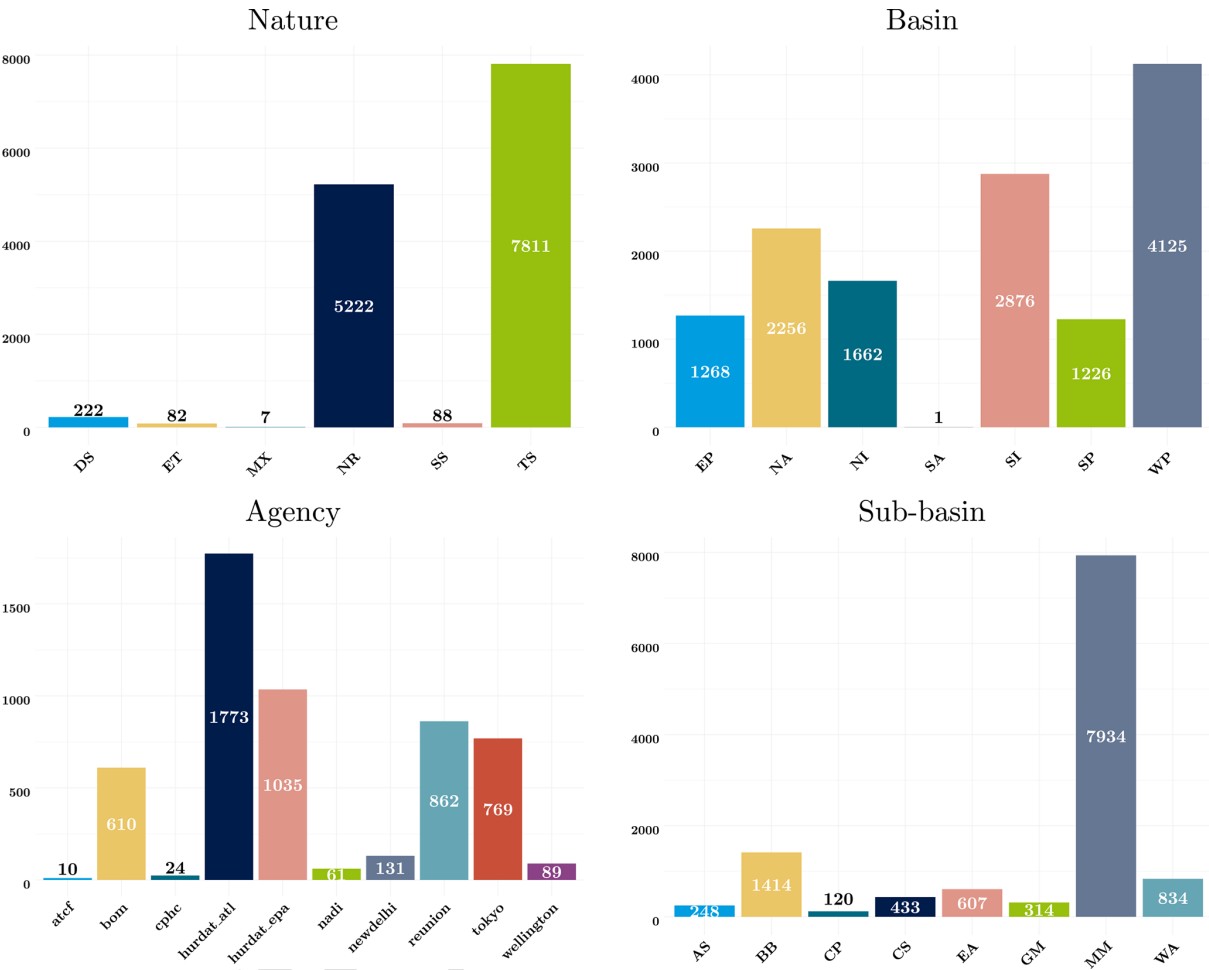

**Figure A1.** Information in IBTrACS. The cyclones are split by nature (DS: disturbance; TS: tropical; TE: extratropical; ST: subtropical; NR: not reported; MX: mixture or contradicting nature reports from different agencies). We remove extratropical cyclones (75) and disturbance (205). The cyclones are reported in the following basins: eastern Pacific (EP); North America (NA); northern India (NI); South Atlantic (SA); southern India (SI); South Pacific (SP); western Pacific (WP). One can also explore sub-basin decomposition (Eberenz et al., 2021) (MM: missing – no sub-basin for this basin (no sub-basins provided for WP, SI); CS: Caribbean Sea; GM: Gulf of Mexico; CP: central Pacific; BB: Bay of Bengal; AS: Arabian Sea; WA: western Australia; EA: eastern Australia), but we chose to use basins. The different agencies worldwide report central pressure and maximum wind speed but sometimes use sometimes different standards. In particular, the reporting can vary in terms of sustained wind speed. According to the dataset documentation, the North Atlantic – US Miami (NOAA NHC) bureau (hurdat/atcf) gives the 1 min winds speed, while Tokyo, i.e., RSMC Tokyo (JMA), provides directly the 10 min sustained wind speed. (Similarly, newdelhi corresponding to RSMC New Delhi (IMD) gives the 3 min wind speed, reunion RSMC La Reunion (MFLR), the Australian TCWCs (TCWC Perth, Darwin, Brisbane) (BOM), and the RSMC Nadi (FMS); TCWC Wellington (NZMS) provides the 10 min sustained wind speed and (CMA) 2 min sustained wind.) The lack of reporting standards between agencies is a source of uncertainty in the input data.

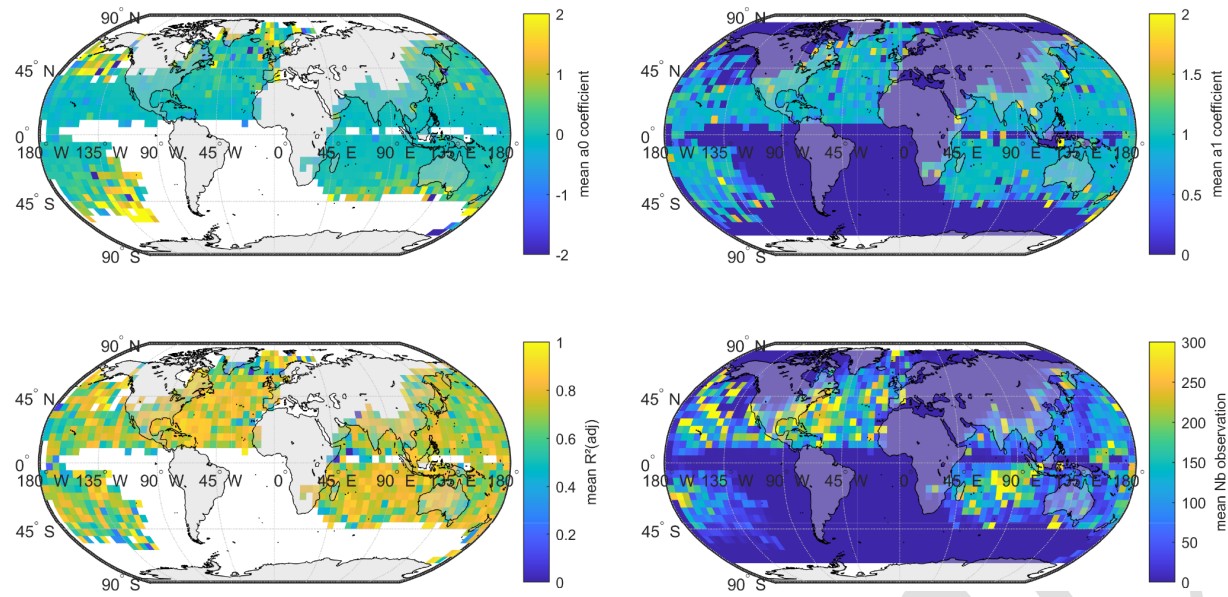

**Figure A2.** Map of longitude variation mean coefficients fitted on a $5 \times 5$ grid grouped per month.

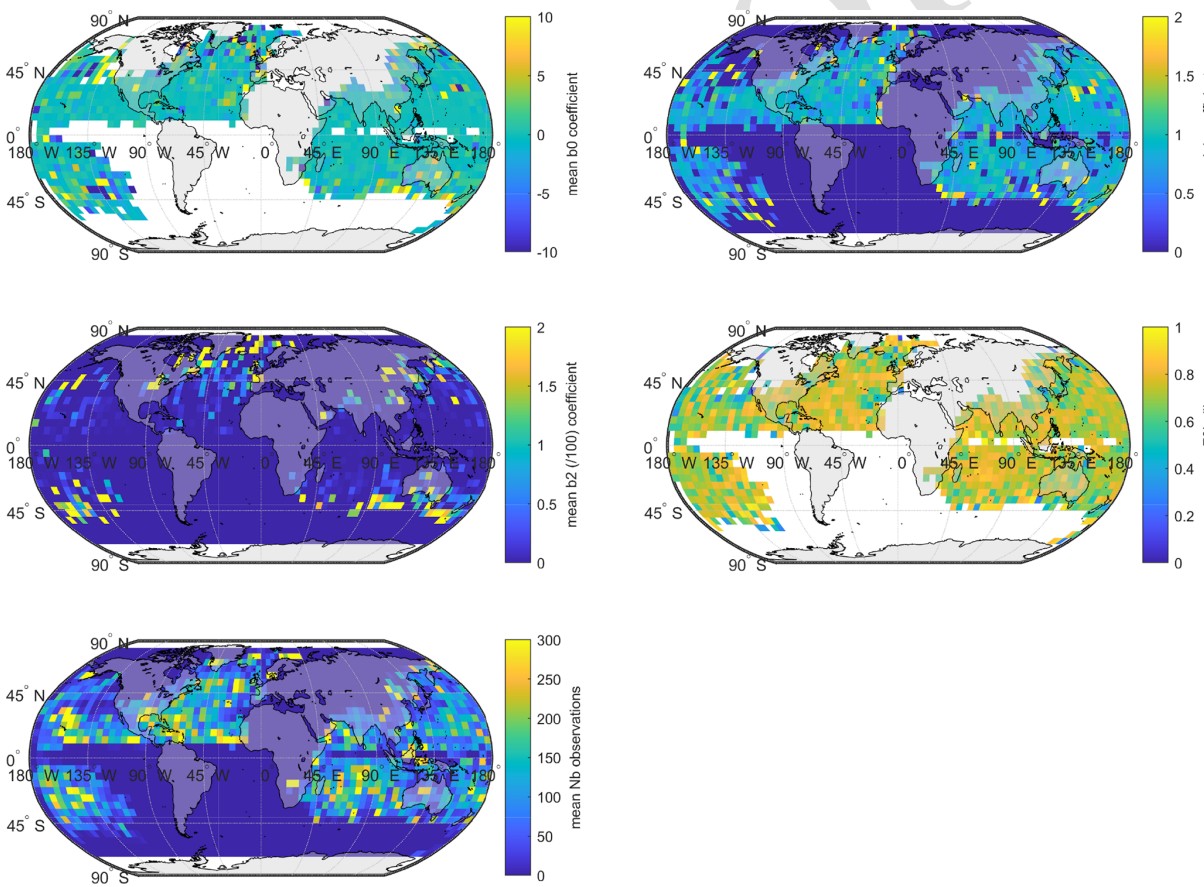

**Figure A3.** Map of latitude variation mean coefficients fitted on a $5 \times 5$ grid grouped per month.

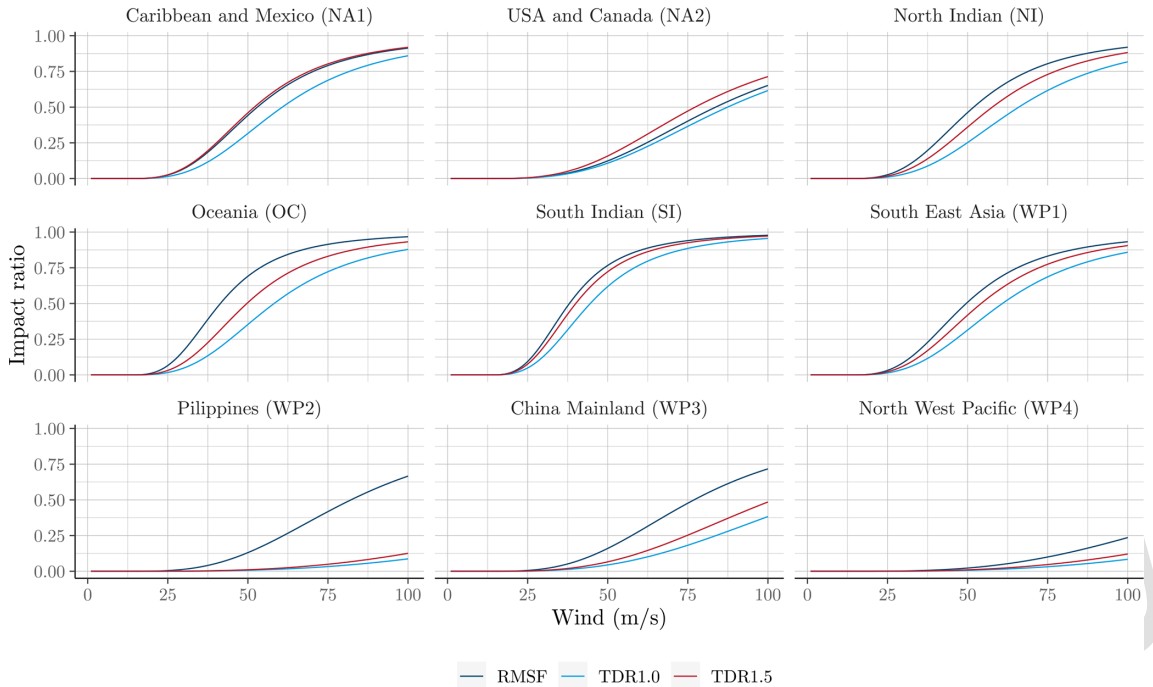

**Figure A4.** Regional optimal damage functions of the CLIMADA package (Eberenz et al., 2021).

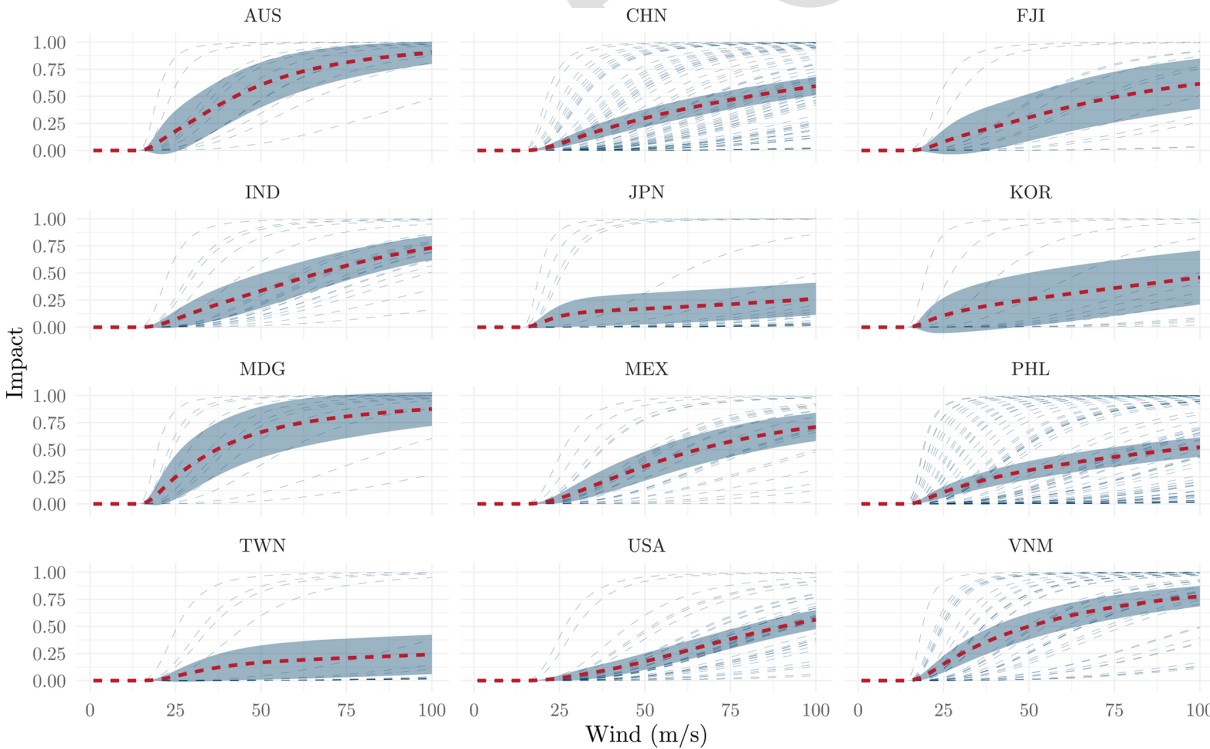

**Figure A5.** Country calibration of damage functions (when the countries have been hit more than five times).

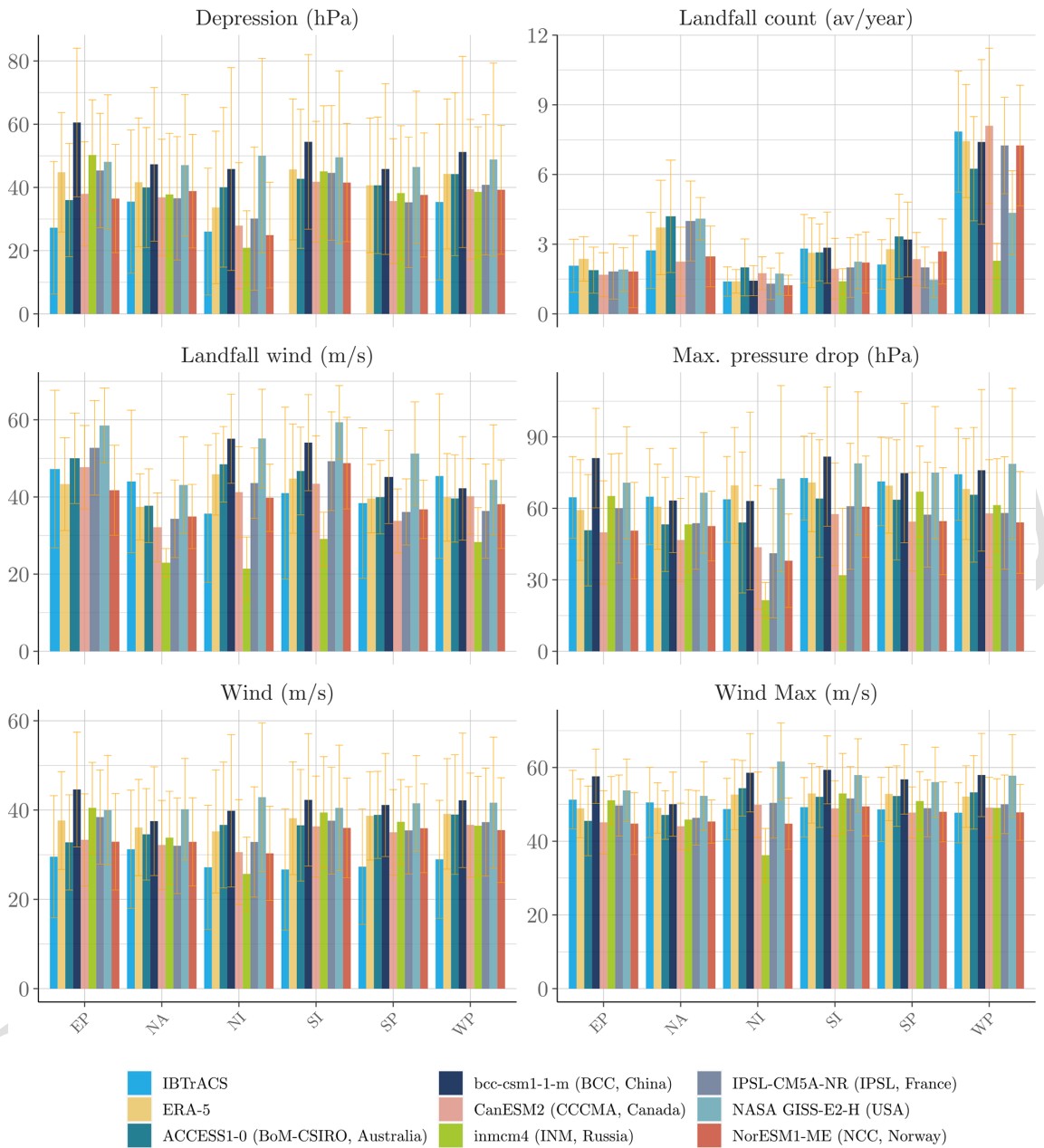

**Figure A6.** Evidence of model uncertainty. A comparison of the properties obtained by generating 30 years of tropical cyclones with different models from the CMIP5 raw climate data vs. ERA5. The average number of cyclones making landfall is computed by averaging the number of events with a number of steps on land positive ($s_l > 0$). Maximum pressure drop and wind are, respectively, computed by extracting the maximum value for the corresponding variables of the cyclone tracks. The light blue bars represent the mean of the variable of interest among the IBTrACS-filtered dataset (with the confidence interval drawn from the standard deviation of the distribution). The yellow bars represent the same variable extracted from the synthetic data generated by Algorithm 1 using ERA5 data. In terms of average values, the models produce consistent tracks in every basin. Then, we compare the output of this algorithm with different climate data produced on the historical period by the climate models taking part in the fifth phase of the CMIP. A general observation is the poorer performance of the model in the northern Indian basin. This could be due to the smaller number of intense cyclones remaining in the sample after the filtration by intensity (1.6 storms per year with winds exceeding $35\,\mathrm{m\,s^{-1}}$).

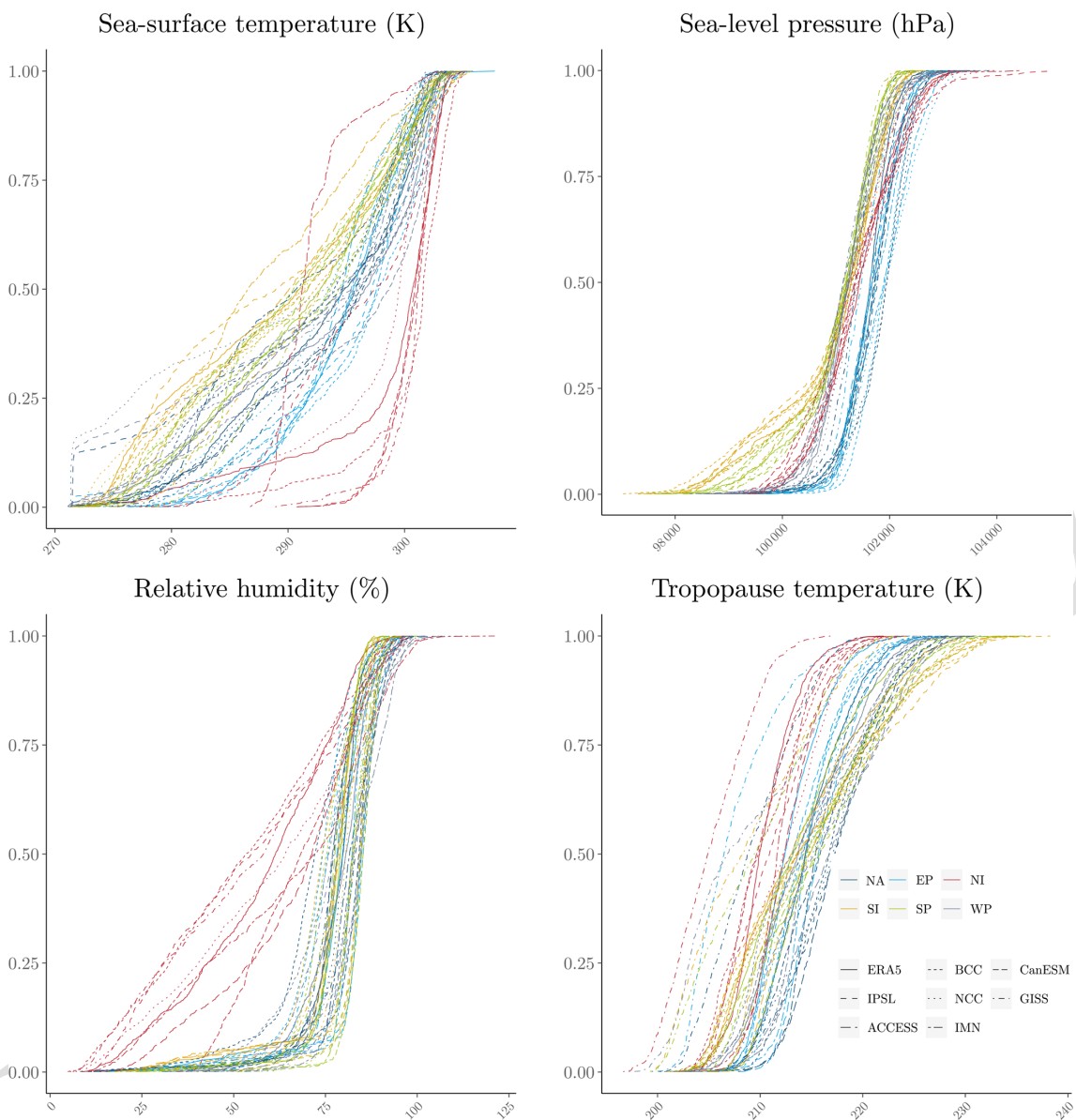

**Figure A7.** Cumulative distribution functions per basin for the variables of interest along synthetic tracks produced with ERA5 and extracted (at the same location) from climate data produced by the seven climate models. NA: North America, EP: eastern Pacific, NI: northern India, SI: southern India, SP: South Pacific, WP: western Pacific. The bias-correction module is indeed fitted on a larger range of climate conditions. By definition, for the genesis of the cyclones, the time of year and the location are in line with historical cyclone data. However, in the bias-correction module, the synthetic tracks are generated without climate constraints; i.e., cyclones are allowed to drift relatively far away from their genesis location (in the limits of their initial basin) and therefore can cover conditions which do not lead to the formation of tropical cyclones. At this stage, these tracks are not to be considered "TC tracks" but "candidate" tracks. In the following stage, TC tracks will be generated from candidate tracks by filtering those ones where meteorological conditions for cyclone formation are satisfied.

**Table A1.** EM-DAT reporting proportions[a]. Tropical cyclone-related statistics were emphasized.

|  | Total damages (USD) | Total deaths | Insured (USD) | Total affected | Reconst. costs[b] (USD) |
|---|---|---|---|---|---|
| Ash fall | 0.12 % | 1.09 % | 0.00 % | 0.09 % | 0.00 % |
| Avalanche | 0.02 % | 0.20 % | 0.02 % | 0.00 % | 0.00 % |
| Coastal flood | 0.30 % | 0.14 % | 0.14 % | 0.32 % | 0.00 % |
| Cold wave | 0.37 % | 0.70 % | 0.51 % | 0.25 % | 0.00 % |
| Convective storm | 9.86 % | 0.68 % | 21.78 % | 4.51 % | 0.00 % |
| Drought | 5.29 % | 24.75 % | 2.58 % | 36.35 % | 0.00 % |
| Extra-tropical storm | 1.60 % | 0.02 % | 3.09 % | 0.06 % | 3.31 % |
| Flash flood | 2.07 % | 2.58 % | 1.18 % | 3.03 % | 0.00 % |
| Forest fire | 2.45 % | 0.08 % | 3.86 % | 0.09 % | 0.76 % |
| Ground movement | 16.75 % | 26.78 % | 6.59 % | 2.61 % | 87.07 % |
| Heat wave | 0.68 % | 7.09 % | 0.03 % | 0.08 % | 0.00 % |
| Land fire[c] | 0.80 % | 0.03 % | 1.76 % | 0.02 % | 0.00 % |
| Landslide | 0.14 % | 1.04 % | 0.01 % | 0.15 % | 0.00 % |
| Lava flow | 0.02 % | 0.00 % | 0.00 % | 0.01 % | 0.00 % |
| Mudslide | 0.10 % | 0.21 % | 0.00 % | 0.02 % | 0.00 % |
| Pyroclastic flow | 0.01 % | 0.02 % | 0.00 % | 0.03 % | 0.00 % |
| Riverine flood | 16.61 % | 5.20 % | 6.59 % | 38.68 % | 4.46 % |
| Rockfall | 0.00 % | 0.02 % | 0.00 % | 0.00 % | 0.00 % |
| Severe winter cond. | 0.74 % | 0.16 % | 0.20 % | 1.27 % | 0.00 % |
| Subsidence | 0.00 % | 0.01 % | 0.00 % | 0.00 % | 0.00 % |
| **Tropical cyclone** | **34.26 %** | **18.33 %** | **45.64 %** | **12.34 %** | 2.23 % |
| Tsunami | 7.81 % | 10.87 % | 6.01 % | 0.11 % | 2.16 % |

Notes: [a] proportion excludes damages whose subtypes are not reported. [b] Reconstruction costs are not well reported over disaster subtypes. [c] Brush, bush, pasture. Among disaster subtypes, tropical cyclones present a higher quality in reporting and represent a large share of total damages.

**Table A2.** Wind pressure relationship coefficients (i.e., $V = a\,(\Delta P)^b$, $a$ in $\mathrm{hPa}^{-1}\,\mathrm{m\,s}^{-1}$).

| Basin | Coefficient | Estimate | Std error |
|---|---|---|---|
| EP | $a$ | 5.181 | 0.023 |
| EP | $b$ | 0.550 | 0.001 |
| NA | $a$ | 4.020 | 0.037 |
| NA | $b$ | 0.589 | 0.002 |
| NI | $a$ | 3.707 | 0.065 |
| NI | $b$ | 0.632 | 0.005 |
| SI | $a$ | 3.012 | 0.016 |
| SI | $b$ | 0.653 | 0.001 |
| SP | $a$ | 2.935 | 0.025 |
| SP | $b$ | 0.660 | 0.002 |
| WP | $a$ | 3.652 | 0.011 |
| WP | $b$ | 0.598 | 0.001 |

**Table A3.** Comparison STORM–CATHERINA cyclone intensification module.

|                   | STORM                        | CATHERINA                    |
| ----------------- | ---------------------------- | ---------------------------- |
| No. of variables  | 2                            | 4                            |
| MDP definition    | SST–MPD relationship – Eq. (4) | SST–MPD relationship – Eq. (4) |
|                   | $0.25 \times 0.25$           | Basin                        |
|                   | $0.1\,^\circ$C bins          | $0.1\,^\circ$C bins          |
| MDP use           | Infer MPI                    | Cap basin pres. drop         |
| MPI definition    | MPI $= P^{\mathrm{env}}$ – MPD | Holland (1997)              |
| Unrealistic values | Bister and Emanuel (2002)   | MPD                          |

**Table A4.** Mean of the maximum wind speed of synthetic cyclones in representative representation pathways ($\mathrm{m\,s^{-1}}$).

|                               | RCP26 | RCP45 | RCP85 |
| ----------------------------- | ----- | ----- | ----- |
| ACCESS1-0 (BoM-CSIRO, Australia) |       | 66.42 | 70.21 |
| bcc-csm1-1-m (BCC, China)     | 65.65 | 71.00 | 71.91 |
| CanESM2 (CCCMA, Canada)       | 59.22 |       | 66.70 |
| GISS-E2-H (NASA, USA)         |       | 71.79 | 75.02 |
| inmcm4 (INM, Russia)          |       | 61.42 | 64.70 |
| IPSL-CM5A-MR (IPSL, France)   | 60.70 | 63.89 | 69.61 |
| NorESM1-ME (NCC, Norway)      | 57.40 | 59.02 | 63.32 |
| Mean                          | 60.74 | 65.59 | 68.78 |

**Table A5.** Mapping countries to IIASA regions.

| Admin | Region | Admin | Region |
|---|---|---|---|
| Mexico | R32MEX | Australia | R32ANUZ |
| United States of America | R32USA | Madagascar | R32SSA-L |
| Canada | R32CAN | Mozambique | R32SSA-L |
| Cuba | R32LAM-M | South Africa | R32SSA-L |
| Haiti | R32LAM-L | United Republic of Tanzania | R32SSA-L |
| Dominican Republic | R32LAM-M | Indonesia | R32IDN |
| Mauritania | R32LAM-M | Mauritius | R32SSA-M |
| Nicaragua | R32LAM-L | France | R32EU15 |
| Guatemala | R32LAM-L | Malawi | R32SSA-L |
| Belize | R32LAM-L | Zimbabwe | R32SSA-L |
| Venezuela | R32LAM-M | Swaziland | R32SSA-L |
| Honduras | R32LAM-L | Lesotho | R32SSA-L |
| Jamaica | R32LAM-M | Zambia | R32SSA-L |
| Puerto Rico | R32LAM-M | Comoros | R32SSA-L |
| Costa Rica | R32LAM-M | New Zealand | R32ANUZ |
| The Bahamas | R32LAM-M | Vanuatu | R32OAS-L |
| El Salvador | R32LAM-M | Fiji | R32OAS-L |
| Panama | R32LAM-M | Solomon Islands | R32OAS-L |
| Colombia | R32LAM-M | New Caledonia | R32EU15 |
| Grenada | R32LAM-M | French Polynesia | R32EU15 |
| Antigua and Barbuda | R32LAM-M | Papua New Guinea | R32OAS-L |
| Barbados | R32LAM-M | Wallis and Futuna | R32EU15 |
| Cabo Verde | R32SSA-L | Cambodia | R32OAS-CPA |
| India | R32IND | Thailand | R32OAS-L |
| Pakistan | R32PAK | Laos | R32OAS-CPA |
| Bangladesh | R32OAS-L | Vietnam | R32OAS-CPA |
| Oman | R32MEA-H | South Korea | R32KOR |
| United Arab Emirates | R32MEA-H | Japan | R32JPN |
| Iran | R32MEA-M | Russia | R32RUS |
| Myanmar | R32OAS-L | Philippines | R32OAS-L |
| Nepal | R32OAS-L | Taiwan | R32TWN |
| Yemen | R32MEA-M | North Korea | R32OAS-L |
| Saudi Arabia | R32MEA-H | United States of America | R32USA |
| Qatar | R32MEA-H | Mongolia | R32OAS-CPA |
| Kuwait | R32MEA-H | Hong Kong S.A.R. | R32CHN |
| Iraq | R32MEA-M | Federated States of Micronesia | R32OAS-L |
| Sri Lanka | R32OAS-M | Macao S.A.R. | R32CHN |
| Afghanistan | R32PAK | Guam | R32OAS-M |
| China | R32CHN | Malaysia | R32OAS-M |

**Table A6.** Examples of countries' damage statistics in USD billion ($\alpha_1 = 1$, $\alpha_2 = 1$).

| | SSP | Config | Mean | Std error | per50 | per66 | per75 | per95 |
|---|---|---|---|---|---|---|---|---|
| | | RCP26 | 13.19 | 2.46 | 0.01 | 0.24 | 1.26 | 41.36 |
| | SSP2 | RCP45 | 22.88 | 3.10 | 0.07 | 0.88 | 3.35 | 99.62 |
| | | RCP85 | 24.57 | 3.51 | 0.06 | 0.95 | 3.59 | 85.48 |
| | | RCP26 | 6.37 | 1.21 | 0.01 | 0.11 | 0.59 | 19.47 |
| | SSP3 | RCP45 | 10.85 | 1.47 | 0.04 | 0.40 | 1.46 | 46.30 |
| AUS | | RCP85 | 11.55 | 1.68 | 0.03 | 0.44 | 1.55 | 36.53 |
| | | RCP26 | 11.79 | 2.21 | 0.01 | 0.21 | 1.12 | 36.01 |
| | SSP4 | RCP45 | 20.27 | 2.74 | 0.06 | 0.77 | 2.83 | 87.33 |
| | | RCP85 | 21.71 | 3.12 | 0.05 | 0.85 | 3.11 | 73.63 |
| | | RCP26 | 25.14 | 4.66 | 0.02 | 0.49 | 2.62 | 79.28 |
| | SSP5 | RCP45 | 44.76 | 6.17 | 0.15 | 1.83 | 7.12 | 202.33 |
| | | RCP85 | 48.46 | 6.82 | 0.13 | 2.04 | 7.42 | 170.75 |
| | | RCP26 | 8.67 | 0.73 | 1.42 | 3.19 | 5.70 | 46.16 |
| | SSP2 | RCP45 | 10.12 | 0.56 | 2.32 | 6.08 | 10.25 | 46.55 |
| | | RCP85 | 13.14 | 0.79 | 3.06 | 7.52 | 12.48 | 64.57 |
| | | RCP26 | 7.31 | 0.60 | 1.27 | 2.87 | 4.71 | 39.87 |
| | SSP3 | RCP45 | 8.68 | 0.48 | 2.10 | 5.40 | 9.06 | 39.45 |
| CHN | | RCP85 | 11.26 | 0.67 | 2.82 | 6.77 | 11.17 | 53.47 |
| | | RCP26 | 5.15 | 0.44 | 0.81 | 1.84 | 3.17 | 26.53 |
| | SSP4 | RCP45 | 5.84 | 0.33 | 1.29 | 3.43 | 5.71 | 27.61 |
| | | RCP85 | 7.56 | 0.46 | 1.67 | 4.35 | 7.05 | 38.88 |
| | | RCP26 | 13.48 | 1.16 | 2.07 | 4.84 | 8.52 | 73.92 |
| | SSP5 | RCP45 | 15.75 | 0.89 | 3.46 | 9.17 | 15.86 | 72.24 |
| | | RCP85 | 20.51 | 1.26 | 4.63 | 11.28 | 19.06 | 100.22 |
| | | RCP26 | 2.40 | 0.22 | 0.28 | 0.87 | 1.48 | 12.27 |
| | SSP2 | RCP45 | 2.87 | 0.23 | 0.33 | 1.03 | 2.01 | 14.08 |
| | | RCP85 | 3.78 | 0.34 | 0.54 | 1.44 | 2.80 | 18.06 |
| | | RCP26 | 1.38 | 0.12 | 0.19 | 0.57 | 0.91 | 7.08 |
| | SSP3 | RCP45 | 1.63 | 0.13 | 0.23 | 0.66 | 1.22 | 7.77 |
| JPN | | RCP85 | 2.16 | 0.18 | 0.37 | 0.96 | 1.70 | 10.13 |
| | | RCP26 | 2.10 | 0.19 | 0.24 | 0.76 | 1.29 | 10.54 |
| | SSP4 | RCP45 | 2.50 | 0.21 | 0.29 | 0.90 | 1.70 | 12.36 |
| | | RCP85 | 3.28 | 0.29 | 0.47 | 1.24 | 2.37 | 15.52 |
| | | RCP26 | 4.57 | 0.43 | 0.51 | 1.63 | 2.85 | 21.28 |
| | SSP5 | RCP45 | 5.55 | 0.44 | 0.64 | 1.99 | 4.02 | 27.04 |
| | | RCP85 | 7.32 | 0.68 | 1.02 | 2.87 | 5.49 | 33.72 |
| | | RCP26 | 9.18 | 0.91 | 0.52 | 2.41 | 5.26 | 50.38 |
| | SSP2 | RCP45 | 12.11 | 0.92 | 1.19 | 4.30 | 8.29 | 71.05 |
| | | RCP85 | 15.54 | 1.15 | 2.06 | 6.47 | 11.93 | 80.18 |
| | | RCP26 | 9.63 | 0.96 | 0.56 | 2.68 | 5.68 | 53.54 |
| | SSP3 | RCP45 | 12.94 | 0.98 | 1.32 | 4.74 | 9.07 | 72.48 |
| MEX | | RCP85 | 16.65 | 1.23 | 2.25 | 7.10 | 12.32 | 87.09 |
| | | RCP26 | 4.76 | 0.48 | 0.27 | 1.23 | 2.67 | 28.14 |
| | SSP4 | RCP45 | 6.11 | 0.46 | 0.62 | 2.04 | 4.29 | 34.82 |
| | | RCP85 | 7.72 | 0.59 | 1.00 | 3.07 | 5.77 | 38.43 |
| | | RCP26 | 10.91 | 1.09 | 0.56 | 2.66 | 5.86 | 61.01 |
| | SSP5 | RCP45 | 14.08 | 1.09 | 1.30 | 4.69 | 9.41 | 81.86 |
| | | RCP85 | 18.09 | 1.36 | 2.24 | 7.07 | 13.52 | 93.66 |
| | | RCP26 | 131.80 | 13.76 | 17.99 | 51.49 | 88.19 | 588.12 |
| | SSP2 | RCP45 | 150.20 | 10.45 | 26.06 | 64.81 | 120.59 | 747.43 |
| | | RCP85 | 210.89 | 14.22 | 39.68 | 93.71 | 176.39 | 1 078.51 |
| | | RCP26 | 82.27 | 8.83 | 10.46 | 31.31 | 52.37 | 372.94 |
| | SSP3 | RCP45 | 92.22 | 6.61 | 14.40 | 39.85 | 72.11 | 455.65 |
| USA | | RCP85 | 127.71 | 8.73 | 22.04 | 53.80 | 101.54 | 661.06 |
| | | RCP26 | 115.13 | 12.16 | 15.28 | 44.81 | 76.52 | 512.94 |
| | SSP4 | RCP45 | 130.25 | 9.18 | 21.91 | 56.20 | 103.88 | 649.04 |
| | | RCP85 | 181.86 | 12.32 | 32.58 | 79.12 | 148.01 | 944.76 |
| | | RCP26 | 258.84 | 26.45 | 38.70 | 102.77 | 183.70 | 1 085.39 |
| | SSP5 | RCP45 | 301.50 | 20.48 | 51.24 | 133.52 | 249.84 | 1 450.39 |
| | | RCP85 | 430.81 | 29.13 | 81.24 | 197.55 | 386.07 | 2 107.37 |

**Table A7.** Simple credit spread model.

| | \multicolumn{4}{c}{Dependent variable} |
|---|---|---|---|---|
| | \multicolumn{4}{c}{OAS (bp)} |
| | (1) | (2) | (3) | (4) |
| $\Delta \mathcal{C}_T$ | | | $-486.681^{***}$ | $-467.219^{***}$ |
| | | | (114.675) | (91.876) |
| Duration | | | | $8.639^{***}$ |
| | | | | (1.659) |
| VIX | | | $17.094^{***}$ | $15.377^{***}$ |
| | | | (2.510) | (2.120) |
| TED | | | $-13.753$ | $-7.395$ |
| | | | (113.318) | (91.791) |
| $r_{US,10Y}$ | | | $-9.809^{***}$ | $-8.478^{***}$ |
| | | | (2.199) | (1.756) |
| $\dfrac{L}{GDP}$ | $367.461^{***}$ | $229.629^{***}$ | $336.718^{***}$ | $377.035^{***}$ |
| | (26.060) | (21.319) | (25.809) | (79.759) |
| $\dfrac{Reserves}{GDP}$ | | | $-244.949^{***}$ | $-1176.650^{***}$ |
| | | | (41.034) | (185.851) |
| Rating $< B-$ | | $1692.843^{***}$ | $1697.096^{***}$ | $1391.009^{***}$ |
| | | (42.121) | (41.011) | (36.858) |
| Countries | | | | $X$ |
| Constant | $207.254^{***}$ | $222.600^{***}$ | $1,410.488^{***}$ | $1521.181^{***}$ |
| | (15.583) | (12.732) | (330.158) | (269.092) |
| Observations | 2212 | 1860 | 1832 | 1832 |
| $R^2$ | 0.083 | 0.509 | 0.542 | 0.723 |
| Adjusted $R^2$ | 0.082 | 0.509 | 0.541 | 0.715 |
| Residual std error | 411.571 (df = 2210) | 316.591 (df = 1857) | 308.258 (df = 1824) | 242.855 (df = 1778) |
| $F$ statistic | 198.832$^{***}$ (df = 1; 2210) | 962.874$^{***}$ (df = 2; 1857) | 308.792$^{***}$ (df = 7; 1824) | 87.609$^{***}$ (df = 53; 1778) |

Note: $^{*}p < 0.1$; $^{**}p < 0.05$; $^{***}p < 0.01$. Dropping Argentina in the constant in (4).

*Code and data availability.* Code is available from the E4C data hub at https://www.e4c.ip-paris.fr/#/fr/datahub/projects (last access: 14 October 2022) and Zenodo at https://doi.org/10.5281/zenodo.5645516 (Le Guenedal et al., 2021). The deposit includes R scripts with functions, a fitting script, the parameters of the fitting of all the relationships on ERA5, historical and future synthetic tracks, and a sample of simulated annual damage per country and scenario. We included a user guide for data exploration and a quick-start guide.

*Author contributions.* The methodology was conceptualized and established by PD, TLG and PT. The data curation, software development and visualization were realized by TLG. The original draft was written by TLG and PT, and the revised version, including review and editing, by PD, TLG and PT. The project was supervised by PD and PT.

*Competing interests.* The contact author has declared that none of the authors has any competing interests.

*Acknowledgements.* The authors are very grateful to Florian Raymond, Thierry Roncalli, Takaya Sekine and Lauren Stagnol for helpful comments.

*Financial support.* This work contributes to the Energy4Climate Interdisciplinary Center (E4C) of the Institut Polytechnique de Paris and the Ecole des Ponts ParisTech, supported by the 3rd Programme d'Investissements d'Avenir (ANR-18-EUR-0006-02). Peter Tankov was financially supported by the FIME Research Initiative. This work was also supported by Amundi Asset Management.

*Review statement.* This paper was edited by Jinkyu Hong and reviewed by three anonymous referees.

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

**Remarks from the typesetter**

TS1 Please give an explanation of why this needs to be changed. We have to ask the handling editor for approval. Thanks.