# Peer review of "Cyclone generation Algorithm including a THERmodynamic module for Integrated National damage Assessment (CATHERINA 1.0) compatible with CMIP climate data"

_Geoscientific Model Development, 2021_

## Author Comment (AC1)

**Answers to Referee 1: Technical details**

January 2022

**Q1 - Cyclone description accuracy**  It is not clear that this tropical cyclone model leads to accurate forecasts of storms. Changes in wind patterns can have no effect in this model. Is the cyclone model in this paper as accurate as the models developed by Emanuel? Or is this model a step backwards?

The model is based on the theory of Emanuel (1988), which proposes to express the maximum potential intensity (i.e. the minimum central pressure achievable given local climate conditions) as:

$$\text{MPI} = P_{env} \cdot \exp(-X) \tag{1}$$

To demonstrate the importance of introducing the additional variables in the definition of $X$, we reiterate the principle of the demonstration. Let $x = \exp(-X)$ define the depression ratio: $x = \frac{P_c}{P_{env}}$. The central pressure equation was shown in Emanuel (1988) to be expressed in the form:

$$\ln(x) = A\left[\frac{1}{x} - B\right]$$

Noting $\mathcal{E} = \dfrac{\text{SST} - T_{tropo}}{\text{SST}}$ the thermodynamic efficiency, Emanuel (1988, p. 1145) writes:

$$A = \frac{\mathcal{E}}{1 - \mathcal{E}} \frac{L_v}{R_v T_{\text{tropo}}} \frac{e_s}{P_{env}} \times \tag{2}$$

$$\left[ 1 - \underbrace{\frac{g\bar{z_0}^\star}{\mathcal{E}L_v}}_{(a)} - \underbrace{\frac{1}{8}\frac{f^2\bar{r_1^2}^\star}{\mathcal{E}L_v}}_{(b)} + \underbrace{\frac{C_l T_{\text{tropo}}}{\mathcal{E}L_v}\left(\mathcal{E} + (1 - \mathcal{E})\ln(1 - \mathcal{E})\right)}_{(c)} \right] \tag{3}$$

where $L_v$ is the latent heat of vaporization, $R_v$ the gas constant of water vapor, $C_l$ is the heat capacity of liquid water where (a) (b) and (c) are functions of the thermodynamic efficiency (so SST and tropopause temperature) and:

$$B = \text{RH}\left[ 1 + \frac{e_s}{P_{env}}\frac{\ln(\text{RH})}{A} \right] + \frac{1}{4}\underbrace{\frac{f^2 r_{env}^2\left(1 + \text{RH}\frac{e_s}{P_{env}}\right)}{R_d T_{\text{tropo}}(1 - \mathcal{E})A}}_{(d)} \tag{4}$$

where $r_{env}$ denotes the outer radius (fixed at 500 km in CATHERINA), and (d) is a function of all the raw variables required in our model. This theoretical expression can be – and has been – simplified. For instance, in the expression of A, the term (a) refers to the potential energy used to lift-up water substance, this factor is inversely proportional to the thermodynamic efficiency $\mathcal{E}$. The second factor (b) represents the energy require for radial translation of water substance (this term is neglected in Emanuel (1988, p. 1146)). The last factor (c) is the contribution of water substance to heat capacity.

Following this formulation of (1) and integrating additional simplifications proposed in subsequent papers Emanuel (1991), we finally use the framework summarized in Holland (1997) to define locally the MPI in such a way that it provides a statistically significant instrumental variable in the description of tropical cyclone intensification, which is the aim of the algorithm.

**Q3 - Damage function calibration**   The model depends a great deal on the damage function. But it is not clear how this damage function was estimated.

The optimization is performed by Eberenz et al. (2021) to define the regional damage functions. Eberenz et al. (2021) first define the event damage ratio (EDR) as a fraction error between normalized reported (NRD) and simulations (SED) for each cyclone ($\text{EDR}(i) = \frac{\text{SED}(i, v_h(j))}{\text{NRD}(i)}$) and the total damage ratio (TDR) is defined in each region summing over events: ($\text{TDR}(j) = \frac{\sum_i \text{SED}(i, v_h(j))}{\sum_i \text{NRD}(i)}$) For each event, there is a value for $v_h$ allowing to optimally calibrate the explicit damage function described in Emanuel (2011).

Eberenz et al. (2021) propose two complementary optimization methodologies to find the value of $v_h^\star$ maximizing the prediction of the regional damages Eberenz et al. (2021):

($i$) Root mean square fraction (RMSF), minimizing the spread of the event damage ratios (EDR) – defined as the ratio of simulated damage vs. reported damage:

$$v_h{_{\text{RMSF}}^\star}(j) = \text{argmin}_j \exp\left(\sqrt{\frac{1}{N}\sum\left(\ln(\text{EDR(i)})\right)^2}\right) \tag{5}$$

($ii$) Total damage ratio (TDR), finding the value of $v_h^\star$, such as the ratio of total simulated damage – obtained summing over event damage – and total reported damage tends to 1:

$$v_h{_{\text{TDR}}^\star}(j) = \text{argmin}_j |\text{TDR}(j) - 1| \tag{6}$$

and the optimal parameters are provided in the paper. We chose not to include the formulation of these optimization problems in the prior manuscript because we did not compute the optimal values and used those proposed by Eberenz et al. (2021). However, we will clarify the calibration in the revised version.

**Q5/6 - Spatial distribution/National asset dynamics** The model appears to assume the spatial distribution of assets are fixed within a country.The paper does allow national assets to change over time, but they do not describe how this is done.

To estimate future exposures along the cyclone track in each scenario, we use the downscaled estimation for the exposed wealth (Eberenz et al., 2020) and the coefficients representing the change between the current state and the future scenario in the framework of the shared-socioeconmic pathwways (Jones & O'Neill, 2020; O'Neill et al., 2017; O'Neill et al., 2014).

The local physical exposure at the coordinates $(x, y)$ at time $t$ in a region $j$ in scenario $k$ is defined as follows:

$$\Phi(x, y, j, k, t) \quad = \quad \underbrace{F_{\text{GDP}}^{cap}(j, k, t)}_{\text{Global macro factor}} \cdot \underbrace{F_{pop}(x, y, k, t) \cdot \mathcal{L}_P(x, y)}_{\text{Local factor}}. \tag{7}$$

where $\mathcal{L}_P(x, y)$ is the data from Eberenz et al. (2020) (discussed in question 4). The factor $F_{\text{GDP}}^{cap}$ is the projected GDP per capita growth for each region:

$$F_{\text{GDP}}^{cap}(j, k, t) \quad = \quad \frac{\text{GDP}(j, k, t)/\text{GDP}(j, t = 2020)}{P(j, k, t)/P(j, t = 2020)} \tag{8}$$

where $P$ is the total population of the region retrieved from SSP database (Riahi et al., 2017)[1]. We use the most granular projections of GDP per capita variation curves i.e. the projection for each region. The factor $F_{pop}$ is defined as follows:

$$F_{pop}(x, y, k, t) \quad = \quad \frac{p(x, y, k, t)}{p(x, y, t = 2020)} \tag{9}$$

where $p(x, y, k, t)$ represents the local projections of population Jones and O'Neill (2020).
* * *
[1] https://tntcat.iiasa.ac.at/SspDb/.

Figure 1: Variation of population exposure in 2100

(a) The middle road (SSP2)             (b) The rocky road (SSP3)

(c) Inequality (SSP4)                     (d) Highway (SSP5)

[Figure]

The scenario-based population grid generation is detailed by Jones and O'Neill (2020) with a last version downscaled at 1km following Gao (2020). This population grid is available every 10 years. We use the closest value in the definition of the exposure.

Figure 2: Regional $F_{cap}$ factor variation in SSPs IIASA database

Source : https://tntcat.iiasa.ac.at/SspDb/.

**Q9 - Results: quantiles vs. expected** Figure 19 suggests the model predicts a small probability of very large damage but an expected value that is quite small. What explains this large tail to the distribution of damage? Is this simply the probability of a large storm striking a large coastal city? What is the expected value of damage?

Table 1: Simulated expected annual damage (USD Billions)

| SSP | RCP | Mean damage USD Bn |
|---|---|---|
| Historical | Historical | 51,34 |
| SSP2 | RCP26 | 65,36 |
| SSP2 | RCP45 | 89,80 |
| SSP2 | RCP85 | 118,43 |
| SSP3 | RCP26 | 45,74 |
| SSP3 | RCP45 | 63,43 |
| SSP3 | RCP85 | 84,13 |
| SSP4 | RCP26 | 44,24 |
| SSP4 | RCP45 | 60,22 |
| SSP4 | RCP85 | 82,02 |
| SSP5 | RCP26 | 113,09 |
| SSP5 | RCP45 | 155,88 |
| SSP5 | RCP85 | 208,46 |

**References**

Eberenz, S., Lüthi, S., & Bresch, D. N. (2021). Regional tropical cyclone impact functions for globally consistent risk assessments. *Natural Hazards and Earth System Sciences*, *21*(1), 393–415.

Eberenz, S., Stocker, D., Röösli, T., & Bresch, D. N. (2020). Asset exposure data for global physical risk assessment. *Earth Syst*, *12*. https://doi.org/10.5194/essd-12-817-2020

Emanuel, K. A. (1988). The maximum intensity of hurricanes. *Journal of the Atmospheric Sciences*, *45*(7), 1143–1155.

Emanuel, K. A. (1991). The theory of hurricanes. *Annual Review of Fluid Mechanics*, *23*(1), 179–196.

Emanuel, K. A. (2011). Global warming effects on us hurricane damage. *Weather, Climate, and Society*, *3*(4), 261–268.

Gao, J. (2020). Downscaling global spatial population projections from 1/8-degree to 1-km grid cells. *Technical Notes NCAR, National Center for Atmospheric Researcher, Boulder, CO., USA*. https://doi.org/10.7927/q7z9-9r69

Holland, G. J. (1997). The maximum potential intensity of tropical cyclones. *Journal of the atmospheric sciences*, *54*(21), 2519–2541.

Jones, B., & O'Neill, B. C. (2020). Global one-eighth degree population base year and projection grids based on the shared socioeconomic pathways. *Palisades*, (Revision 01).

O'Neill, B. C., Kriegler, E., Ebi, K. L., Kemp-Benedict, E., Riahi, K., Rothman, D. S., van Ruijven, B. J., van Vuuren, D. P., Birkmann, J., Kok, K., et al. (2017). The roads ahead: Narratives for shared socioeconomic pathways describing world futures in the 21st century. *Global environmental change*, *42*, 169–180.

O'Neill, B. C., Kriegler, E., Riahi, K., Ebi, K. L., Hallegatte, S., Carter, T. R., Mathur, R., & van Vuuren, D. P. (2014). A new scenario framework for climate change research: The concept of shared socioeconomic pathways. *Climatic change*, *122*(3), 387–400.

Riahi, K., van Vuuren, D. P., Kriegler, E., Edmonds, J., O'Neill, B. C., Fujimori, S., Bauer, N., Calvin, K., Dellink, R., Fricko, O., Lutz, W., Popp, A., Cuaresma, J. C., KC, S., Leimbach, M., Jiang, L., Kram, T., Rao, S., Emmerling, J., . . . Tavoni, M. (2017). The shared socioeconomic pathways and their energy, land use, and greenhouse gas emissions implications: An overview. *Global Environmental Change*, *42*, 153–168. https://doi.org/10.1016/j.gloenvcha.2016.05.009

---

## Author Comment (AC2)

**Answers to Referee 2**

April 15, 2022

**1 Detailed comments**

**CGMs** Table 1: the selection of GCMs used should be justified. This could be through reference to model performance literature for key parameters, a specific evaluation process or perhaps simply availability of required variables for the analysis (though I note the variables used are available for all CMIP5 models).

The choice of the CGMs was driven by the availability of the variables of interest in the Copernicus Climate data store (CDS) in the representative concentration pathways used in the exercise (RCP 2.6, 4.5 and 8.5 W/m2) in both single and pressure levels monthly data in the same ensemble (r1i1p1). We also aimed at having multiple regions represented.

**Shifted extraction** Section 2.2: the MSLP from ERA-5 is sampled 500 km from the centre of the cyclone. Is the same done for the other variables? Since the data are sampled from monthly means,it's possible the sampled values may not accurately represent the conditions at the time of TC passage (especially relevant for variables with sharp gradients such as SST).

We retrieve both pressure (MSLP) and humidity (RH) to define $P^{env}$ and $RH^{env}$ (Holland, 1997) away from the center because TC maximum potential intensity (MPI) - through thermodynamic efficiency and moist entropy - arise from the deviations from the normal conditions. We acknowledge that monthly averaging may indeed "smooth" values so that the data may not represent the conditions at the time of cyclone passage. Therefore, using monthly means, this translation is mainly made for reasons of theoretical coherence. In future studies, this model will be applied with higher temporal resolution and performing this translation would be more important. In the present version of our paper, because the CMIP5 projections of the sea-level temperature were only available at monthly frequency in the CDS, we chose to perform the exercise using monthly data to illustrate our approach. In addition the monthly sampled data allowed us to build a statistically significant description of the MPI in the historical period. The possibility of improving the model using high frequency data will be emphasized in the revised version of the manuscript.

**Common metrics** Section 3.2: Given the literature of TC track generation methods, comparison with common metrics is encouraged. Specifically, as landfall is critical to reliable performance of a damage model, it would be helpful to present a comparison of the observed and simulated landfall rates (see for example Arthur (2021), Hall and Jewson (2007), and Lee et al. (2016)). This would strengthen the quality of the track generation results significantly.

In the revised version we will compute landfall rates and compare them to relevant results from the literature.

**WPR** Eq 3 - note that most best track data used wind pressure relations (WPRs) to determine Pc. Typically the work flow involves determining the Dvorak T number, converting this to a sustained wind speed, followed by regionally-specific WPR to determine Pc. The conversion back to wind speed from reported Pc using a single WPR will introduce errors, as an array of WPRs are used to operationally estimate Pc, not only between basins but within basins as well (J. Courtney, 2009; J. Courtney & Burton, 2018; J. B. Courtney et al., 2021; Harper, 2002).

We acknowledge that the use of a single WPR (in line with Bloemendaal et al. (2020)) introduces errors. Figure 1, presents the WPR estimated from historical track data in different basins, and shows that when using a single WPR the winds are likely to be underestimated in the East Pacific and north Indian basin.In the revised version we shall estimate WPR parameters separately for each basin, include the references provided by the reviewer and update Figure 9 consequently.

Figure 1: WPR per basin

**Uses of MDP** Eq 10 describes the dominant control on the maximum intensity of TCs (maximum pressure drop - MDP). This is tied only to SSTs. The model uses maximum potential intensity (MPI) to control the depression dynamics (i.e. intensification rates). The formulation of MPI is directly applicable to the problem of estimating the maximum intensity, accounting for factors beyond SST alone that control maximum intensity. This suggests using SST as the only predictor of the MDP is deficient.

Indeed, we already acknowledge that the SST alone in not a good predictor of whether individual TC will intensify. Therefore, we use the thermodynamic definition in the cyclone dynamics specification. On the other hand, we still define a "MPD" taking the maximum observed pressure drop for a given SST across all events in each basin. You are right to point out that this appears to be inconsistent. However, we use this maximum depression (MPD) estimated over the historical period for a given sea surface temperature only to cap the depressions in the simulations, to avoid generating events intensifying beyond past observations and make the simulated tracks more realistic. This is a limitation of our approach, however this is relatively common in "statistical" models. Alternatively, we could make the maximum depression depend on the four variables of interest, however, this would make estimation more difficult and reduce the significance of this statistic.

In the revised version we will compare our definition with an alternative definition of the MPD, using the thermodynamic definition used for the MPI, and substituting extreme values of temperature and humidity.

**Dynamics factor**   Further,Chen et al. (2021) suggest rapid intensification is dependent on dynamical (e.g. upper divergence and wind shear) as well as thermodynamical factors. While the difference between Pc and MPI is a factor in predicting rapid intensification, and the dynamical factors are probably accounted for by the random innovation (Eq. 12), these other dynamical factors should be acknowledged.

Indeed, the components explaining the noise term in the pressure dynamics should be better identified, and we will acknowledge them in the next version of the manuscript. However, in the context of our exercise we had to focus on explanatory factors that are available in the CMIP5 simulations which reduced our scope to thermodynamical factors.

**CDF-t**   Apply CDF-t to model variables, then evaluate MPI - I suggest comparing quantiles of ERA5 MPI against the bias corrected CMIP MPI values to demonstrate the effect of bias correction. Q-Q plots would be an effective way to do this. One risk with this approach is that correcting individual variables may lead to unrealistic combinations when evaluating MPI - e.g. extremely low tropopause temperatures in combination with very high SSTs that lead to unrealistic lapse rates and therefore unrealistically large MPI. Two solutions present themselves: 1) apply the bias correction methods to calculated MPI or (2) consider the joint distributions of variables when evaluating the bias corrections.

This is a very relevant point. Indeed, individual variables entering the MPI computation may be strongly correlated as shown in Figure 2. In the revised version we will follow the reviewer's suggestion and apply bias correction directly to calculated MPI.

**SST distributions**   The distributions of SST presented in Figure 16 do not appear representative of SSTs sampled in the vicinity of TCs, and is inconsistent

Figure 2: Correlation levels (global) of modeled variables affecting the MPI with their reference values in ERA5

[Figure]

with the distribution shown in Figure 10. SSTs of 26C (299K) are typically considered a lower bound for TC formation (Gray, 1979), but median values from the ERA5 are well below that - for example based on Figure 16 the median SST for the South Pacific basin along synthetic tracks is 290-292K, for the Western Pacific 295K. Only the N Indian basin has a median SST near 300K. This suggests that the synthetic tracks are traversing areas not typically covered by TCs, or occurring at the wrong time of year for the respective basin leading to the unusual SST distribution.

The bias-correction module is indeed fitted on a larger range of climate conditions. By definition, for the genesis of the cyclones, the time of year and location are in line with historical cyclone data. However, in the bias-correction module, the synthetic tracks are generated without climate constraints, i.e. cyclones are allowed to drift relatively far away from their genesis location (in the limits of their initial basin), and therefore can cover conditions which do not lead to the formation of tropical cyclones. At this stage, these tracks are not to be considered as 'TCs tracks' but as 'candidate' tracks. In the following stage, TC tracks will be generated from candidate tracks by filtering those ones where meteorologiocal conditions for cyclone formation are satisfied.

**Discussion**    Completely absent is any discussion on TC rates in the projections. Comprehensive literature reviews and expert elicitations indicate a global decline in TC frequency (albeit with generally low-medium confidence) (Knutson et al., 2020). Changes in TC rates will have a significant impact on the annualised losses. This is an important component that should be addressed. In parallel, there is no discussion on changes in track behaviour. Observed trends

in TC translation speed (Kossin, 2018) and poleward migration of maximum intensity (Kossin et al., 2014) should be considered in projections of TC activity. This has profound implications for TC-related risk in key marginal areas (Bruyère et al., 2019) where vulnerabilities are high, but present-day frequency of TCs is low.

These two comments will be included in the conclusion of the revised version of our paper as they reflect important limitations of our exercise. Indeed, we kept the genesis rates constant for each basin. The number of cyclones each year are drawn from Poisson distribution. It is possible to reduce the intensity parameter in the projections, and to introduce cyclones in regions where the present-day frequency is low, however, in this study, we focused on the changes in thermodynamic potentials. Moreover, as our approach is a statistical one, we had to focus on areas where relationships could be extrapolated from historical data. We will add a comment to account for this possible improvement.

**Shared Socioeconomic Pathways**   Section 5.2: Consideration of SSPs in determining the effects on damage is novel, but the explanation is very limited. Given growth of exposure is constrained in existing high exposure regions, regional growth may not be in areas exposed to TC impacts.

In the revised version, we will provide more explanations about the shared-socioeconomic pathways used to project exposure. Indeed, we did not consider that areas subject to cyclones would face additional economic growth constraints in our projections. Historically, high exposure regions were not particularly constrained in terms of growth (e.g. the East Coast of the United States of America can be considered as a high exposure region as well as most regions in South Korea, Japan, Australia). In addition, climate change increases tropical cyclone intensity allowing them to reach regions where current TC impacts are low.

**Description projection exposure**   The description of the implementation of projections of local physical asset value dynamics is very limited, but probably the most novel part of the connected modelling system. There should be a more substantial discussion on how the SSP definitions are used to modify asset values.

To estimate future exposures along the cyclone track in each scenario, we use the downscaled estimation for the exposed wealth (Eberenz et al., 2020) and the coefficients representing the change between the current state and the future scenario in the framework of the shared-socioeconmic pathways (Jones & O'Neill, 2020; O'Neill et al., 2017; O'Neill et al., 2014).

The local physical exposure at the coordinates $(x, y)$ at time $t$ in a region $j$ in scenario $k$ is defined as follows:

$$\Phi(x, y, j, k, t) \quad = \quad \underbrace{F_{\text{GDP}}^{cap}(j, k, t)}_{\text{Global macro factor}} \cdot \underbrace{F_{pop}(x, y, k, t) \cdot \mathcal{L}_P(x, y)}_{\text{Local factor}}. \qquad (1)$$

where $\mathcal{L}_P(x, y)$ is the data from Eberenz et al. (2020). The factor $F_{\text{GDP}}^{cap}$ is the projected GDP per capita growth for each region:

$$F_{\text{GDP}}^{cap}(j, k, t) \quad = \quad \frac{\text{GDP}(j, k, t)/\text{GDP}(j, t = 2020)}{P(j, k, t)/P(j, t = 2020)} \tag{2}$$

where $P$ is the total population of the region retrieved from SSP database (Riahi et al., 2017)[1]. Figure 3 displays the scenario-based projections of GDP and population in the five SSP, at the regional level by the IIASA model.

Figure 3: Regional $F_{cap}$ factor variation in SSPs IIASA database

Source : https://tntcat.iiasa.ac.at/SspDb/.

The factor $F_{pop}$ is defined as follows:

$$F_{pop}(x, y, k, t) \quad = \quad \frac{p(x, y, k, t)}{p(x, y, t = 2020)} \tag{3}$$

where $p(x, y, k, t)$ represents the local projections of population Jones and O'Neill (2020) illustrated Figure 4. We will detail this methodology in the manuscript.
* * *
[1] https://tntcat.iiasa.ac.at/SspDb/.

Figure 4: Variation of population exposure in 2100

(a) The middle road (SSP2)          (b) The rocky road (SSP3)

[Figure]

(c) Inequality (SSP4)               (d) Highway (SSP5)

The scenario-based population grid generation is detailed by Jones and O'Neill (2020) with a last version downscaled at 1km following Gao (2020). This population grid is available every 10 years. We use the closest value in the definition of the exposure.

**2   Technical comments**

- Page 4, footnote 2: Please use the full reference for the Copernicus Climate data store (Hersbach et al., 2020)

- done

- Figures need to be larger to be legible.

- Figure 3: Recommend plotting each track with the same vertical scale (on the pressure and wind axes respectively) - i.e. use a scale of 0-75 m/s for wind speed and 880 - 1025 hPa for pressure on all panels. This will aid intercomparison of the time histories

- We will review all Figures in the next version

- Line 135: Please include an equation label

- done

- Page 7 - footnote: References to World Bank (2019b) and Credit Suisse Research Institute (2017) in the footnote of page 7 are not included in the bibliography.

- done

- Page 11: Footnote 13 should be in the body of the manuscript, as this is a key difference between James and Mason (2005) and the implementation in the current study. Following on from this, Tables A2, A4 and A6 reflect basin-wide fits, while the tracking method uses 5-by-5 degree grid. It may be more appropriate to provide maps of the relevant coefficients on the grids in the Appendix rather than the tables.

  - done for the footnote and we replace table by relevant maps 5 to illustrate the specification.

Figure 5: Example of maps of coefficient and statistics

- Page 17, line 289: Does "This study" refer to the current manuscript, or to the previously referenced Unawa et al. (2000). Please clarify.

- done

- Page 20, line 333: Change "non-EOCD" to "non-OECD"

- Page 24, line 376: suggest changing "unbiased" to "bias-corrected"

- done

- Figure A8: Add units to the horizontal axes of the plot, or indicate what the values are in the caption. Ideally, each of the sub-plots should also use the same horizontal scale to aid comparison

**References**

Arthur, W. C. (2021). A statistical–parametric model of tropical cyclones for hazard assessment. *Natural Hazards and Earth System Sciences, 21*(3), 893–916.

Bloemendaal, N., Haigh, I. D., de Moel, H., Muis, S., Haarsma, R. J., & Aerts, J. C. (2020). Generation of a global synthetic tropical cyclone hazard dataset using storm. *Scientific Data, 7*(1), 1–12.

Bruyère, C., Holland, G., Prein, A., Done, J., Buckley, B., Chan, P., Leplastrier, M., & Dyer, A. (2019). Severe weather in a changing climate. *Insurance Australia Group and National Center for Atmospheric Research, November. https://www. iag. com. au/sites/default/files/documents/Severe-weather-in-a-changing-climate-report-011119. pdf.*

Chen, Y., Gao, S., Li, X., & Shen, X. (2021). Key environmental factors for rapid intensification of the south china sea tropical cyclones. *Frontiers in Earth Science, 8*, 727.

Courtney, J. (2009). Adapting the knaff and zehr wind-pressure relationship for operational use in tropical cyclone warning centres. *Australian Meteorological and Oceanographic Journal, 58*(3), 167.

Courtney, J., & Burton, A. (2018). Joint industry project for objective tropical cyclone reanalysis: Final report. *Bureau of Meteorology, 87pp.*

Courtney, J. B., Foley, G. R., van Burgel, J. L., Trewin, B., Burton, A. D., Callaghan, J., & Davidson, N. E. (2021). Revisions to the australian tropical cyclone best track database. *Journal of Southern Hemisphere Earth Systems Science, 71*(2), 203–227.

Eberenz, S., Stocker, D., Röösli, T., & Bresch, D. N. (2020). Asset exposure data for global physical risk assessment. *Earth Syst, 12*. https://doi. org/10.5194/essd-12-817-2020

Gao, J. (2020). Downscaling global spatial population projections from 1/8-degree to 1-km grid cells. *Technical Notes NCAR, National Center for Atmospheric Researcher, Boulder, CO., USA.* https://doi.org/10.7927/q7z9-9r69

Hall, T. M., & Jewson, S. (2007). Statistical modelling of north atlantic tropical cyclone tracks. *Tellus A: Dynamic Meteorology and Oceanography, 59*(4), 486–498.

Harper, B. (2002). Tropical cyclone parameter estimation in the australian region. *Systems Engineering Australia Pty Ltd for Woodside Energy Ltd, Perth, 83.*

Holland, G. J. (1997). The maximum potential intensity of tropical cyclones. *Journal of the atmospheric sciences, 54*(21), 2519–2541.

Jones, B., & O'Neill, B. C. (2020). Global one-eighth degree population base year and projection grids based on the shared socioeconomic pathways. *Palisades*, (Revision 01).

Knutson, T., Camargo, S. J., Chan, J. C., Emanuel, K., Ho, C.-H., Kossin, J., Mohapatra, M., Satoh, M., Sugi, M., Walsh, K., et al. (2020). Tropical cyclones and climate change assessment: Part ii: Projected response to

anthropogenic warming. *Bulletin of the American Meteorological Society*, *101*(3), E303–E322.

Kossin, J. P. (2018). A global slowdown of tropical-cyclone translation speed. *Nature*, *558*(7708), 104–107.

Kossin, J. P., Emanuel, K. A., & Vecchi, G. A. (2014). The poleward migration of the location of tropical cyclone maximum intensity. *Nature*, *509*(7500), 349–352.

Lee, C.-Y., Tippett, M. K., Sobel, A. H., & Camargo, S. J. (2016). Rapid intensification and the bimodal distribution of tropical cyclone intensity. *Nature Communications*, *7*(1), 1–5.

O'Neill, B. C., Kriegler, E., Ebi, K. L., Kemp-Benedict, E., Riahi, K., Rothman, D. S., van Ruijven, B. J., van Vuuren, D. P., Birkmann, J., Kok, K., et al. (2017). The roads ahead: Narratives for shared socioeconomic pathways describing world futures in the 21st century. *Global environmental change*, *42*, 169–180.

O'Neill, B. C., Kriegler, E., Riahi, K., Ebi, K. L., Hallegatte, S., Carter, T. R., Mathur, R., & van Vuuren, D. P. (2014). A new scenario framework for climate change research: The concept of shared socioeconomic pathways. *Climatic change*, *122*(3), 387–400.

Riahi, K., van Vuuren, D. P., Kriegler, E., Edmonds, J., O'Neill, B. C., Fujimori, S., Bauer, N., Calvin, K., Dellink, R., Fricko, O., Lutz, W., Popp, A., Cuaresma, J. C., KC, S., Leimbach, M., Jiang, L., Kram, T., Rao, S., Emmerling, J., . . . Tavoni, M. (2017). The shared socioeconomic pathways and their energy, land use, and greenhouse gas emissions implications: An overview. *Global Environmental Change*, *42*, 153–168. https://doi.org/10.1016/j.gloenvcha.2016.05.009

---

## Author Comment (AC3)

**Answers to Referee 3**

April 15, 2022

**1 Specific Comments**

**Literature/ Simplicity**  Line 17: I disagree that we are lacking tools to assess impacts of future TCs. See for example Geiger et al. (2021)

Line 7 and Line 390: I disagree with the claim that the framework is 'a simple solution'. The framework requires expertise across multiple disciplines.

Thank you, in the revised version we will remove the mention of simple solution, include the reference provided and rephrase the text as follows: *Tools to assess the impact of future cyclones in shared socioeconomic pathways are starting to appear in the literature, for example, Geiger et al. (2021) evaluate the population exposure. Our study, instead focuses on tropical cyclones damage costs with the aim to include these advanced signals in integrated economic modeling.*

**Assertion l32-34**  It seems odd to make this assertion in the introduction without any supporting evidence. I suggest reframing this statement as a hypothesis to be tested.

Your are correct to point that this sentence requires supporting evidence. We propose to rephrase it as follows: *Recently, Bloemendaal et al. (2020) developed a modeling framework to simulate realistic synthetic tropical cyclone tracks: the Synthetic Tropical cyclOnes geneRation Model (STORM). This model computes the maximum pressure intensity (MPI) associated to the sea-surface temperature (SST), and uses this potential as a predictor in the central pressure dynamics (James & Mason, 2005). In line with Merrill (1987), we find that although the sea-surface temperature plays a major role, this variable alone is not a reliable predictor of whether a given storm will intensify. Thus, we prefer to rely on Holland (1997) formulation and model the effect of climate change on the maximum potentials in the different scenarios through a better description of the underlying thermodynamic phenomenon, well described by Emanuel (1988) and Holland (1997) or Emanuel (1999).*[1]

Indeed, the maximum pressure drop and MPI in STORM are defined using the sea-surface temperature only (cf. Eq (4) in the next question). On the other hand, Merrill (1987) suggests, on the contrary, that this predictor is material
* * *
[1]See section 3.3.3. for further details.

as a capping function but does not provide (alone) a good indication whether a given storm will intensify:

> "*Empirical studies by Miller (1958) and Merrill (1988) and theoretical results of Emanuel (1986) imply that SST specifies an upper bound on tropical cyclone intensity. The SST capping function is developped for the period 1974-1985 (Fig. 3) by tabulation maximum winds by 0.5 degree climatological (Reynolds, 1982) SST classes and computing the 99th percentile (Fig. 4)[...]*
>
> *The reason for treating SST as a capping function rather than as a direct predictor is evident from Fig. 4. Compare the top three curves (90th percentile and greater intensity) which increase sharply above 27°C with the median (50th percentile) which is nearly above 26°C. Knowing the climatological SST reveals little about the intensity of the average storm but much about the extreme intensity likely to occur*" (Merrill, 1987, pp. 11–12).

Figure 1 provides the Figures 3 and 4 of the empirical analysis evoked in the quote above (Merrill, 1987, pp. 11–12) and the Figures computed in the context of our study that led us to the same conclusions. In our work, we plotted the maximum pressure drop only (and not all percentiles) to lighten the graph which already accounts for multiple basins. We will add some elements from this quote in a note in section 3.3.3.

Figure 1: SST-MPD relationship

Merrill (1987, pp. 11–12)

[Figure]

Fig. 3. Scatter plot of relative intensity versus climatological sea surface temperature. The SST capping function based on the 99th percentile (see Fig. 4) is superimposed.

Fig. 4. Percentiles (values as in Fig. 1) fitted to the intensity-SST data shown in Fig. 3. The 30°C class (29.5°–30.5°) contains very few observations and is probably not representative of the extreme intensities possible.

CATHERINA

[Figure]

[Figure]

**Unclear comparison with STORM** This is perhaps my most important comment. I don't think the difference between your TC model and STORM is made clear enough. STORM appears to use the same SST-pressure drop relationship as you do, and STORM also uses MPI (calculated using the Bister and Emanuel formulation) to limit TC intensification. I don't understand what is new in your TC intensity formulation. Please clarify exactly what is new in the text. Is it the use of local MPI and SST along the synthetic tracks?

Thank you very much for this comment, the distinction is indeed quite unclear in the current version. In Bloemendaal et al. (2020), the MPI is defined subtracting the maximum pressure drop (MPD) from the normal environmental pressure (MSLP). Prior to that, the MPD is defined as a function of the SST (rounded and grouped per 0.1 bins) grouped spatially. The definition of the MPD is very similar in STORM and CATHERINA, but the role of this quantity differs in the two models. In CATHERINA (in line with Merrill (1987) observation), the MPD is not used in James and Mason (2005) dynamic equation but only as an upper bound for the central pressure. Let us clarify the main differences.

Bloemendaal et al. (2020) describe the construction of the MPI instrumental variable as follows:

> "*[...], we group these monthly mean SSTs in 0.1 °C bins together with their corresponding pressure drop values. We then fit the mean SST and maximum pressure drop per bin to Eq. (4):*
>
> $$P^{env} - P = A + Be^{C(SST-T_0)}, \quad T_0 = 30C \quad (4)$$
>
> *The coefficients A, B and C are estimated using non-linear least-squares. Using this formula, we can calculate the maximum pressure drop at every 0.25° × 0.25° SST grid point. In the final step, **we subtract this maximum pressure drop from the $P^{env}$ fields to derive the MPI.*** (Bloemendaal et al., 2020, p. 5).

Therefore, the MPIs are defined statistically subtracting MPD (function of SST), grouped on a 0.25x0.25 grid, and Bister and Emanuel (2002) values are used to bound their values only:

> "*To inhibit unrealistically low MPI values, the MPI values are bounded by the lowest MPI value per basin and per month derived by Bister and Emanuel (2002)*" (Bloemendaal et al., 2020, p. 5).

Table 1 reiterates the main differences of the two approaches. There is another minor change in the definition of the MPD that results from the use that will be made of this variable. The reason for relaxing the spatial constraint (fitted on 0.25 grid) is that we wanted to enlarge the groups on which we maximized the pressure drops (which have the effect of increasing the maximums). This choice follows from 'the role' of MPD in our process, not as the intensification factor but as a maximum obtainable pressure drop (Merrill, 1987) for a given temperature in each basin. These explanations will be includded in the revised version (section 3.3.3).

Table 1: Comparison STORM-CATHERINA cyclone intensification module

| | STORM | CATHERINA |
|---|---|---|
| Nb of variables | 2 | 4 |
| MDP definition | SST-MPD relationship - Eq. (4) 0.25x0.25 0.1°C bins | SST-MPD relationship - Eq. (4) basin 0.1°C bins |
| MDP use | Infer MPI | Cap basin pres. drop |
| MPI definition | MPI = $P^{env}$ – MPD | Holland (1997) |
| Unrealistic values | Bister and Emanuel (2002) | MPD |

**Evidence of 'better description'**  On a related note, the paper highlights the importance of this new representation of the thermodynamic influence, and makes claims on lines 43-45 that is it better, but this has not been demonstrated. Is it possible (if not too onerous) to run projections with and without this new representation of thermodynamic influence to demonstrate its importance.

The sentence line 43-45 is indeed not clear enough. The better performance of the inclusion of thermodynamic variables concerns the intensification process rather than the long term risk assessment. We propose to rephrase this sentence as follow: *Coupling STORM methodology with an extended the thermodynamic module fitted on four climate variables and CLIMADA, our approach provides a novel long-term tail risk assessment at a national level, providing an adaptive framework to estimate investments required to mitigate this risk.*

Concerning the better statistical significance (of the four variables MPI from Holland (1997)), we compared statistics of the non-linear fits using the two instrumental variables, i.e. conducted the same experiment with two candidates:

- Holland (1997) MPI derived using 4 climate variables (reiterated section 3.3.2 of the manuscript) and,

- the MPD from SST/MPD relationship used in Bloemendaal et al. (2020)

We used our definitions given in Table 1 and the results are provided in Table 2.We posed the following autoregressive stochastic depression dynamics (James & Mason, 2005):

$$\Delta P_t^c = c_0 + c_1 \Delta P_{t-1}^c + c_2 e^{-c_3[P_t^c - X_t]} + \varepsilon_t^P, \tag{1}$$

$$\varepsilon_t^P \sim \mathcal{N}(0, \sigma_{P^c}), \tag{2}$$

where the distance to maximum potential, $P_t^c - X_t$, can be computed either with our four dimensional MPI, or with the MPD depending on the SST only. Table 2 presents the estimates, confidence intervals and significance level of the relationship. The relationship fitted with the MPI that includes tropopause temperature, relative humidity has narrow confidence intervals, and all parameters are statistically significant. On the other hand, when we use the maximum

pressure drop (MPD), we obtain much less significant results fitting equation (1). The confidence interval of the parameters are wider, and $c_2$ (multiplicative factor associated with the exponential term) is not statistically significant.

Table 2: James and Mason (2005) depression dynamics (Equation (1)) parameters , standard errors and confidence intervals estimated using nonlinear least squares on the full sample.

|  | Estimate | Std. Error | 2.5% | 97.5% | Signif. |
|---|---|---|---|---|---|
|  | Using MPI in the exponential | | | | |
| $c_0$ | 17.966 | 3.1618 | 11.770 | 24.162 | *** |
| $c_1$ | -0.516 | 0.00274 | -0.522 | -0.510 | *** |
| $c_2$ | -19.528 | 3.180 | -25.760 | -13.295 | *** |
| $c_3$ | 0.00776 | 0.00134 | 0.005 | 0.010 | *** |
|  | Using MPD in the exponential | | | | |
| $c_0$ | -25.961 | 11.809 | -27.982 | -8.073 | * |
| $c_1$ | -0.415 | 0.004 | -0.423 | -0.407 | *** |
| $c_2$ | 0.753 | 1.585 | -0.187 | 0.323 |  |
| $c_3$ | -0.003 | 0.002 | -0.008 | -0.002 | * |

Signif. codes:  0 '***' 0.001 '**' 0.01 '*' 0.05 '.' 0.1 ' ' 1
The MPI and MPD are computed based on the climate data extracted along tracks from ERA-5, and the depression dynamics are derived from IBTrACS

Table 3: Coefficient correlation

|  | $c_0$ | $c_1$ | $c_2$ | $c_3$ |
|---|---|---|---|---|
| $c_0$ | 1,0000 | 0,0071 | -0,9998 | -0,9937 |
| $c_1$ | 0,0071 | 1,0000 | -0,0066 | -0,0077 |
| $c_2$ | -0,9998 | -0,0066 | 1,0000 | 0,9929 |
| $c_3$ | -0,9937 | -0,0077 | 0,9929 | 1,0000 |

We propose to include this comparison with STORM in a technical appendix.

**Calendar year / favorable conditions**   It's not clear to me how you calculate local SST and MPI along the synthetic tracks. If I am correct, the synthetic track generation samples from the IBTrACS record. If so, how do you assign a calendar year to each synthetic track to extract SST and MPI (from either ERA5 or CMIP)? If it's a random year then the environment might not necessarily be favorable for the synthetic TC (i.e., too cool SST or low MPI).

The process of generation of tropical cyclones is the following. For each year between two dates (2075 and 2100 for example), we sample a number of cyclones per basin[2] following the Poisson law with parameters provided in Bloemendaal et al. (2020). For each event, we retrieve a latitude, longitude and month resampling the IBTrACS past distributions. Therefore, cyclones are generated in similar months as historically observed cyclones (cf. Fig 7 of the manuscript) . Then, the starting day and hour of the day are randomly attributed so the tracks can be defined with a three hours step. It is true that with this procedure some 'candidate tracks' can be initiated in a location, or in a year which is less favorable for intensification. This would have the effect of underestimating the number of cyclones in the simulations. On the other hand, the parameters $\lambda_B$ would have been smaller if estimated using our filtered database on TCs exceeding 35 m/s. However, we maintain Bloemendaal et al. (2020) parameters to compensate the fact that some events will be generated in conditions not favorable for the development of cyclones, and be cleared out of the database. Overall, we obtain relatively similar landfall counts per basin in the simulations as in the historical dataset. In the revised version we will consider adjusting the cyclone frequency to match the landfall counts from the historical dataset precisely.

**Resolution**   ERA5 is still too coarse resolution to capture the most intense TCs. I suggest on Line 110 to change to 'better resolves than climate models'.

Thank you, we made the suggested change.

**Shifted extraction**   Line 110-113: Your method to use data away from the storm center is fine but I don't think it's necessary. You are using monthly data that should smooth out the influence of TCs. This is just a comment – I'm not suggesting to make a change.

Indeed, with the current spatial and temporal resolution, this translation is mainly made for reasons of theoretical coherence. In future studies, this model will be applied with higher temporal resolution and performing this translation would be more important.
* * *
[2]14.5 for the East Pacific (EP), 10.8 for the North Atlantic (NA), 2.0 for the North Indian (NI), 12.3 for the South Indian (SI), 9.3 for the South Pacific (SP), and 22.5 for the West Pacific (WP). In a footnote we also precise that: The parameters $\lambda_B$ would have been smaller if estimated using our filtered database. However, we maintain these parameters to take into consideration the fact that some events will be generated in conditions not favorable for the development of cyclones, and be cleared out of the database.

**ERA5 Availability** Line 117: I note that ERA5 is now available back to 1950, but is considered preliminary.

We will include this remark that could allow us to increase the fitting period. However, as climate change affects the values of the parameter we might prefer focusing on recent historical period.

**North Indian basin** Line 122: Please be more descriptive of what you mean rather than the ambiguous term 'erratic'.

The trajectories in North Indian basins are not well captured by our statistical framework. For displacement, the latitude and longitude description are less statistically significant (Tables A3 and A4). For the maximum pressure drop the relationship is not statistically significant (Table A6). We will include this description in the revised manuscript.

**Figure 3** I'm not sure what I learned from Fig. 3. I think this can be removed.

We will place this figure in the appendix. The aim of this figure is mainly to compare the depression dynamic produced (Figure 12) to existing (and most famous) ones.

**Frequency** Section 5: I think it would be useful to remind readers that you are keeping TC frequency and genesis distribution constant.

In the revised version, we reiterate that genesis frequencies are kept constant in section 5.

**3 steps before decay** Line 278-279: Please further explain why you wait 3 steps before applying the decay.

This step is inspired from Bloemendaal et al. (2020): *When the TC eye is over land for at least three time steps (totaling 9 hours), the decay in TC wind speed in the STORM is modelled following Kaplan and DeMaria (1995).* The decay function we use was introduced in Kaplan and DeMaria (1995) who showed that it provides an acceptable approximation for $t_L \geqslant 12h$. As each time step is 3 hours, we let the TC intensity be driven by Eq. (1) the 9 first hours and apply the decay function for $t_L \geqslant 12h$, e.g. after 3 steps.

**2 Technical Corrections**

- Fig 1: Correct 'Tranform' to 'Transform'

- corrected

- I don't see a reference in the text to Figure 5.

- added

- Figure 8: Please explain the distinction between the red shading vs. the red tracks.

- red sheding build with heatmap from IBTRACs red tracks are generated with CATHERINA

- Line 81: Please correct 'AOCGM' to 'AOGCM' and expand the acronym.

- done

- Line 273: Correct 'Algorithm 1' to 'Figure 1'.

- There should be an Algorithm close to this section (which was sent at the end of the paper is this format)

- The reference to Figure 18 should be to Figure 17. If I am correct, then I'm also not seeing a reference in the text to Figure 18.

- corrected

**References**

Bister, M., & Emanuel, K. A. (2002). Low frequency variability of tropical cyclone potential intensity 1. interannual to interdecadal variability. *Journal of Geophysical Research: Atmospheres*, *107*(D24), ACL–26.

Bloemendaal, N., Haigh, I. D., de Moel, H., Muis, S., Haarsma, R. J., & Aerts, J. C. (2020). Generation of a global synthetic tropical cyclone hazard dataset using storm. *Scientific Data*, *7*(1), 1–12.

Emanuel, K. A. (1988). The maximum intensity of hurricanes. *Journal of the Atmospheric Sciences*, *45*(7), 1143–1155.

Emanuel, K. A. (1999). Thermodynamic control of hurricane intensity. *Nature*, *401*(6754), 665–669.

Emanuel, K. A. (1986). An air-sea interaction theory for tropical cyclones. part i: Steady-state maintenance. *Journal of Atmospheric Sciences*, *43*(6), 585–605.

Geiger, T., Gütschow, J., Bresch, D. N., Emanuel, K., & Frieler, K. (2021). Double benefit of limiting global warming for tropical cyclone exposure. *Nature Climate Change*, *11*(10), 861–866.

Holland, G. J. (1997). The maximum potential intensity of tropical cyclones. *Journal of the atmospheric sciences*, *54*(21), 2519–2541.

James, M., & Mason, L. (2005). Synthetic tropical cyclone database. *Journal of waterway, port, coastal, and ocean engineering*, *131*(4), 181–192.

Kaplan, J., & DeMaria, M. (1995). A simple empirical model for predicting the decay of tropical cyclone winds after landfall. *Journal of applied meteorology*, *34*(11), 2499–2512.

Merrill, R. T. (1987). *An experiment in statistical prediction of tropical cyclone intensity change* (Vol. 34). US Department of Commerce, National Oceanic; Atmospheric Administration.

Merrill, R. T. (1988). Environmental influences on hurricane intensification. *Journal of Atmospheric Sciences*, *45*(11), 1678–1687.

Miller, B. I. (1958). On the maximum intensity of hurricanes. *Journal of Atmospheric Sciences*, *15*(2), 184–195.

Reynolds, R. W. (1982). *A monthly averaged climatology of sea surface temperature* (Vol. 31). US Department of Commerce, National Oceanic; Atmospheric Administration . . .

---

## Author Response (AR1)

**Response to Referee**

Dear Referees,

We would like to reiterate our thanks for your time and helpful comments. We did our best to take them all into account in this new version.

In the present document we provide a detailed point-by-point response to all referee comments and specify all changes in the revised manuscript. Our response to the Referees is structured as follow: *(1) comments from Referees,* (2) response to interactive discussion, (3) our final changes in manuscript. When we refer to a line, we refer to the file with the track of the modifications.
* * *
**REFEREE 1**

**RC1.1:** *It is not clear that this tropical cyclone model leads to accurate forecasts of storms. Changes in wind patterns can have no effect in this model. Is the cyclone model in this paper as accurate as the models developed by Emanuel? Or is this model a step backwards?*

**AC1.1: Interactive discussion.** We reiterate that building precise cyclone forecasts is not the aim of CATHERINA. We propose an algorithm designed to assess the future cost distribution for country-level damage assessment. The cyclone intensification process used is inspired from STORM model (Bloemendaal et al., 2020), which includes a single climate variable, and extended following Holland (1997) and Emanuel (1988) to encompass 2 more variables. We found that this extension provides statistically significant instrumental variables in the description of tropical cyclone intensification, which is the aim of the algorithm. Another step forward in our methodology is the use of state-of-the-art bias correction module for integrating climate model projections.

Consequently, even if some thermodynamical processes have been simplified in this approach, our approach is still a step forward with respect to the existing state-of-the-art in the context integrated assessment model (IAM) for climate impact analysis. Indeed, our approach can integrate any CMIP simulation with limited set of available variables (only a few vertical levels, some only available at monthly time scale, with some variables not always available). The adaptability of our algorithm to any CMIP exercise and simulation comes with a constraint implying necessary simplifications. Our approach combined with a bias correction module makes our algorithm easy to implement, more sophisticated in terms of processes included with respect to existing IAM and bias-corrected. We will add this explanation in the section 3.2.3 of revised version of the manuscript.

**AR1.1: Final response:** As mentioned in the interactive discussion, we added disclaimers in the introduction (texdiff file line 79) and conclusion (line 668) to reiterate the main objective of the paper. We also reiterated the initial objective of creating signals

material for financial practitioners in the abstract (in a bottom-up fashion) (texdiff file line 10).

***RC1.2:*** *The estimates of the effect of each hurricane are crude.  The model assumes that all damage is from wind whereas only 40% of cyclone damage is wind related.  Another 40% of cyclone damage is from storm surge. But storm surge strikes largely just the coastline.  The remaining 20% of damage is from excess precipitation which often falls far from where the cyclone strikes land.*

**AC1.2: Interactive discussion**. The model does not distinguish sub-perils, associated with key thermodynamical processes of cyclones (heavy precipitation, storm surge and associated flooding, strong winds) but instead uses a statistical relationship to estimate the global damage induced by a cyclone from a proxy variable given by the maximum wind speed (which is the proxy used in Saffir-Simpson Hurricane wind scale to define the intensity of a cyclone). We will include these elements in the third section (3.2.1) of the paper. The damage function is fitted on multiple events from the total damage reported in the global disaster database EM-DAT (Guha-Sapir et al., 2018). This database, used in most studies on the topic, accounts for the total reported damage (sum over all sub-perils) and does not distinguish damages from sub-perils. So, despite the relevance of the reviewer's comment, for our application (see our comment in the introduction of our reply, which are now included in the manuscript introduction to clarify the context of our study), distinguishing the sub-perils generating the impacts is not needed.

**AR1.2: Final response.** We describe explicitly the sources of damage / sub-perils in section 4.1 (line 451). However, as mentioned, we cannot integrate this information because EMDAT does not provide the split according to those sub-perils.

***RC1.3:*** *The model depends a great deal on the damage function. But it is not clear how this damage function was estimated.*

**AC1.3: Interactive discussion.** We use region-specific damage functions from Eberenz, Lüthi, et al. (2020).  This method uses a parametric function following Emanuel (2011). The parameters are estimated for each region with machine learning techniques from the reported damage estimates in the International Disaster Database (EM-DAT) Guha-Sapir et al. (2018) crossed with cyclone tracks (IBTrACS), and geographic and socio-economic information along these tracks.

We recall the main steps of the optimization performed by Eberenz, Lüthi, et al. (2020) and Lüthi (2019) to define the regional damage functions. The authors first defined the event damage ratio (EDR) as a fraction error between normalized reported (NRD) and simulations (SED) for each cyclone and the total damage ratio (TDR) is defined in each region summing over events. For each event, there is a value for vh allowing to optimally calibrate the explicit damage function described in Emanuel (2011). Then, the authors proposed two complementary optimization methodologies to find the value of vh maximizing the prediction of the regional damages Eberenz, Lüthi, et al. (2020):

(i)     Root mean square fraction (RMSF), minimizing the spread of the event damage ratios (EDR) - defined as the ratio of simulated damage vs. reported damage.
(ii)    Total damage ratio (TDR), finding the value of vh, such that the ratio of total simulated damage - obtained summing over event damage - and total reported damage is closest to 1.

We will clarify the calibration of these functions in section 4.3. In particular, we will review the approach of Eberenz, Lüthi, et al. (2020) to find the values of vh (c.f. technical supplement).

**AR1.3: Final response.** We included the equation used by Eberenz et al. (2021) to define the optimal vh coefficient per region in section 4.3. We also refined the estimation introducing a correction ratio by country.

*RC1.4: The estimates of how national assets are distributed across space are crude. Light times population is not going to allocate national assets carefully. I am specifically concerned about how well they model the assets near the coast.*

**AC1.4: Interactive discussion.** We chose to build our model based on state-of-the-art estimates, in such a way that the methodology is uniform country-wise. This dataset (Eberenz, Stocker, et al., 2020) is also used for the calibration of damage functions in Eberenz, Lüthi, et al. (2020) (discussed in Q3). Therefore, using this data allows to estimate the exposure in a consistent manner. To verify the accuracy of estimation, a back-test has been performed (Section 4.4). As we mention in the beginning, the only way to improve the estimates of asset value distribution would be to use the actual asset distribution from asset-level databases, but such databases are not yet available at the global scale. We will add this explanation when introducing the exposure dataset in section 2.4.

**AR1.4: Final response.** We added a remark in line with this concern in section 2.4 and 4.4 to stress that regional application would need to be refined by both finer asset allocation and specific damage calibration.

*RC1.5. The model appears to assume the spatial distribution of assets are fixed within a country.*

**AC1.5: Interactive discussion.** The model assumes that the spatial distribution varies with population changes proposed in the Socio-Economic Data Application Center dataset presented by Jones and O'Neill (2017, 2020). In particular, the spatial distribution of the population is different in varying shared socioeconomic pathways (SSPs). These projections are available with a one-eigth degree resolution.

**AR1.5: Final response.** The dynamic of population in SSPs is borrowed from the literature and was better exposed in the manuscript.

We added a map (Figure 20) showing that spatial population distribution clearly varies in SSPs (in particularly SSP3). The spatial distribution is therefore not fixed within countries.

*RC1.6 The paper does allow national assets to change over time, but they do not describe how this is done.*

**AC1.6: Interactive discussion.** To estimate future exposures along the cyclone track in each scenario, we use the downscaled estimation for the exposed wealth and the coefficients representing the change between the current state and the future scenario. We use the most granular projections of GDP per capita variation curves (Figure 2 - Data Source : https://tntcat.iiasa.ac.at/). Merging the two datasets (regional GDP per capita and local population) we build a dynamic projection of exposure factor.

**AR1.6: Final response.** We included a section to describe this issue in details (Section 2.6 and 4.5).

*RC1.7. There is no effort to measure adaptation by the country being hit or how that might change over time.*

**AC1.7. Interactive discussion.** Indeed, we left this question for further research. Supposing that adaptation increases with time alone would not be a relevant hypothesis. However, this question could be one of the direct applications of the model. For example, measuring the investment costs required to shift the value of vh (or vt) - and thus reduce the risk of future damage - can be a research question derived from this model simulations. In the revised paper, we will present more clearly the possible application of the model integrating the adaptation scenario, changing the values for the vulnerability parameter (vh and vt) in the section 4.2.

**AR1.7: Final response.** We added a discussion of this point in section 4.2.

*RC1.8. The initial forecasts of windspeed from the climate models are very inaccurate. The corrections appear to matter a great deal. However, these corrections have been made are on the historic data. So once they adjust historic data to actual historic outcomes, they do fine. But how well the model predicts future wind speeds is unclear.*

**AC1.8: Interactive discussion** Our bias correction approach is the standard in the climate community (see http://ccafs-climate.org/bias_correction/) . We do not have reanalysis data for the future. Therefore, there is no 'reference' value[1] to evaluate the prediction of the model. Therefore, we control the bias using the past distributions, where we can compare climate models and reanalysis and assume that errors between the two are similarly distributed in the future. We reiterate that this assumption is relatively classical in the climate community, and we will integrate these precisions in the paper in section 5.1.

**AR1.8: Final response.** We included the reference on bias correction (provided in the interactive discussion) in the manuscript.

*RC1.9. Figure 19 suggests the model predicts a small probability of very large damage but an expected value that is quite small. What explains this large tail to the distribution of damage? Is this simply the probability of a large storm striking a large coastal city? What is the expected value of damage?*

**AC1.9: Interactive discussion.** We ran the 7 models over 300 representative years to obtain these distributions. There is an effect due to certain large coastal cities exposure for the 'very unlikely' band (between 95 to 99 percentile) of annual damages. However, given the scale observed more than one city have been hit by storms. Because the aim of the model was also to stress test the resiliency of the financial and economic systems, looking at the expected value of damage was less interesting that studying the quantile value especially in the context of events with large tail risk. Coronese et al. (2019) investigating the increase of economic damage due to extreme natural disasters supports this thesis showing that the impact of climate change is particularly striking for
* * *
[1] Navarro-Racines, C., Tarapues, J., Thornton, P., Jarvis, A., and Ramirez-Villegas, J. 2020. High-resolution and bias-corrected CMIP5 projections for climate change impact assessments. Sci Data 7, 7, doi:10.1038/s41597-019-0343-8

extreme events (See for example, Coronese et al. 2019, Figure 2A). The table below contains the expected value of damage after bias correction.

The revised version will integrate this summary table with the expected value of the damage in the section 5.2 as well as the precisions above to explain the focus on quantiles in the visualization.

**AR1.9: Final response.** We included the comment above, the reference and the table of expected damage in the manuscript.

*RC1.10. Why does going from historic (1980-2020) to RCP2.5 lead to more damage than going from RCP2.5 to RCP8.5? Going from historic temperature to RCP2.5 is a 1C increase whereas going from RCP2.5 to RCP8.5 is going from 2C to 5.4C? Given the assumption that wind speed increases more rapidly as sea surface temperature rises, this outcome is hard to understand.*

**AC.1.10: Interactive discussion** Socio-economic change leads to wider differences than climate change, and this was expected (cf. Mendelsohn et al. (2012), Figure 3 for example). The explanation for this is contained in the dynamics of (i) GDP and (ii) population in SSPs. In the revised version we add further explanation about this result including more references to discuss the results of our simulations.

**AR1.10: Final response.** We explained that socio-economic factors' contributions to shift in exposure is important (or even dominant) in the revised manuscript.

*RC1.11. How much confidence do the authors have that they understand the relative damage caused by tropical cyclones at the end of the century across countries? How much of this is simply assuming the same distribution as today?*

**AC1.11. Interactive discussion.** Thank you for this very interesting question. We can see in Figure 4 (20 in the paper) that the distribution across countries is different from one SSP to another. For example, we have sensibly the same distribution in SSP2 and SSP5 with a higher expected damage in SSP5 because of the growth hypothesis this scenario relies on. However, SSP3 (rocky road) or SSP4 (inequality) are distributed differently. The scenario emphasizing inequalities -and its interpretation by scientists in terms of (i) socioeconomic developments (Riahi et al., 2017) and (ii) population distribution (Jones & O'Neill, 2017) - increases damage concentration in the United-States. On the other hand, the rocky-road scenario, linked to higher and more rural population, lower GDP and national rivalry sees the damage more equally distributed on other nations. We integrate this precision in the final version.

**AR1.11: Final response.** The distribution of damage across countries varies in SSP, but this variation is somewhat reduced when introducing the correction factor in damage estimation.

*RC1.12. It is not likely that anyone could design adaptation measures from this study given the crudeness of both the tropical cyclone predictions as well as the damage predictions. Is there any reliable prediction of a change in tropical cyclone outcomes from current outcomes other than they will get uniformly more powerful?*

**AC1.12. Interactive discussion.** The current dataset - with low resolution data, and maybe not entirely sufficient number or realizations - might not be accurate enough to

calibrate adaptation measures. However, we believe that the framework presented here is perfectly adapted to project a dense set of trajectories, compute expected and damage percentile over the next decades and therefore measure the investment required for adaptation and mitigation measures in the next fifty years. This work also reflects a practical exercise not carried out until now, of cross-referencing the latest data sets, putting into perspective both the socio-economic and climatic development hypotheses, and carrying out a bottom-up, rather than top-down, damage calculation. The conclusion of the revised manuscript will mention the limits of the current application and better explain the scope of applicability of the model.

**AR1.12: Final response.** We added multiple disclaimers in the paper for the exposed results. We also added a correction factor to reduce the errors in damage modeling as well as other improvements relative to other referee comments. Furthermore, we reiterate that this model description paper presents an integrated methodology, rather than a fully operational software.
* * *
**REFEREE 2**

**_RC2.1._** _Table 1: the selection of GCMs used should be justified. This could be through reference to model performance literature for key parameters, a specific evaluation process or perhaps simply availability of required variables for the analysis (though I note the variables used are available for all CMIP5 models)._

**AC1.1. Interactive discussion.** The choice of the CGMs was driven by the availability of the variables of interest in the Copernicus Climate data store (CDS) in the representative concentration pathways used in the exercise (RCP 2.6, 4.5 and 8.5 W/m2) in both single pressure level and multiple pressure levels monthly data in the same ensemble (r1i1p1). We also aimed at having multiple regions represented.

**AR1.1: Final response.** We added this explanation in the manuscript (line 118).

**_RC2.2._** _Section 2.2: the MSLP from ERA-5 is sampled 500 km from the centre of the cyclone. Is the same done for the other variables? Since the data are sampled from monthly means, it's possible the sampled values may not accurately represent the conditions at the time of TC passage (especially relevant for variables with sharp gradients such as SST)._

**AC2.2. Interactive discussion.** We retrieve both pressure (MSLP) and humidity (RH) away from the center because TC maximum potential intensity (MPI) - through thermodynamic efficiency and moist entropy - arises from the deviations from the normal conditions.

We acknowledge that monthly averaging may indeed "smooth" values so that the data may not represent the conditions at the time of cyclone passage. Therefore, using monthly means, this translation is mainly made for reasons of theoretical coherence. In future studies, this model will be applied with higher temporal resolution and performing this translation would be more important. In the present version of our paper, because

the CMIP5 projections of the sea-level temperature were only available at monthly frequency in the CDS, we chose to perform the exercise using monthly data to illustrate our approach. In addition, the monthly sampled data allowed us to build a statistically significant description of the MPI in the historical period. The possibility of improving the model using high frequency data will be emphasized in the revised version of the manuscript.

**AR2.2: Final response.** We added this explanation in a footnote of the revised manuscript (line 139)

***RC2.3.*** *Section 3.2: Given the literature of TC track generation methods, comparison with common metrics is encouraged. Specifically, as landfall is critical to reliable performance of a damage model, it would be helpful to present a comparison of the observed and simulated landfall rates (see for example Hall and Jewson, 2007; Lee et al., 2018; Arthur, 2021). This would strengthen the quality of the track generation results significantly.*

**AC2.3. Interactive discussion.** In the revised version we will compute landfall rates and compare them to relevant results from the literature.

**AR2.3: Final response.** We added a comparison with past landfall rates of "damaging" cyclones (Figure 7). We note however that the framework can be improved by modulating number of initialized cyclones to match the exact average number of cyclones making landfall. This would require an additional optimization.

***RC2.4.*** *Eq 3 - note that most best track data used wind pressure relations (WPRs) to determine Pc. Typically the work flow involves determining the Dvorak T number, converting this to a sustained wind speed, followed by regionally-specific WPR to determine Pc. The conversion back to wind speed from reported Pc using a single WPR will introduce errors, as an array of WPRs are used to operationally estimate Pc, not only between basins but within basins as well (e.g. Harper, 2002; Courtney and Knaff, 2009; Courtney and Burton, 2018; Courtney et al. 2021).*

**AC2.4. Interactive discussion.** We acknowledge that the use of a single WPR introduces errors. [..]

**AR2.4: Final response.** We made the suggested correction and introduced the parameters at a basin level. We included the comment that it could still introduce an error alongside with the provided reference. We find that the basin level estimation is a sufficient proxy in the context of this illustration of the framework.

***RC2.5.*** *Eq 10 describes the dominant control on the maximum intensity of TCs (maximum pressure drop - MDP). This is tied only to SSTs. The model uses maximum potential intensity (MPI) to control the depression dynamics (i.e. intensification rates). The formulation of MPI is directly applicable to the problem of estimating the maximum intensity, accounting for factors beyond SST alone that control maximum intensity. This suggests using SST as the only predictor of the MDP is deficient.*

**AC2.5. Interactive discussion.** Indeed, we already acknowledge that the SST alone in not a good predictor of whether individual TC will intensify. Therefore, we use the thermodynamic definition in the cyclone dynamics specification. On the other hand, we still define a "MPD" taking the maximum observed pressure drop for a given SST across

all events in each basin. You are right to point out that this appears to be inconsistent. However, - we use this maximum depression (MPD) estimated over the historical period for a given sea surface temperature only to cap the depressions in the simulations, to avoid generating events intensifying beyond past observations and make the simulated tracks more realistic. This is a limitation of our approach; however, this is relatively common in "statistical" models. Alternatively, we could make the maximum depression depend on the four variables of interest, however, this would make estimation more difficult and reduce the significance of this statistic.

In the revised version we will compare our definition with an alternative definition of the MPD, using the thermodynamic definition used for the MPI, and substituting extreme values of temperature and humidity.

**AR2.5: Final response.**   The MPD serves as a capping value only. Therefore, we kept this statistical formulation in this version. However, to account for wider variation due to other factors, we added a translation upward to relax the constraint, and potentially allow storms to intensify further, accounting for those factors in the projection.

*__RC2.6.__ Further, Chen et al. (2021) suggest rapid intensification is dependent on dynamical (e.g. upper divergence and wind shear) as well as thermodynamical factors. While the difference between Pc and MPI is a factor in predicting rapid intensification, and the dynamical factors are probably accounted for by the random innovation (Eq. 12), these other dynamical factors should be acknowledged.*

**AC1.6. Interactive discussion.** Indeed, the components explaining the noise term in the pressure dynamics should be better identified, and we will acknowledge them in the next version of the manuscript. However, in the context of our exercise we had to focus on explanatory factors that are available in the CMIP5 simulations which reduced our scope to thermodynamical factors.

**AR2.6: Final response.**  We clarified the description of the depression dynamics module in the revised version. Thanks to your comment (and those of R3) we could correct the fitting of the parameters used in the cyclone dynamics depression equation in this new version.

*__RC2.7.__ Apply CDF-t to model variables, then evaluate MPI - I suggest comparing quantiles of ERA5 MPI against the bias corrected CMIP MPI values to demonstrate the effect of bias correction. Q-Q plots would be an effective way to do this. One risk with this approach is that correcting individual variables may lead to unrealistic combinations when evaluating MPI - e.g. extremely low tropopause temperatures in combination with very high SSTs that lead to unrealistic lapse rates and therefore unrealistically large MPI. Two solutions present themselves: 1) apply the bias correction methods to calculated MPI or (2) consider the joint distributions of variables when evaluating the bias corrections.*

**AC1.7. Interactive discussion.** This is a very relevant point. Indeed, individual variables entering the MPI computation may be strongly correlated. In the revised version we will follow the reviewer's suggestion and apply bias correction directly to calculated MPI.

**AR2.7: Final response**. Indeed, the local thermodynamic potentials defined using variables corrected independently introduced a bias (b).

[Figure]

We made the correction on the thermodynamic efficiency which corrected the low bias of the MPIs generated by the algorithm. The final QQ-plot (over all model and basins) is represented in Figure (c).

***RC2.8.*** *The distributions of SST presented in Figure 16 do not appear representative of SSTs sampled in the vicinity of TCs, and is inconsistent with the distribution shown in Figure 10. SSTs of 26C (299K) are typically considered a lower bound for TC formation (Gray, 1979), but median values from the ERA5 are well below that - for example based on Figure 16 the median SST for the South Pacific basin along synthetic tracks is 290-292K, for the Western Pacific 295K. Only the N Indian basin has a median SST near 300K. This suggests that the synthetic tracks are traversing areas not typically covered by TCs, or occurring at the wrong time of year for the respective basin leading to the unusual SST distribution.*

**AC1.8. Interactive discussion.** The bias-correction module is indeed fitted on a larger range of climate conditions. For the genesis of the cyclones, the time of year and location are in line with historical cyclone data. However, in the bias-correction module, the synthetic tracks are generated without climate constraints, i.e. cyclones are allowed to drift relatively far away from their genesis location (in the limits of their initial basin), and therefore can cover conditions which do not lead to the formation of tropical cyclones. At this stage, these tracks are not to be considered as `TCs tracks' but as 'candidate' tracks. In the following stage, TC tracks will be generated from candidate tracks by filtering those ones where meteorological conditions for cyclone formation are satisfied.

**AR2.8: Final response.** We explained that the bias correction was done on a sub-sample build from track candidates in the new version.

***RC2.9.*** *Completely absent is any discussion on TC rates in the projections. Comprehensive literature reviews and expert elicitations indicate a global decline in TC frequency (albeit with generally low-medium confidence) (Knutson et al. 2020). Changes in TC rates will have a significant impact on the annualized losses. This is an important component that should be addressed.*

***RC2.10 .*** *In parallel, there is no discussion on changes in track behaviour. Observed trends in TC translation speed (Kossin, 2018) and poleward migration of maximum intensity (Kossin et al., 2014) should be considered in projections of TC activity. This has profound implications for TC-related risk in key marginal areas (e.g. Bruyere et al., 2020) where vulnerabilities are high, but present-day frequency of TCs is low.*

**AC1.9-10. Interactive discussion.** These two comments will be included in the conclusion of the revised version of our paper as they reflect important limitations of our exercise. Indeed, we kept the genesis rates constant for each basin. The number of cyclones each year are drawn from Poisson distribution. It is possible to reduce the intensity parameter in the projections, and to introduce cyclones in regions where the present-day frequency is low, however, in this study, we focused on the changes in thermodynamic potentials. Moreover, as our approach is a statistical one, we had to focus on areas where relationships could be extrapolated from historical data. We will add a comment to account for this possible improvement.

**AR2.9-10: Final response** These comments were included in the new version.

*RC2.11.* Section 5.2: Consideration of SSPs in determining the effects on damage is novel, but the explanation is very limited. Given growth of exposure is constrained in existing high exposure regions, regional growth may not be in areas exposed to TC impacts.

**AC2.11. Interactive discussion.** In the revised version, we will provide more explanations about the shared-socioeconomic pathways used to project exposure. Indeed, we did not consider that areas subject to cyclones would face additional economic growth constraints in our projections. Historically, high exposure regions were not particularly constrained in terms of growth (e.g. the East Coast of the United States of America can be considered as a high exposure region as well as most regions in South Korea, Japan, Australia). In addition, climate change increases tropical cyclone intensity allowing them to reach regions where current TC impacts are low.

*RC2.12.* The description of the implementation of projections of local physical asset value dynamics is very limited, but probably the most novel part of the connected modelling system. There should be a more substantial discussion on how the SSP definitions are used to modify asset values.

**AR2.12: Final response** We detailed this in the new version (Figure 1, section 2.6, section 4.5)
* * *
**REFEREE 3**

*RC3.1.* Line 17: I disagree that we are lacking tools to assess impacts of future TCs. See for example Geiger et al. (2021)

*RC3.2.* Line 7 and Line 390: I disagree with the claim that the framework is 'a simple solution'. The framework requires expertise across multiple disciplines.

**AC3.1-2. Interactive discussion.** Thank you, in the revised version we will remove the mention of simple solution, include the reference provided and rephrase the text as follows: Tools to assess the impact of future cyclones in shared socioeconomic pathways are starting to appear in the literature, for example, Geiger et al. (2021) evaluate the population exposure. Our study instead focuses on tropical cyclones damage costs with the aim to include these advanced signals in integrated economic modeling.

**AR3.1-2: Final response.** Thank you, we made the corrections.

**RC3.3.** *Line 32-34: It seems odd to make this assertion in the introduction without any supporting evidence. I suggest reframing this statement as a hypothesis to be tested.*

**AC3.3. Interactive discussion.** You are correct to point that this sentence requires supporting evidence. […]

**AR3.3: Final response**. We rephrased this statement.

**RC3.4.** *This is perhaps my most important comment. I don't think the difference between your TC model and STORM is made clear enough. STORM appears to use the same SST-pressure drop relationship as you do, and STORM also uses MPI (calculated using the Bister and Emanuel formulation) to limit TC intensification. I don't understand what is new in your TC intensity formulation. Please clarify exactly what is new in the text. Is it the use of local MPI and SST along the synthetic tracks?*

**AC3.4. Interactive discussion** […]

**AR3.4: Final response.** The main differences between our approach and STORM are (i) the possibility to use the model with CGM projections to compute realistic trajectories of future cyclones in the context of climate change, whereas STORM focuses on generating cyclones with identical characteristics to those in IBTrACS database. This objective in particular required to introduce a state-of-the art bias correction module, adapted to the cyclone modeling exercise; (ii) the use of three climate variables (SST, tropopause temperature and relative humidity) in the formula describing cyclone intensification. It was necessary to include these additional variables because it has been demonstrated in the literature that they have a strong impact on tropical cyclones and will change with the advent of climate change.

**RC3.5.** *On a related note, the paper highlights the importance of this new representation of the thermodynamic influence, and makes claims on lines 43-45 that is it better, but this has not been demonstrated. Is it possible (if not too onerous) to run projections with and without this new representation of thermodynamic influence to demonstrate its importance.*

**AR3.5: Final response.** We acknowledge that the suggested exercise is of great interest and that measuring the sensitivity of the damages to the inclusion of each climate variable is a relevant research question. However, the main advantage of including additional variables is probably related to better representation of the future cyclone intensification processes in the context of climate change. Since for the future period we do not have a reference dataset of cyclone tracks to which alternative simulations could be compared, in our view, the choice of variables should rather be guided by theoretical considerations, and the literature suggests that TC intensification in not driven by SST only.

**RC3.6.** *It's not clear to me how you calculate local SST and MPI along the synthetic tracks. If I am correct, the synthetic track generation samples from the IBTrACS record. If so, how do you assign a calendar year to each synthetic track to extract SST and MPI (from either ERA5 or CMIP)? If it's a random year then the environment might not necessarily be favorable for the synthetic TC (i.e., too cool SST or low MPI).*

**AC3.6. Interactive discussion.** The process of generation of tropical cyclones is the following. For each year between two dates (2075 and 2100 for example), we generate

several cyclones per basin following the Poisson law with parameters provided in STORM:  14.5 for the East Pacific (EP), 10.8 for the North Atlantic (NA), 2.0 for the North Indian (NI), 12.3 for the South Indian (SI), 9.3 for the South Pacific (SP), and 22.5 for the West Pacific (WP)). These parameters would have been smaller if estimated using our filtered database. However, we maintain these parameters to take into consideration the fact that some events will be generated in conditions not favorable for the development of cyclones and be cleared out of the database. More precisely, for each event, we retrieve a latitude, longitude, and month re-sampling the IBTrACS past distributions. Therefore, cyclones are generated in similar months as historically observed cyclones (cf. Fig 7 of the manuscript). Then, the starting day and hour of the day are randomly attributed so the tracks can be defined with a three-hour step. With this procedure some 'candidate tracks' can be initiated in a location, or in a year which is less favorable for intensification.  This would have the effect of underestimating the number of cyclones in the simulations therefore we kept higher intensity values to compensate this effect. Overall, we obtain relatively similar landfall counts per basin in the simulations and in the historical dataset. In the revised version, we will consider adjusting the cyclone frequency to match the landfall counts from the historical dataset precisely.

**AR3.6: Final response.** We improved the description of the cyclone genesis module. A calendar year is indeed assigned to each synthetic cyclone. If the conditions are not satisfied (for e.g. too cool), the local MPIs would be closer to normal conditions (i.e. potential pressure drops smaller), and resulting cyclones will be less intense. We present the landfall counts for storms with wind speeds higher than 35m/s and show that they are similar to those observed in historical data.

*RC3.7.* *ERA5 is still too coarse resolution to capture the most intense TCs. I suggest on Line 110 to change to 'better resolves than climate models'.*

**AC3.7. Interactive discussion** We made the change

*RC3.8.* *Line 110-113: Your method to use data away from the storm center is fine but I don't think it's necessary. You are using monthly data that should smooth out the influence of TCs. This is just a comment – I'm not suggesting to make a change.*

**AC2.8. Interactive discussion** Indeed, with the current spatial and temporal resolution, this translation is mainly made for reasons of theoretical coherence. In future studies, this model will be applied with higher temporal resolution and performing this translation would be more important.

*RC3.9.* *Line 117: I note that ERA5 is now available back to 1950, but is considered preliminary.*

**AR3.9:** We made the precision line 623.

**AC3.9. Interactive discussion** We will include this remark that could allow us to increase the fitting period. However, as climate change affects the values of the parameter we might prefer focusing on recent historical period.

*RC3.10.* *Line 122: Please be more descriptive of what you mean rather than the ambiguous term 'erratic'.*

**AC3.10. Interactive discussion** The trajectories in North Indian basins are not well captured by our statistical framework. For displacement, the latitude and longitude description are less statistically significant (Tables A3 and A4). For the maximum pressure drop the relationship is not statistically significant (Table A6). We will include this description in the revised manuscript.

**AR3.10: Final response.** We rephrased the explanation. The lower significance of Indian basin trajectories is mainly due to lower number of observations, in particular when focusing on cyclones above 35m/s.

**RC3.11.** *I'm not sure what I learned from Fig. 3. I think this can be removed.*

**AC3.11. Interactive discussion:** We will place this figure in the appendix. The aim of this figure is mainly to compare the depression dynamic produced (Figure 12) to existing (and most famous) ones.

**AR3.11:** We removed this Figure.

**RC3.12.** *Section 5: I think it would be useful to remind readers that you are keeping TC frequency and genesis distribution constant.*

**AC2.12. Interactive discussion.** In the revised version, we reiterate that genesis frequencies are kept constant in section 5.

**AR3.12:** We made the precision line 623.

**RC3.13.** *Line 278-279: Please further explain why you wait 3 steps before applying the decay.*

**AC2.13. Interactive discussion.**This step is inspired from STORM : " *When the TC eye is over land for at least three time steps (totaling 9 hours), the decay in TC wind speed in the STORM is modelled following Kaplan & De Maria (1995)*". The decay function we use was introduced in Kaplan & De Maria (1995) who showed that it provides an acceptable approximation for t_L>12h. As each time step is 3 hours, we let the TC intensity be driven by the pressure dynamic module the 9 first hours and apply the decay function for t_L>12h, e.g. after 3 steps.

**AR3.13:** We made the precision line 411.

---

## Author Response (AR2)

**Cyclone generation Algorithm including a THERmodynamic module for Integrated National damage Assessment (CATHERINA 1.0) compatible with CMIP climate data**

Philippe Drobinski, Théo Le Guenedal and Peter Tankov

Answer to the review of the revised version

**Topical editor comment:** Please check the reviewer comments. Particularly, I want to ask you to make clear and well organized manuscript structures for better readability.

We thank the topical editor for the valuable comments. In the new version, we revised the structure of the paper to improve readability and in particular to better distinguish the modeling of (1) hazard intensity, (2) asset exposure, (3) damage modeling and (4) applications and results.

**Referee 1**

We thank the reviewer for the careful reading of our manuscript and for the insightful comments, which as we hope allowed us to improve the paper.

**Q1.** The modelers have gone to great effort to reproduce the historic climatology of global tropical cyclones over the last 40 years. The aggregate difference between their simulations of the last 40 years and the historic cyclones is small. There is no similar way to test whether the model captures the future behavior of cyclones in new climate conditions. What you can see is that the future cyclone results are uncertain. There is no clear statistical difference between cyclone characteristics under RCP2.6, RCP4.5, and RCP8.5 by the end of the century. But this may be because the true difference is small not necessarily because the model is flawed

It may have not been clear enough in the last version, but each of the seven climate models considered in this paper does predict significantly stronger winds in higher concentration pathways. To make this appear clearly, in the new version results are now grouped by model instead of by RCP (see Figure 23). Table 1 below shows the average maximum wind speed for each concentration pathway and for each model as well as the average over all models. We see that there is a increase of 5 m/s in terms of maximum wind speed between RCP26

and RCP45 and an increase of 8 m/s between RCP26 and RCP85, which is quite substantial.

Table 1: Mean of the maximum wind speed of synthetic cyclones in representative representation pathways

|  | RCP26 | RCP45 | RCP85 |
|---|---|---|---|
| ACCESS1-0 (BoM-CSIRO, Australia) |  | 66.42 | 70.21 |
| bcc-csm1-1-m (BCC, China) | 65.65 | 71.00 | 71.91 |
| CanESM2 (CCCMA, Canada) | 59.22 |  | 66.70 |
| GISS-E2-H (NASA, USA) |  | 71.79 | 75.02 |
| inmcm4 (INM, Russia) |  | 61.42 | 64.70 |
| IPSL-CM5A-MR (IPSL, France) | 60.70 | 63.89 | 69.61 |
| NorESM1-ME (NCC, Norway) | 57.40 | 59.02 | 63.32 |
| Mean | 60.74 | 65.59 | 68.78 |

**Q2.** The modeling of the damage from the cyclones does not seem nearly as careful as the modeling of the cyclones themselves. The paper makes an internally inconsistent assumption that emissions are independent of economic outcomes. However, that is not possible- just as it is not possible for solar radiation to increase without warming ocean temperatures. Without the 3% per year growth of GDP in SSP5, one cannot generate enough emissions to see RCP8.5.

We agree that, at a global level, some RCP-SSP couples cannot be obtained under the assumptions used in state-of-the-art integrated assessment models. In the new version, when illustrating our results at the global level we kept only the most meaningful SSP-RCP couples (see e.g. Figure 24). According to Rogelj et al. (2018) the SSP3 does not allow RCP26, and the RCP85 is reached only in the growth conditions of the SSP5 (cf Table 1) . Note however that SSP2 can by definition be obtained by combination of other SSP at regional level, and cyclone damage depends on regional population and GDP growth, so that it makes sense to consider the couples independently for regional analysis.

**Q3. Damage function**  The paper assumes that the aggregate damage from a cyclone increases with the cube of wind speed. But empirical research has shown that aggregate damage increases faster than the cube (Nordhaus 2010). Further, aggregate damage is more accurately predicted by minimum pressure than maximum wind speed (Mendelsohn et al. 2012).

Our paper uses state of the art damage functions of Eberenz et al. (2021), which are calibrated to optimally represent the fraction of value destroyed by a cyclone with a given wind speed. The damage functions are of the form

$$f(V, v_h^j) = \frac{\left(\max(V - v_0, 0)\right)^3}{(v_h^j - v_0)^3 + \left(\max(V - v_0, 0)\right)^3},$$

Figure 1: Rogelj et al. (2018) SSP-RCP matrix

[Figure]

**Fig. 5 | Variation of carbon prices over SSP and radiative forcing target space.** Values are shown as average global average carbon prices over the 2020–2100 period discounted to 2010 with a 5% discount rate. Mitigation challenges are assumed to increase from left to right across the SSPs (that is, SSP1, SSP4, SSP2, SSP3, SSP5). Each box represents one model–SSP–radiative forcing target combination. A, AIM/CGE; G, GCAM4; I, IMAGE; M, MESSAGE-GLOBIOM; R, REMIND-MAgPIE; W, WITCH-GLOBIOM. All scenarios with a carbon price greater than 0 (that is, all but the baselines) have been designed to reach one of the radiative forcing targets on the vertical axis. Models for which no baseline data are indicated have baselines that result in an end-of-century radiative forcing between 6.0 and 8.5 W m⁻².

where $v_0$ and $v_h^j$ are parameters to be calibrated. For wind speeds just above the threshold $v_0$, the local damage indeed increases as a third power of the *excess wind speed over threshold*, which is much steeper than the third power of the wind speed. On the other hand, for very strong winds the damage function converges to unity, meaning that all wealth may potentially be wiped out. However, the presence of the cubic excess-over-threshold function does not mean that the total damage will be proportional to the third power of the maximum wind speed over threshold - stronger cyclones will also spend more time over land, increasing the exposed value. Thus, the third power in our damage function cannot be compared with the ninth power in (Nordhaus, 2010). Besides, the very high damage exponent estimated in that paper clearly cannot be used in projections, since even a moderate increase of the wind speed will quickly lead to the entire world GDP being wiped out. Finally, the estimates of (Nordhaus, 2010) may be statistically biased because of missing variables such as local GDP density (as acknowledged by the author himself).

Furthermore, some authors do indeed recommend to use the central pressure as a proxy for damages, however the vast majority of the papers on storm damage estimation use the maximum wind speed (Cusack, 2013; Donat et al.,

2011; Elsner et al., 2011; Etienne & Beniston, 2012; Feuerstein et al., 2011; Koks et al., 2020; Leckebusch et al., 2007; Pinto et al., 2007; Schwierz et al., 2010), and state of the art damage functions available in the literature are based on this parameter. Since there is a strong relationship between central pressure and wind speed, using central pressure would not make a significant difference and would require calibrating the damage function for each region from scratch instead of using the results of Eberenz et al., 2021. We therefore choose to build upon the existing literature in the context of this illustration, and use the maximum wind speed as a proxy for damages.

**Q4.** It is not totally clear, but it appears that the authors have assumed damage will increase proportionally with GDP and population over time. The paper makes no mention of adaptation to tropical cyclones. However, it is quite clear that all countries but the USA have adapted to tropical cyclones (Bakkensen and Mendelsohn 2019). Damage has increased only slightly with GDP and population density in almost every country. Only in the USA have damages increased proportionally with GDP. Even in the USA, is it reasonable to assume that this will continue to 2100?

Indeed, the present framework does not capture future adaptation. We mention that it can be introduced through the local parameter $v_h^j$ of the damage function, and past adaptation is already included in the model through the calibration of this $v_h^j$. We can see through the shape of damage functions that the USA still suffer significant damage when the wind increases while other countries are more resilient (e.g., Philippines, China, Japan, South Korea, Hong-Kong and Taiwan). To better account for future adaptation and disentangle the impact of different risk components, we now introduce parameters $\alpha_1$ and $\alpha_2$ into the model for future exposed value:

$$\Phi(x,y,j,k,t) = \underbrace{(F_{\text{GDP}}^{cap}(j,k,t))^{\alpha_1}}_{\text{Global macro factor}} \cdot \underbrace{(F_{pop}(x,y,k,t))^{\alpha_2} \cdot \mathcal{L}_P(x,y)}_{\text{Local factor}}. \quad (1)$$

The choice $\alpha_1 = \alpha_2 = 1$ assumes linear increase of exposure both with respect to GDP per capita and with respect to local exposed population. Using $\alpha_1 = 1/3$, which is close to the value estimated in (Bakkensen and Mendelsohn, 2019), allows to account for possible future adaptation . The configurations $\alpha_1 = 1, \alpha_2 = 0$ and $\alpha_1 = 0, \alpha_2 = 1$ allow to decompose the risk contribution between GDP and exposed population. Finally, the choice $\alpha_1 = 0, \alpha_2 = 0$ uses the current exposure value with no socioeconomic growth factor.

Figure 2 and Table 2, which are also included in the new version of the paper, present the damage statistics for different combinations of $\alpha_1$ and $\alpha_2$. Assuming no future adaptation ($\alpha_1 = \alpha_2 = 1$), and using the historical simulations with ERA5 data as baseline, over the period 2070-2100, we find that the RCP 2.6 scenario, which is in line with the Paris Agreement and keeps global warming below $2°$C by 2100, involves a growth of expected global annual financial losses from tropical cyclones by a factor of 4.2 on average. Ignoring socioeconomic and population growth factors ($\alpha_1 = 0$ and $\alpha_2 = 0$) our model suggest that

the expected financial loss would grow by a factor of 1.6 due to increasing cyclone intensity. Taking into account adaptation, i.e. limiting the growth of damage with respect to GDP per capita ($\alpha_1 = 1/3$ and $\alpha_2 = 1$), the expected damage would grow by a factor 2.6. In the case of SSP2-RCP 4.5 (between 1.7 and 3.2°C warming by 2100) and SSP5-RCP 8.5 (between 3.2 and 5.4°C warming by 2100), the average expected damage will be multiplied by 5.4 and 14.2 respectively without adaptation. In the RCP8.5 the expected damage will still grow by a factor 2.8 ignoring the change in GDP per capita and population ($\alpha_1 = 0$ and $\alpha_2 = 0$).

Thus, even if the amount of wealth exposed to tropical cyclones in the future is the same as today, our model still projects that the expected annual damage will grow by a factor 1.6 in RCP2.6 and by a factor 2.8 in RCP8.5 due to increasing cyclone intensity.

**Q5.** The link between simulated tracks and expected future damage is not clear in the paper. The damage depends on the strength of the cyclone, whether it strikes at low or high tide, and whether it hits a major metropolitan area or not. Are there enough simulated cyclones to get the proportion of these different outcomes correct for each country?

If our understanding is correct, the question is whether the number of simulated cyclone trajectories is sufficient to reliably estimate expected future damage. First of all, the main objective of this submission to GMD is not to provide precise estimates of cyclone damage, but to present a new model for cyclone damage estimation. The illustrations in the paper were obtained by simulating 300 representative years for each of the 7 climate models. Since simulations from each model were done independently this gives us 2100 statistically independent samples for yearly cyclone damage. Using our code which is publicly available, anyone can simulate many more trajectories, this is just a question of time and computing power.

This being said, we are confident that at the global level and for large countries, which are hit by cyclones several times every year, the number of simulations is sufficient. We provide the standard errors in table 2 at the global level and it is clear that they are quite small compared to the average damage estimate with relative errors well below 10%. At country level (Table 3), the relative standard errors are higher (up to 20%) and we acknowledge that further launches would improve the results, especially for small countries.

More generally, our estimates are subject to three types of uncertainty: internal climate variability, climate model uncertainty and socio-economic uncertainty related to future exposure growth, adaptation measures and concentration pathways. The first two types of uncertainty are quantified by the standard errors in Table 2, while the last one may be evaluated by performing simulations under different assumptions on adaptation, SSP narratives and representative concentration pathways as illustrated in Figure 2.

**Q6.** The authors never explain how CATHERINA1.0 can possibly lead to an estimate of the risk to sovereign debt or other major macroeconomic outcomes.

We added a section to illustrate how the additional damage could be integrated in a simplified credit model (section 5.3).

**Q7.** Most of the future results in the paper do not come from a change in the cyclones themselves but rather in the damage each cyclone causes. This is entirely because the authors are assuming there will be a lot more in harm's way in the future. It is possible but not at all likely that the world will be this foolish.

We agree that socioeconomic factors strongly impact the future damage in accordance with the academic literature (Noy, 2016; Pielke Jr et al., 2008; Weinkle et al., 2018; Weinkle et al., 2012; Ye et al., 2020). However, we are confident that the evolution of future cyclone's intensity also plays a role. To illustrate this, we computed a damage projection assuming no GDP growth (with $\alpha = 0$). We can see in Table 2 that without GDP per capita or population factor the simulated damage increase by 1.6 in the SSP2-RCP26, by 2 in the SSP2-RCP45 and by 2.8 in SSP5-RCP85.

[Figure]

Figure 2: Expected value of global annual damage (in USD billions) in different SSP-RCP and exposure projection hypothesis configurations. The vertical dotted line corrsponds to historical simulations with ERA5 data.

Table 2: Simulation statistics (in USD billion)

[revised manuscript text omitted]

---

## Author Response (AR3)

**Cyclone generation Algorithm including a THERmodynamic module for Integrated National damage Assessment (CATHERINA 1.0) compatible with CMIP climate data**

Philippe Drobinski, Théo Le Guenedal and Peter Tankov

Answer to the review of the revised version

**Topical editor comment:**   Please carefully check units in Figures and Tables. (e.g., Fig. 20, Fig. 25, Fig. 27, Table 2, Table A4, etc) . Thank you for this remark. We have added information to the figures and tables mentioned and some minor corrections to the manuscript.